

# Matrix models and holography: Mass deformations of long quiver theories in 5d and 3d

**Mohammad Akhond[1*], Andrea Legramandi[2,3†],**
**Carlos Nunez[2‡], Leonardo Santilli[4,∘] and Lucas Schepers[2§]**

**1** Yukawa Institute for Theoretical Physics, Kyoto University, Kyoto 606-8502, Japan
**2** Department of Physics, Swansea University, Swansea SA2 8PP, United Kingdom
**3** Pitaevskii BEC Center, CNR-INO and Physics Department,
Universitá di Trento, I-38123 Trento, Italy
**4** Yau Mathematical Sciences Center, Tsinghua University, Beijing, 100084, China

* akhond@yukawa.kyoto-u.ac.jp , † andrea.legramandi@unitn.it , ‡ c.nunez@swansea.ac.uk ,
∘ santilli@tsinghua.edu.cn , § 988532@Swansea.ac.uk

## Abstract

We enlarge the dictionary between matrix models for long linear quivers preserving eight supercharges in $d = 5$ and $d = 3$ and type IIB supergravity backgrounds with $\text{AdS}_{d+1}$ factors. We introduce mass deformations of the field theory that break the quiver into a collection of interacting linear quivers, which are decoupled at the end of the RG flow. We find and solve a Laplace problem in supergravity which realises these deformations holographically. The free energy and expectation values of antisymmetric Wilson loops are calculated on both sides of the proposed duality, finding agreement. With our matching procedure, the free energy satisfies a strong version of the F-theorem.



# 1 Introduction

The cross-fertilisation between AdS/CFT [1] and Matrix Models has a long history. Matrix model techniques have been instrumental in checking holographic results and suggesting ideas for string theory calculations. A selection of early papers [2–8] shows various instances in which exact results in field theory, calculated using matrix models, were matched with calculations in supergravity. The field experienced a rapid growth with the advent of supersymmetric localisation [9], with an impressive amount of refined checks of dualities between AdS geometries and CFTs on a sphere performed in three [10–21], four [22–30] and five dimensions [31–37].

Let us focus on $d$-dimensional superconformal field theories (SCFTs) preserving eight supercharges, whose dual supergravity solutions are geometries containing $\text{AdS}_{d+1}$ factors. The construction of the ten-dimensional configurations, consisting of a metric, Ramond and Neveu–Schwarz background fields, has been systematised for the case of eight preserved supercharges for $d = 1$ [38–40], $d = 2$ [41–47], $d = 3$ [48–57], $d = 4$ [58–64], $d = 5$ [65–75] and $d = 6$ [76–81].

Based on localisation of the path integral [9], for every SCFT of the class mentioned above which is connected to a gauge theory via Renormalization Group (RG) flow, matrix models have been developed that accurately calculate various observable quantities that preserve a fraction of the supersymmetry, most notably the free energy and Wilson loop expectation values. These matrix models encode exact information in principle, but their intricacies grow with $d$ and with the complexity of the quiver. Whilst, in various cases, large ranks or long

quiver approximations are needed to solve the matrix model, these usually coincide with the regime of validity of the dual supergravity background.

Five-dimensional $\mathcal{N} = 1$ SCFTs are inherently strongly coupled [82] and do not admit a Lagrangian description. They therefore pose a challenge to traditional approaches to calculating CFT observables. A fruitful strategy to obtain information about their strongly coupled dynamics, is to deform the theory away from the conformal point, where it may admit a quiver gauge theory description, and compute their partition functions, possibly decorated with Wilson loops.[1] Three-dimensional $\mathcal{N} = 4$ SCFTs also enjoy many rich properties, chief amongst them the infrared symmetry enhancement and mirror symmetry [86]. The method just outlined applies to three-dimensional $\mathcal{N} = 4$ theories as well. These, though, have the advantage that the gauge kinetic term is $Q$-exact, thus the $\mathbb{S}^3$ partition function can be evaluated directly in the SCFT.

In this paper, we focus on the matrix models and holographic backgrounds for 5d and 3d SCFTs. Remarkably, on both sides of the holographic duality, the theory is characterized by a potential function which is the solution of a 2d electrostatic problem. On the supergravity side, the two-dimensional space is the unconstrained part of the internal space and the Laplace equation emerges as a consequence of the BPS conditions. On the SCFT side, the electrostatic problem arises as a saddle point equation of the matrix model. One of the two dimensions has an immediate interpretation as the direction along the quiver. The second direction is harder to interpret from the field theory perspective, and it emerges at large $N$ from the spectrum of eigenvalues of the matrix model, which effectively becomes a continuum.

With this dictionary, the supergravity solution is reliable when the dual SCFT is given by a long quiver, whose solution was pioneered in [87] for 5d $\mathcal{N} = 1$ SCFTs and in [88, 89] for 3d $\mathcal{N} = 4$ SCFTs. It has been proven for all balanced linear quiver theories [55, 74] that the solution to the aforementioned electrostatic problem, which is defined in terms of a single charge distribution, gives rise to the supergravity background dual to these long quivers.

In this paper, we aim to back up the expectation that more general electrostatic problems provide holographic duals for a very large class of theories. This approach is indeed amenable to extend the gauge/gravity dictionary away from $\text{AdS}_{d+1}/\text{CFT}_d$, testing the correspondence at arbitrary points of RG flows. In particular, our goal is to identify a holographic manifestation of turning on real mass deformations.[2] We will argue that this operation corresponds, in the electrostatic setup, to introducing a second charge density, separated in the 2d plane from the original charge density by a finite distance proportional to the mass. Part of the results were announced in [91].

## 1.1 General idea and organisation of this paper

In this work, we aim to provide another entry in the holographic dictionary between supergravity and CFT, with the field theory side expressed via matrix models. We study the situation in which a long balanced linear quiver is deformed by one or more real mass parameters. Note that we do not simply give mass to a single matter field, but suitably choose a configuration of masses involving a large number of fields.

This setup also admits a description in terms of two or more interacting linear quivers, with an effective interaction term controlled by the mass parameter(s). These $d$-dimensional ($d = 5$ or $d = 3$ in this paper) supersymmetric field theories describe the full RG flow from a single SCFT, when the mass is switched off, to a collection of decoupled SCFTs when the mass is very large compared to the scale set by the inverse of the radius of the sphere $\mathbb{S}^d$ on which

---

[1]Complementary approaches that address directly the conformal point are geometric engineering of the SCFTs in M-theory [83, 84] or using fivebrane webs in Type IIB string theory [85].

[2]In 5d, the only supersymmetry-preserving relevant deformations are mass deformations, including the deformations leading to gauge theory phases [90].

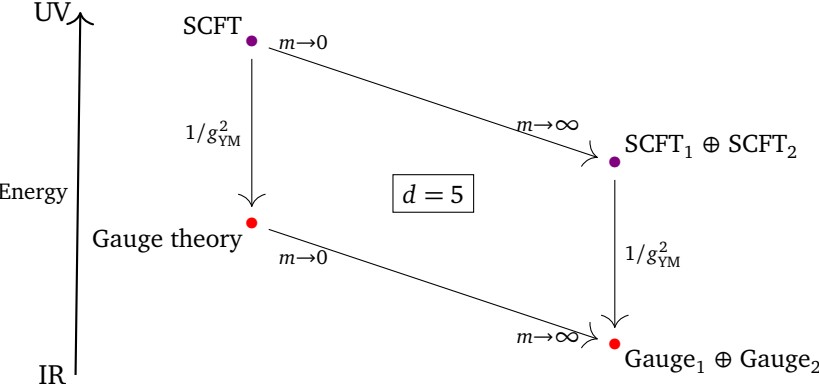

Figure 1: Schematic representation of the conformal and gauge theories involved in our construction, the corresponding energy scales, and RG flows among them in 5d.

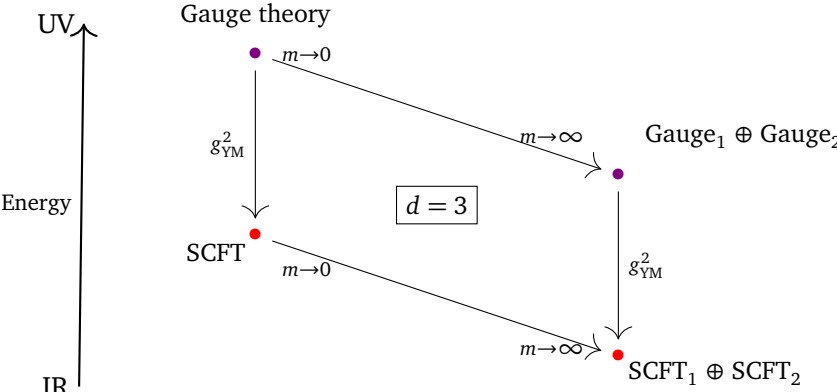

Figure 2: Schematic representation of the conformal and gauge theories involved in our construction, the corresponding energy scales, and RG flows among them in 3d.

the theory is placed. We attempted to capture this information in Figure 1 and Figure 2 for five and three spacetime dimensions, respectively.

The above mentioned conformal points are well captured by the new supergravity solutions carefully derived and explained in this work. The matching of the free energy, calculated for the field theory and in the holographic dual background, gives credit to the proposal we put forward for the holographic dual of the mass deformation.

The material is organised in two long sections, followed by conclusions and appendices. A detailed account of the contents is given at the beginning of each section.

Section 2, containing Part I of this work, describes the holographic side of the problem. A convenient formalism based on a two dimensional Laplace equation is reviewed and a new holographic situation is analysed. Calculations of the holographic central charge (i.e., the quantity holographically dual to the free energy) and expectation values of Wilson loops in arbitrary antisymmetric representations in this new setup are presented. Finally, almost as a curiosity, a simple matrix model is written that matches the holographic results of some of the systems in Part I.

In Section 3, containing Part II of this work, we initiate the field theory study of linear quivers deformed by a special choice of real mass. The theories are equivalently presented as interacting quivers. The two complementary viewpoints are explained in great detail using the matrix models derived from localisation: as the one-matrix model of a single quiver deformed by one (or more) real massive parameters, or as the multi-matrix model of two (or more)

linear quivers with an interaction term (see [92] for related manipulations). The free energy is calculated and found to match with the supergravity results of Part I, up to adding a local counterterm for background fields in $d = 5$. The matching gives support to our proposal for the holographic duals of a mass deformation in a given quiver CFT. To add more credit to this, the field theoretical calculation of Wilson loops in antisymmetric representations is performed using the associated matrix model and nicely reproduce the result of Part I. As a byproduct of our analysis, the F-theorem is shown to hold in these types of RG flows.

Conclusions and prospects for future developments are collected in Section 4. The technically intensive Appendices A and C-D complement the presentation of Sections 2 and 3, respectively, while Appendix B establishes the dictionary between our methods in $d = 5$ and the M-theory engineering of the SCFTs.

We aim at a clear and pedagogical presentation, thus detail many steps of the computations.

# 2 Part I: Supergravity

In this section, we discuss the supergravity solutions used in this paper. We summarise the backgrounds preserving eight supercharges, i.e. $\mathcal{N} = 1$ supersymmetry in five dimensions and $\mathcal{N} = 4$ in three dimensions. Supersymmetry is preserved subject to a linear PDE being satisfied. We solve the PDE and briefly comment on the quantised charges and the associated dual CFTs.

To make the section self-contained, we give a detailed account of its contents. In Subsection 2.1 we present generalities about the problem under study: the type of quivers (linear and balanced), the rank function formalism and how it is used to calculate the relevant numbers of the quiver. In Subsection 2.2, an infinite family of type IIB supergravity backgrounds dual to 5d linear quivers with eight supercharges is presented. The problem of finding these backgrounds boils down to the resolution of a two-dimensional Laplace equation with suitable boundary conditions. The charges associated with NS5 branes, D5 (colour) branes and D7 (flavour) branes are calculated. Subsection 2.3 presents an analogous discussion for the case of 3d balanced quivers with eight supercharges. The holographic problem is reduced to *the same* Laplace equation as in the 5d case. The analogies between the descriptions go along the rest of the paper. The two dimensions in which the Laplace problem is set are denoted by $(\sigma, \eta)$. Interestingly, the same Laplace problem arises in the matrix model treatment of Part II.

Subsection 2.4 discusses, for the backgrounds of the previous subsections, an observable called the holographic central charge. This coincides with the free energy of the dual CFTs. We present exact expressions for this observable, in 5d and 3d.

Subsection 2.5 poses a generalisation in the context of our Laplace-based formalism. Namely, we consider the problem that consists of two (or more) rank functions. We calculate the holographic central charge in this case, providing exact expressions. This problem motivates some of the developments in Part II of this work.

Indeed, whilst there is a clean holographic interpretation of the $\eta$-direction, the field theoretical significance of the $\sigma$-direction is more elusive. This is one of the problems addressed in the Part II of this work, building on the material in this section.

Subsection 2.6 studies the novel calculation of the vacuum expectation value of Wilson loops in antisymmetric representations of a given gauge node, for the system with two rank functions. Exact expressions are again given, and will be recovered in Part II with a calculation in field theory. This gives support to the field theoretical interpretation we present.

To close Part I of this work, Subsection 2.8 presents a very simple matrix model matching the Laplace problem of Subsections 2.2-2.3 and their holographic central charge found in



Subsection 2.4. An extensive investigation of the analogous matrix models for the new systems of Subsection 2.5 is left for the future.

## 2.1 Quivers, SCFTs, and rank functions

Let us begin by presenting the problem from the perspective of Quantum Field Theory (QFT). Consider field theories in five and three spacetime dimensions preserving eight Poincaré supercharges. We restrict our attention to the class of field theories whose dynamics will be encoded in framed linear quivers, drawn in Figure 3.

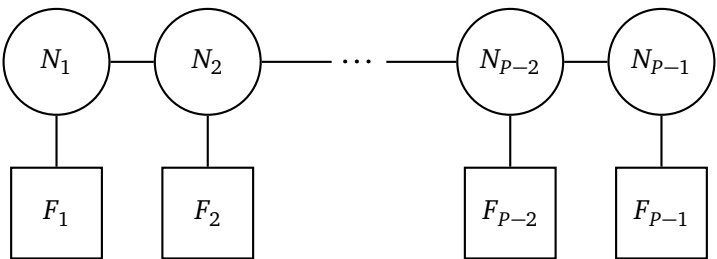

Figure 3: Linear quiver of length $P-1$ with gauge nodes $U(N_j)$ in 3d or $SU(N_j)$ in 5d, and flavour nodes $F_j$. The quiver is *balanced* if $F_j = 2N_j - N_{j-1} - N_{j+1}$.

In five spacetime dimensions, there is by now a wealth of evidence that there exist strongly coupled conformal fixed points which, when deformed by suitable relevant operators, flow to these gauge theories. The conformal fixed point is commonly referred to as UV SCFT, meaning that it sits at a higher energy scale, compared with the scale set by $1/g_{\text{YM}}^2$. Different quivers are encountered, in principle, along RG flows connected to different UV CFTs, although a given CFT typically admits several gauge theory deformations. A necessary condition for the five-dimensional gauge theories in Figure 3 to admit a UV completion, is $F_j \leq 2N_j - N_{j-1} - N_{j+1}$ [83], possibly augmented up to $F_j \leq 2N_j - N_{j-1} - N_{j+1} + 4$ at certain gauge nodes [93, 94].

Instead, if we work in three spacetime dimensions, the proposal is that each member of the infinite family of field theories described by Figure 3 which is *ugly* or *good* in the Gaiotto–Witten classification [95] flows at low energies (compared with the scale set by $1/g_{\text{YM}}^2$) to a strongly coupled conformal fixed point. Different quivers typically give rise to distinct CFTs, although there exist dualities, most notably 3d mirror symmetry [86], that relate distinct quivers flowing to the same CFT. In this work, we aim to pursue a unified description of the linear quivers in five and three dimensions. We will restrict the flavour ranks $F_j$ to

$$F_j = 2N_j - N_{j+1} - N_{j-1} \tag{2.1}$$

from now on. In other words, the quiver is *balanced* [95].

In what follows, we present the holographic description of the infinite families of conformal fixed points. We rely on previous work [55, 74], that we summarise below. An important ingredient that we take from the field theoretical description is the presence of a *rank function*. For the quiver in Figure 3, this rank function is a convex, piecewise linear and continuous function given by

$$\mathcal{R}(\eta) = \begin{cases} N_1 \eta, & 0 \leq \eta \leq 1, \\ N_j + (N_{j+1} - N_j)(\eta - j), & j \leq \eta \leq j+1, \quad j = 1, ..., P-2, \\ N_{P-1}(P - \eta), & (P-1) \leq \eta \leq P. \end{cases} \tag{2.2}$$

The rank function evaluated at integer values of the coordinate $\eta$ gives the ranks of each of the gauge groups,[3] that is, $\mathcal{R}(j) = N_j$. The second derivative of the rank function,

$$\mathcal{R}''(\eta) = \sum_{j=1}^{P-1} F_j \delta(\eta - j), \tag{2.3}$$

with $F_j$ as in (2.1), gives the ranks of the flavour groups that make the quiver balanced. The convexity of the rank function guarantees that the ranks of the flavour groups are non-negative. We emphasise that $\mathcal{R}''(\eta)$ in (2.3) really means derivative from the right, i.e.

$$\mathcal{R}''(\eta) = \lim_{\varepsilon \to 0^+} \frac{1}{\varepsilon} \left[ \partial_\eta \mathcal{R}|_{\eta - \frac{\varepsilon}{2}} - \partial_\eta \mathcal{R}|_{\eta + \frac{\varepsilon}{2}} \right].$$

The ordinary derivative together with the balancing condition (2.1) would give the opposite sign. This difference in the minus sign convention between supergravity and QFT will resurface later on.

If the function $\mathcal{R}(\eta)$ is taken to satisfy $\mathcal{R}(\eta = 0) = \mathcal{R}(\eta = P) = 0$, we refer to this as a situation *without offsets*. Otherwise, if $\mathcal{R}(\eta)$ is non-zero at either $\eta = 0$ or $\eta = P$ we refer to it as a situation *with offsets*. We discuss the case with offsets in Subsection 2.4.

Below, we summarise the supergravity backgrounds in Type IIB string theory which provide a holographic description of the SCFTs associated with these quivers in five and three dimensions. This approach to the problem serves to emphasise that the rank function encoding the kinematic data of the QFT (ranks of the gauge and flavour groups), is common to five and three dimensions. Next, we show that the rank function serves as initial condition to a boundary value problem encoding the supergravity description.

## 2.2 The Type IIB backgrounds dual to 5d SCFTs

We present an infinite family of supergravity backgrounds in Type IIB string theory preserving eight Poincaré supersymmetries with an $\text{AdS}_6$ factor. The space also contains a two-sphere parametrised by coordinates $(\theta, \varphi)$. The isometries of this manifold are in correspondence with the $SO(2,5) \times SU(2)_R$ bosonic subalgebra of the superconformal algebra of the dual $\mathcal{N} = 1$ five-dimensional SCFTs.

The full configuration consists of a metric, dilaton, $B_2$-field in the Neveu–Schwarz sector and $C_2$ and $C_0$ fields in the Ramond sector. The configuration is written in terms of a potential function $V_5(\sigma, \eta)$ that solves the linear partial differential equation

$$\partial_\sigma \left( \sigma^2 \partial_\sigma V_5 \right) + \sigma^2 \partial_\eta^2 V_5 = 0. \tag{2.4}$$

The type IIB background in string frame is [74],

$$ds_{10,st}^2 = f_1(\sigma, \eta) \left[ ds^2(\text{AdS}_6) + f_2(\sigma, \eta) ds^2(\mathbb{S}^2) + f_3(\sigma, \eta)(d\sigma^2 + d\eta^2) \right], \tag{2.5a}$$

with fields

$$e^{-2\Phi} = f_6(\sigma, \eta), \qquad B_2 = f_4(\sigma, \eta)\text{Vol}(\mathbb{S}^2), \qquad C_2 = f_5(\sigma, \eta)\text{Vol}(\mathbb{S}^2), \qquad C_0 = f_7(\sigma, \eta), \tag{2.5b}$$

---

[3]To avoid verbosity, we slightly abuse of nomenclature and refer to $N_j$ as the "rank" of both $U(N_j)$ and $SU(N_j)$.

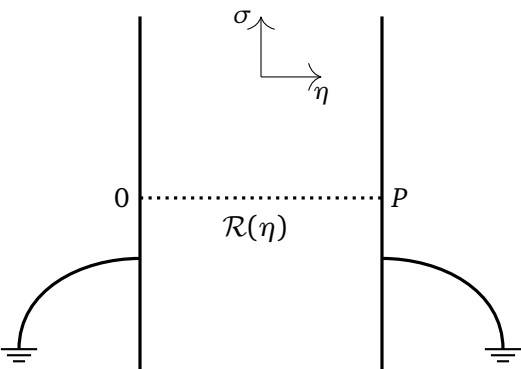

Figure 4: The electrostatic problem for $\hat{W}_5$. The two conducting planes at $\eta = 0, P$ have zero potential, while at $\sigma = 0$ we have a charge distribution equal to $\mathcal{R}(\eta)$.

and warp factors

$$f_1 = \frac{3\pi}{2}\sqrt{\sigma^2 + \frac{3\sigma\partial_\sigma V_5}{\partial_\eta^2 V_5}}, \qquad f_2 = \frac{\partial_\sigma V_5 \partial_\eta^2 V_5}{3\Lambda}, \qquad f_3 = \frac{\partial_\eta^2 V_5}{3\sigma\partial_\sigma V_5},$$

$$f_4 = \frac{\pi}{2}\left(\eta - \frac{(\sigma\partial_\sigma V_5)(\partial_\sigma\partial_\eta V_5)}{\Lambda}\right),$$

$$f_5 = \frac{\pi}{2}\left(V_5 - \frac{\sigma\partial_\sigma V_5}{\Lambda}(\partial_\eta V_5(\partial_\sigma\partial_\eta V_5) - 3(\partial_\eta^2 V_5)(\partial_\sigma V_5))\right),$$

$$f_6 = 12\frac{\sigma^2\partial_\sigma V_5\partial_\eta^2 V_5}{(3\partial_\sigma V_5 + \sigma\partial_\eta^2 V_5)^2}\Lambda, \qquad f_7 = 2\left(\partial_\eta V_5 + \frac{(3\sigma\partial_\sigma V_5)(\partial_\sigma\partial_\eta V_5)}{3\partial_\sigma V_5 + \sigma\partial_\eta^2 V_5}\right),$$

$$\Lambda = \sigma(\partial_\sigma\partial_\eta V_5)^2 + (\partial_\sigma V_5 - \sigma\partial_\sigma^2 V_5)\partial_\eta^2 V_5.$$

(2.5c)

The paper [74] proves that this infinite family of backgrounds is in exact correspondence with the solutions discussed in [67–70, 96, 97].

Let us briefly summarise the study of [74] for the PDE, with boundary conditions leading to a proper interpretation of the solutions, with quantised Page charges and avoiding badly-singular behaviours.

### 2.2.1 Resolution of the PDE and quantisation of charges

We make the change $V_5(\sigma, \eta) = \frac{\hat{W}_5(\sigma, \eta)}{\sigma}$, which implies that the PDE in (2.4) reads like a Laplace equation in flat space,

$$\partial_\sigma^2 \hat{W}_5 + \partial_\eta^2 \hat{W}_5 = 0. \tag{2.6}$$

We choose the variable $\eta$ to be bounded in the interval $[0, P]$ and $\sigma$ to range over the real axis $-\infty < \sigma < \infty$. We impose the boundary conditions,

$$\hat{W}_5(\sigma \to \pm\infty, \eta) = 0, \quad \hat{W}_5(\sigma, \eta = 0) = \hat{W}_5(\sigma, \eta = P) = 0,$$

$$\lim_{\epsilon \to 0}\left(\partial_\sigma \hat{W}_5(\sigma = +\epsilon, \eta) - \partial_\sigma \hat{W}_5(\sigma = -\epsilon, \eta)\right) = -\mathcal{R}(\eta). \tag{2.7}$$

These can be interpreted as the boundary conditions for the electrostatic problem of two conducting planes (at zero electrostatic potential) as depicted in Figure 4. The conducting planes extend over the $\sigma$-direction and are placed at $\eta = 0$ and $\eta = P$. We also have a charge density $\mathcal{R}(\eta)$ at $\sigma = 0$, extended along $0 \le \eta \le P$, as indicated by the difference of the normal components of the electric field in (2.7).

The solution is found by separating variables, see [74] for the details. It is convenient to Fourier expand the function $\mathcal{R}(\eta)$ as

$$\mathcal{R}(\eta) = \sum_{k=1}^{\infty} R_k \sin\left(\frac{k\pi}{P}\eta\right), \quad R_k = \frac{2}{P} \int_0^P \mathcal{R}(\eta) \sin\left(\frac{k\pi\eta}{P}\right) d\eta. \tag{2.8}$$

Following [74], the solution reads,

$$\hat{W}_5(\sigma, \eta) = \sum_{k=1}^{\infty} a_k \sin\left(\frac{k\pi}{P}\eta\right) e^{-\frac{k\pi}{P}|\sigma|}, \qquad a_k = \frac{P}{2\pi k} R_k. \tag{2.9}$$

The potentials in (2.9) solve the equation (2.6) subject to the conditions (2.7). Another way to encode the boundary condition at $\sigma = 0$ is through a source term that transforms the Laplace equation into a Poisson equation. Indeed, the potential $\hat{W}_5(\sigma, \eta)$ in (2.9) satisfies

$$\partial_\eta^2 \hat{W}_5 + \partial_\sigma^2 \hat{W}_5 = -\mathcal{R}(\eta)\delta(\sigma), \qquad \sigma \in \mathbb{R}, \quad \eta \in [0, P]. \tag{2.10}$$

Imposing the quantisation of the conserved Page charges of the background (2.5), the authors of [74] found that the function $\mathcal{R}(\eta)$ must be a convex piecewise linear function. If the rank function has no offsets $\mathcal{R}(0) = \mathcal{R}(P) = 0$, it takes the form (2.2). In this case, the authors of [74] computed the values of the quantised brane charges in each interval $\eta \in [j, j+1]$

$$Q_{D7}[j, j+1] = \mathcal{R}''(j) = (2N_j - N_{j+1} - N_{j-1}), \tag{2.11a}$$
$$Q_{D5}[j, j+1] = \mathcal{R}(\eta) - \mathcal{R}'(\eta)(\eta - j) = N_j \tag{2.11b}$$

(the second summand in (2.11b) can be thought of as coming from a large gauge transformation of the $B_2$ field) and in the whole system,[4]

$$Q_{NS5}^{total} = P, \qquad Q_{D7}^{total} = (N_1 + N_{P-1}) = \int_0^P \mathcal{R}''(\eta) d\eta, \qquad Q_{D5}^{total} = \int_0^P \mathcal{R}(\eta) d\eta. \tag{2.11c}$$

For the generic rank function $\mathcal{R}(\eta)$ quoted in (2.2), the supergravity background is proposed to be dual to the strongly coupled fixed point in the UV of the quiver in Figure 3, with $F_i = 2N_i - N_{i+1} - N_{i-1}$. In other words, the quiver is balanced.

## 2.3 The Type IIB backgrounds dual to 3d SCFTs

We now discuss the Type IIB backgrounds dual to three dimensional SCFTs preserving eight supercharges. The formalism is very much analogous to the five dimensional one, hence we will be more sketchy. All the details can be found in [55].

We are after solutions dual to 3d $\mathcal{N} = 4$ superconformal field theories. To match the $\mathcal{N} = 4$ superconformal symmetry of the field theory, the background must have isometries $SO(2,3) \times SU(2)_C \times SU(2)_H$ and preserve eight Poincaré supercharges. Hence, our geometries must contain an AdS$_4$ factor and a pair of two-spheres $\mathbb{S}_1^2(\theta_1, \varphi_1)$ and $\mathbb{S}_2^2(\theta_2, \varphi_2)$. There are two extra directions labelled by $(\sigma, \eta)$. The requirement of an isometry group that contains $SO(2,3) \times SU(2)_C \times SU(2)_H$ allows for warp factors that depend only on $(\sigma, \eta)$. The Ramond and Neveu–Schwarz fields must also respect these isometries.

---

[4]In this paper we define the $'$-derivative $\mathcal{R}'$ as a derivative from the right — see below (2.3). This flips the sign in front of the rank functions in the differential equation, compared to [74]. The integration domains in the Page charges are chosen accordingly.

The preservation of eight Poincaré supersymmetries implies that the generic type IIB background can be cast in terms of a function $V_3(\sigma, \eta)$. In string frame the solution reads [55],

$$ds^2_{10,st} = f_1(\sigma, \eta)\Big[ds^2(\text{AdS}_4) + f_2(\sigma, \eta)ds^2(\mathbb{S}^2_1) + f_3(\sigma, \eta)ds^2(\mathbb{S}^2_2) + f_4(\sigma, \eta)(d\sigma^2 + d\eta^2)\Big],$$
(2.12a)

with

$$e^{-2\Phi} = f_5(\sigma, \eta), \qquad B_2 = f_6(\sigma, \eta)\text{Vol}(\mathbb{S}^2_1),$$
$$C_2 = f_7(\sigma, \eta)\text{Vol}(\mathbb{S}^2_2), \qquad \tilde{C}_4 = f_8(\sigma, \eta)\text{Vol}(\text{AdS}_4),$$
(2.12b)

and

$$f_1 = \frac{\pi}{2}\sqrt{\frac{\sigma^3\partial^2_{\eta\sigma}V_3}{\partial_\sigma(\sigma\partial_\eta V_3)}}, \qquad f_2 = -\frac{\partial_\eta V_3\partial_\sigma(\sigma\partial_\eta V_3)}{\sigma\Lambda}, \qquad f_3 = \frac{\partial_\sigma(\sigma\partial_\eta V_3)}{\sigma\partial^2_{\eta\sigma}V_3},$$

$$f_4 = -\frac{\partial_\sigma(\sigma\partial_\eta V_3)}{\sigma^2\partial_\eta V_3}, \qquad f_5 = -16\frac{\Lambda\partial_\eta V_3}{\partial^2_{\eta\sigma}V_3}, \qquad f_6 = \frac{\pi}{2}\left(\eta - \frac{\sigma\partial_\eta V_3\partial^2_\eta V_3}{\Lambda}\right),$$

$$f_7 = -2\pi\left(\partial_\sigma(\sigma V_3) - \frac{\sigma\partial_\eta V_3\partial^2_\eta V_3}{\partial^2_{\eta\sigma}V_3}\right), \qquad f_8 = -\pi^2\sigma^2\left(3\partial_\sigma V_3 + \frac{\sigma\partial_\eta V_3\partial^2_\eta V_3}{\partial_\sigma(\sigma\partial_\eta V_3)}\right),$$

$$\Lambda = \partial_\eta V_3\partial^2_{\eta\sigma}V_3 + \sigma\left((\partial^2_{\eta\sigma}V_3)^2 + (\partial^2_\eta V_3)^2\right).$$
(2.12c)

As usual, the fluxes are defined from the potentials as

$$F_1 = 0, \quad H_3 = dB_2, \quad F_3 = dC_2, \quad F_5 = d\tilde{C}_4 + *d\tilde{C}_4.$$

The configuration in (2.12) is solution to the Type IIB equations of motion, if the function $V_3(\sigma, \eta)$ satisfies

$$\partial_\sigma\left(\sigma^2\partial_\sigma V_3\right) + \sigma^2\partial^2_\eta V_3 = 0.$$
(2.13)

As proven in [55], this infinite family of solutions is equivalent to the backgrounds described by D'Hoker, Estes and Gutperle in [49].

### 2.3.1 Resolution of the PDE and quantisation of charges

Following [55], define $V_3(\sigma, \eta) = \frac{\hat{V}_3(\sigma, \eta)}{\sigma}$ and $\hat{V}_3(\sigma, \eta) = \partial_\eta\hat{W}_3(\sigma, \eta)$. Consider the coordinates to range in $0 \leq \eta \leq P$ and $-\infty < \sigma < \infty$. The differential equation (2.13) must be supplemented by boundary and initial conditions. In terms of $\hat{W}_3(\sigma, \eta)$ the problem reads

$$\partial^2_\sigma\hat{W}_3(\sigma, \eta) + \partial^2_\eta\hat{W}_3(\sigma, \eta) = 0, \qquad \text{(almost everywhere)}$$
$$\hat{W}_3(\sigma \to \pm\infty, \eta) = 0, \quad \hat{W}_3(\sigma, \eta = 0) = \hat{W}_3(\sigma, \eta = P) = 0,$$
$$\lim_{\epsilon \to 0}\left(\partial_\sigma\hat{W}_3(\sigma = +\epsilon, \eta) - \partial_\sigma\hat{W}_3(\sigma = -\epsilon, \eta)\right) = -\mathcal{R}(\eta).$$
(2.14)

In the current conventions, we have ensured that $\hat{W}_3$ satisfies the same electrostatic problem as $\hat{W}_5$ in the five dimensional case above — see eqs. (2.6)-(2.7). The function $\mathcal{R}(\eta)$ is an input.

Using a Fourier decomposition for the rank function $\mathcal{R}(\eta)$ as in the five-dimensional case, cf. (2.8), the solution to the problem in (2.14) is

$$\hat{W}_3(\sigma, \eta) = \sum_{k=1}^{\infty} a_k \sin\left(\frac{k\pi\eta}{P}\right)e^{-\frac{k\pi|\sigma|}{P}}, \qquad a_k = \frac{P}{2k\pi}R_k.$$
(2.15)

The reader can check that

$$\partial_\eta^2 \hat{W}_3 + \partial_\sigma^2 \hat{W}_3 = -\mathcal{R}(\eta)\delta(\sigma).$$

The study of the quantised charges for NS5 branes enforces the size of the interval $P$ to be an integer, consistently with the boundary conditions in (2.14), exactly as it occurs in the five dimensional system. Also in analogy with the 5d case, to have quantised numbers of D3 and D5 branes, the rank function must be a piecewise linear and continuous function of the exact same form as in the five dimensional case — see (2.2).

Let us again consider the piecewise linear, convex and continuous function (2.2) without offsets. For such a rank function, the number of D3 (colour) branes and D5 (flavour) branes in the interval $[j, j+1]$ and the total number of branes are given by

$$Q_{\mathrm{D3}}[j, j+1] = N_j, \quad Q_{\mathrm{D5}}[j, j+1] = 2N_j - N_{j+1} - N_{j-1}, \tag{2.16}$$

$$Q_{\mathrm{D3}}^{\mathrm{total}} = \int_0^P \mathcal{R}(\eta)d\eta, \quad Q_{\mathrm{D5}}^{\mathrm{total}} = \mathcal{R}'(0) - \mathcal{R}'(P), \quad Q_{\mathrm{NS5}}^{\mathrm{total}} = P.$$

As we emphasised, the supergravity backgrounds we have obtained, holographically dual to five and three dimensional systems, have different dynamics, but at their core are described by a function $\hat{W}_5(\sigma, \eta)$ or $\hat{W}_3(\sigma, \eta)$ solving the same Poisson equation. This translates into analogies between the quivers in different odd dimensions.

We start by calculating the holographic central charge. To add to the already existent bibliography, we study this observable in the case in which the rank function has offsets.

## 2.4 Holographic central charge

In this subsection, we will compute the holographic central charges in the $\mathrm{AdS}_6 \times \mathbb{S}^2$ and $\mathrm{AdS}_4 \times \mathbb{S}^2 \times \mathbb{S}^2$ cases. This can be done in a slightly more general context in which we allow the rank function to have offsets. That means we consider a rank function

$$\mathcal{R}(\eta) = N_j + (N_{j+1} - N_j)(\eta - j), \quad j \le \eta \le j+1, \quad j = 0, ...., P-1, \tag{2.17}$$

which satisfies $\mathcal{R}(j) = N_j$ for integers $j$, including $j = 0, P$. In the case $N_0 = N_P = 0$, the rank function (2.17) reduces to the one quoted in eq. (2.2). This more general rank function was already considered in the 3d case in [55]. We shall now present it in the 5d case. We calculate the Fourier coefficient of the rank function (2.17):

$$R_k = \frac{2}{k\pi}\left(N_0 + (-1)^{k+1}N_P\right) + \frac{2P}{\pi^2 k^2}\sum_{j=1}^{P-1} F_j \sin\left(\frac{k\pi j}{P}\right). \tag{2.18}$$

They are found by first computing the integrals

$$R_k^{(\ell)} := \frac{2}{P}\int_\ell^{\ell+1} \mathcal{R}(\eta)\sin\left(\frac{k\pi\eta}{P}\right)d\eta = \frac{2}{\pi k}\left[N_\ell \cos\left(\frac{\pi k \ell}{P}\right) - N_{\ell+1}\cos\left(\frac{\pi k(\ell+1)}{P}\right)\right]$$
$$+ \frac{2P}{\pi^2 k^2}(N_\ell - N_{\ell+1})\left[\sin\left(\frac{\pi k\ell}{P}\right) - \sin\left(\frac{\pi k(\ell+1)}{P}\right)\right],$$

and then summing the results $R_k = \sum_{\ell=0}^{P-1} R_k^{(\ell)}$. The terms with cosines vanish in the "bulk" but survive on the boundaries $\ell = 0, P-1$, whereas the terms with sines give the terms with flavour groups.

Notice that in the case $N_0 = N_P = 0$ in (2.18), the ranks for boundary flavour groups were given by $F_1 = 2N_1 - N_2$ and $F_{P-1} = 2N_{P-1} - N_{P-2}$. In (2.18), we have extended the definition $F_j = 2N_j - N_{j-1} - N_{j+1}$ to include $j = 0, P$, which now incorporates $N_0$ and $N_P$ into $F_1$ and $F_{P-1}$

respectively. Hence, $N_0$ and $N_P$ give additional flavour symmetry to make the quiver balanced. This effect is clear from the Hanany–Witten brane configurations: stretching D$d$ colour branes to infinity on the left or on the right gives additional flavour symmetry, exactly as if we let them end on D($d + 2$) flavour branes. We postpone a thorough analysis of the supergravity background needed to accurately interpret the situation.

In the AdS$_6$ geometry, the holographic central charge of the theory can be computed as the volume of an internal manifold [74] which works out as

$$c_{hol} = \frac{2}{3\pi^5} \int_{-\infty}^{\infty} d\sigma \int_0^P d\eta \, \sigma^3 (\partial_\sigma V_5)(\partial_\eta^2 V_5) \tag{2.19a}$$

$$= \frac{2}{3\pi^5} \int_{-\infty}^{\infty} d\sigma \int_0^P d\eta \, \partial_\eta^2 \hat{W}_5 \left( \sigma \partial_\sigma \hat{W}_5 - \hat{W}_5 \right), \tag{2.19b}$$

where $V_5 = \frac{\hat{W}_5}{\sigma}$ and $\hat{W}_5$ is given by (2.9). The integration over $\eta$ can be performed using the orthogonality of sine functions, which collapses the double summation into a single summation. After the $\sigma$-integral is performed, one finds

$$c_{hol} = \frac{1}{2\pi^4} \sum_{k=1}^{\infty} k a_k^2 = \frac{P^2}{8\pi^6} \sum_{k=1}^{\infty} \frac{1}{k} R_k^2. \tag{2.20}$$

Plugging in the Fourier coefficients (2.17) gives

$$\begin{aligned}
c_{hol} = {} & \frac{P^2}{4\pi^8} (2N_0^2 + 2N_P^2 + 3N_0 N_P)\zeta(3) \\
& + \frac{P^3}{\pi^9} \sum_{l=1}^{P-1} F_l [N_0 \text{Im}(\text{Li}_4(e^{\frac{il\pi}{P}})) + N_P \text{Im}(\text{Li}_4(e^{\frac{i(P-l)\pi}{P}}))] \\
& - \frac{P^4}{4\pi^{10}} \sum_{l=1}^{P-1} \sum_{k=1}^{P-1} F_l F_k \, \text{Re}\left( \text{Li}_5\left(e^{i\frac{\pi(k+l)}{P}}\right) - \text{Li}_5\left(e^{i\frac{\pi(k-l)}{P}}\right) \right).
\end{aligned} \tag{2.21}$$

The first, second and third line of this expression stem from $\frac{1}{k^3}$, $\frac{1}{k^4}$ and $\frac{1}{k^5}$ contributions respectively. This result precisely matches the free energy of the 5d SCFT as was obtained by using matrix model techniques [87, Eq.(3.17)] (see also Subsection 3.3).

In the AdS$_4$ geometry dual to the three-dimensional case, a similar formula holds [55]

$$c_{hol} = -\frac{1}{2} \int_{-\infty}^{\infty} d\sigma \int_0^P d\eta \, \sigma^2 (\partial_\eta V_3) \partial_\sigma (\sigma \partial_\eta V_3) \tag{2.22a}$$

$$= -\frac{1}{2} \int_{-\infty}^{\infty} d\sigma \int_0^P d\eta \left( \sigma \partial_\eta^2 \hat{W}_3 \right) \left( \partial_\eta^2 \partial_\sigma \hat{W}_3 \right). \tag{2.22b}$$

In this case we use $V_3 = \frac{\hat{V}_3}{\sigma}$ and $\hat{V}_3 = \partial_\eta \hat{W}_3$ before employing the solution (2.15). We follow the same approach and use the orthogonality of sine functions to arrive at

$$c_{hol} = \frac{\pi}{32} \sum_{k=1}^{\infty} k R_k^2. \tag{2.23}$$

To compute this series with offset we need to regularise the sums

$$\sum_{k=1}^{\infty} \frac{1}{k} \overset{\text{reg.}}{=} \gamma_E, \qquad \sum_{k=1}^{\infty} \frac{(-1)^k}{k} \overset{\text{reg.}}{=} -\log 2, \tag{2.24}$$

with $\gamma_E$ the Euler–Mascheroni constant and the symbol $\overset{\text{reg.}}{=}$ meaning $\zeta$-function regularization. The result is

$$
\begin{aligned}
c_{hol} =& \frac{1}{8\pi}(N_0^2 + N_P^2)\gamma_E + \frac{1}{8\pi}N_0 N_P \log(2) \\
&+ \frac{P}{4\pi^2}\sum_{l=1}^{P-1}F_l[N_0\text{Im}(\text{Li}_2(e^{\frac{il\pi}{P}})) + N_P\text{Im}(\text{Li}_2(e^{\frac{i(P-l)\pi}{P}}))] \\
&- \frac{P^2}{16\pi^3}\sum_{l=1}^{P-1}\sum_{k=1}^{P-1}F_l F_k \,\text{Re}\left(\text{Li}_3\left(e^{i\frac{\pi(k+l)}{P}}\right) - \text{Li}_3\left(e^{i\frac{\pi(k-l)}{P}}\right)\right).
\end{aligned}
\tag{2.25}
$$

This formula is, broadly speaking, analogous to (2.21) with a polylogarithm of 2 degrees lower. Also in this case we find agreement with the computation of the free energy of the 3d SCFT [89, Eq.(36)].[5]

## 2.5 The case of two rank functions: two interacting linear quivers

In this section, we consider a natural extension of the Poisson problem (2.10). We will study this problem in both backgrounds dual to the 5d and the 3d quivers. As we mentioned, the differential equations are identical, hence we can study the solution simultaneously. We introduce a second rank function $\mathcal{R}_2$ that places a new charge density at a line $\sigma = \sigma_0$. The Poisson equation of such a problem is

$$
\partial_\eta^2 \hat{W}(\sigma, \eta) + \partial_\sigma^2 \hat{W}(\sigma, \eta) + \mathcal{R}_2(\eta)\delta(\sigma_0 - \sigma) + \mathcal{R}_1(\eta)\delta(\sigma) = 0,
\tag{2.26}
$$

with boundary conditions

$$
\hat{W}(\sigma \to \pm\infty) = \hat{W}(\eta = 0) = \hat{W}(\eta = P) = 0.
\tag{2.27}
$$

The electrostatic problem is depicted in Figure 5. The PDE is still linear, which allows us a simple way to write a solution. We notice that if $\hat{W}[\mathcal{R}_1](\sigma, \eta)$ is a solution to the electrostatic problem with a source $\mathcal{R}_1$ at $\sigma = 0$, this already solves half of the source term in eq. (2.26). We need to add to this a solution which takes care of the source $\mathcal{R}_2$ at $\sigma = \sigma_0$. This is easily done by simply shifting a solution to the problem with a source $\mathcal{R}_2$ at $\sigma = 0$. The resulting solution is

$$
\hat{W}[\mathcal{R}_1, \mathcal{R}_2](\sigma, \eta) = \sum_{k=1}^{\infty}\frac{P}{2k\pi}\sin\left(\frac{k\pi\eta}{P}\right)\left[R_{1,k}e^{-\frac{k\pi}{P}|\sigma|} + R_{2,k}e^{-\frac{k\pi}{P}|\sigma - \sigma_0|}\right],
\tag{2.28}
$$

with the Fourier coefficients of the two rank functions given by

$$
\mathcal{R}_\alpha(\eta) = \sum_{k=1}^{\infty}R_{\alpha,k}\sin\left(\frac{k\pi\eta}{P}\right), \qquad \alpha = 1, 2.
\tag{2.29}
$$

In the AdS$_6$ geometry, we compute the integral (2.19) with the potential in (2.28) and substitute $V_5(\sigma, \eta) = \frac{\hat{W}[\mathcal{R}_1, \mathcal{R}_2](\sigma, \eta)}{\sigma}$. We find

$$
c_{hol}|_{d=5}[\mathcal{R}_1, \mathcal{R}_2] = \frac{P^2}{8\pi^6}\sum_{k=1}^{\infty}\frac{1}{k}\left[R_{1,k}^2 + R_{2,k}^2 + 2R_{1,k}R_{2,k}e^{-\frac{k\pi\sigma_0}{P}}\left(1 + \frac{k\pi\sigma_0}{P}\right)\right].
\tag{2.30}
$$

---

[5]The offset terms are not given in [89], but are easily computed as in the 5d case [87], and follow more generally from Subsection 3.3.

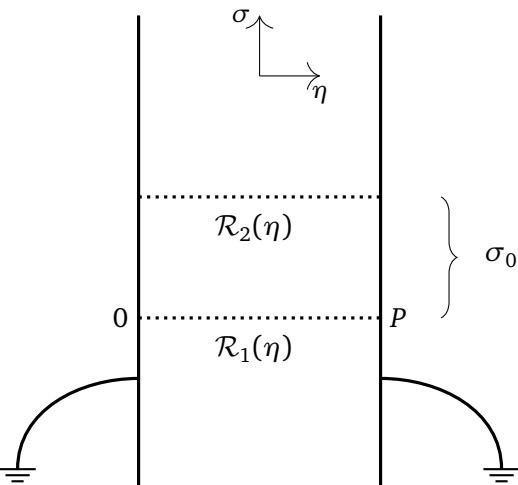

Figure 5: The electrostatic problem for $\hat{W}$ in the two rank functions setup. We have a charge distribution equal to $\mathcal{R}_1(\eta)$ at $\sigma = 0$ and a second charge distribution $\mathcal{R}_2(\eta)$ at $\sigma = \sigma_0$.

In the AdS$_4$ case, the analogous integral to find the holographic central charge is (2.22). Using the potential (2.28) and substituting $V_3(\sigma, \eta) = \frac{\partial_\eta \hat{W}[\mathcal{R}_1, \mathcal{R}_2](\sigma, \eta)}{\sigma}$, we find

$$c_{hol}|_{d=3}[\mathcal{R}_1, \mathcal{R}_2] = \frac{\pi}{32} \sum_{k=1}^{\infty} k \left[ R_{1,k}^2 + R_{2,k}^2 + 2 R_{1,k} R_{2,k} e^{-\frac{k\pi\sigma_0}{P}} \left( 1 + \frac{k\pi\sigma_0}{P} \right) \right]. \qquad (2.31)$$

We observe that, in both cases, the holographic central charge consists of terms $R_{\alpha,k}^2$, which are familiar from the single rank function setups. In addition to that, we have an "interaction" term that is controlled by $R_{1,k} R_{2,k}$. Explicitly,

$$c_{hol}[\mathcal{R}_1, \mathcal{R}_2] - (c_{hol}[\mathcal{R}_1] + c_{hol}[\mathcal{R}_2]) \propto 2 \sum_{k=1}^{\infty} k^{4-d} R_{1,k} R_{2,k} e^{-\frac{k\pi\sigma_0}{P}} \left( 1 + \frac{k\pi\sigma_0}{P} \right),$$

where the symbol $\propto$ here means up to a $d$-dependent but otherwise universal numerical coefficient.

It is interesting to study the limiting cases as we move the charge densities very close to, or very far away from each other. The proper parameter for these limiting procedures is $\frac{\sigma_0}{P}$. If we take $\frac{|\sigma_0|}{P} \gg 1$, the exponential term suppresses the interaction and we are left over with two copies of the holographic central charges from the single rank function setup:

$$c_{hol}[\mathcal{R}_1, \mathcal{R}_2] = c_{hol}[\mathcal{R}_1] + c_{hol}[\mathcal{R}_2], \qquad \frac{\sigma_0}{P} \to \infty. \qquad (2.32)$$

Instead, when the charge densities get very close together, $0 \le \frac{|\sigma_0|}{P} \ll 1$, we have

$$c_{hol}[\mathcal{R}_1, \mathcal{R}_2] = c_{hol}[\mathcal{R}_1 + \mathcal{R}_2], \qquad \frac{\sigma_0}{P} \to 0. \qquad (2.33)$$

In the spirit of a holographic version of the F-theorem [98, 99], we argue in Appendix D that both in $d = 5$ and $d = 3$, $c_{hol}$ satisfies

$$\lim_{\sigma_0 \to 0} c_{hol} > \lim_{\sigma_0 \to \infty} c_{hol},$$

$$\partial_{\sigma_0} c_{hol} \le 0, \qquad \forall 0 \le \sigma_0 < \infty.$$

It is then tempting to think of this as a relevant deformation of a QFT with a rank function $\mathcal{R}_1 + \mathcal{R}_2$ that in a large $\frac{\sigma_0}{P}$ limit factorises the theory into non-interacting theories with rank functions $\mathcal{R}_1$ and $\mathcal{R}_2$. In Section 3 we motivate this claim further, in the context of matrix model computations.

## 2.6 Wilson loops in supergravity

The holographic expectation value for a Wilson loop in the rank-$\ell$ antisymmetric representation $\wedge^\ell$ of $SU(N_j)$ (respectively $U(N_j)$) in long quivers of the type depicted in Figure 3, were investigated in [100–102]. In the usual Hanany–Witten brane setups associated with the 5d (resp. 3d) quiver theories of our interest, insertion of a Wilson loop in the $\wedge^\ell$ representation corresponds to a configuration of $\ell$ fundamental strings stretched between a probe D3 (resp. D5′) brane and the stack of $N_j$ colour D5 (resp. D3) branes, in a manner consistent with the s-rule. The Wilson loop vev is calculated by evaluating the on-shell action of the probe D3 (resp. D5′) brane on which the $\ell$ fundamental strings end.[6]

The result can be written in terms of the potentials that appear in the electrostatic problem

$$\ln\langle\mathcal{W}_{\wedge^\ell}\rangle = (d-2)\pi\left(\sigma\partial_\sigma\hat{W}(\sigma,\eta) - \hat{W}(\sigma,\eta)\right)_{(\sigma_*,\eta_*)}, \tag{2.35}$$

where $\eta_*$ denotes the position in the interval $\eta \in [0,P]$ at which the loop operator is inserted, and $\sigma_*$ fixes the representation via the relation

$$\ell = \left[\partial_\sigma\hat{W}(\sigma,\eta)\right]_{(\sigma_*,\eta_*)}, \tag{2.36}$$

which is the expression for the number of fundamental strings that end on the probe mentioned above. Namely, on the left-hand side of (2.36) we write the number $\ell$ of fundamental strings by definition, and we equate it with the evaluation of the same number in terms of the supergravity function $\hat{W}(\sigma,\eta)$. The right-hand side of (2.36) is obtained from a computation very similar to (but slightly more involved than) the one that led (2.11), see in particular [102, Eq.s(3.2)-(3.3)].

Recall that $\hat{W}$ depends linearly on the Fourier coefficients $R_k$, which grow linearly in $N$, and the same is valid for $\ell$, so that the latter equality is understood by dividing both sides by $N$ and equating the finite results. This parallels the field theory analysis of Subsection 3.7. The large $N$ limit at fixed $\ell$ is recovered as a particular case of this procedure.

In order to evaluate (2.35), we use (2.28) to obtain

$$\sigma\partial_\sigma\hat{W}(\sigma,\eta) = -\frac{1}{2}\sum_{k=1}^\infty \sin\left(\frac{k\pi\eta}{P}\right)\left[|\sigma|R_{1,k}e^{-\frac{k\pi}{P}|\sigma|} + \sigma\,\mathrm{sign}(\sigma-\sigma_0)R_{2,k}e^{-\frac{k\pi}{P}|\sigma-\sigma_0|}\right]. \tag{2.37}$$

Next we use (2.18), setting $N_0 = N_P = 0$, inside (2.37) and (2.28), and substitute into (2.35) to obtain

$$\frac{\ln\langle\mathcal{W}_{\wedge^\ell}\rangle}{d-2} = \frac{P^2}{2\pi^2}\sum_{j=1}^{P-1}F_{1,j}\mathrm{Re}\Big[\mathrm{Li}_3\left(e^{-\frac{\pi}{P}[|\sigma_*|+i(\eta_*+j)]}\right) - \mathrm{Li}_3\left(e^{-\frac{\pi}{P}[|\sigma_*|+i(\eta_*-j)]}\right)$$

$$+\frac{\pi}{P}|\sigma_*|\left(\mathrm{Li}_2\left(e^{-\frac{\pi}{P}[|\sigma_*|+i(\eta_*+j)]}\right) - \mathrm{Li}_2\left(e^{-\frac{\pi}{P}[|\sigma_*|+i(\eta_*-j)]}\right)\right)\Big]$$

$$+\frac{P^2}{2\pi^2}\sum_{j=1}^{P-1}F_{2,j}\mathrm{Re}\Big[\mathrm{Li}_3\left(e^{-\frac{\pi}{P}[|\sigma_*-\sigma_0|+i(\eta_*+j)]}\right) - \mathrm{Li}_3\left(e^{-\frac{\pi}{P}[|\sigma_*-\sigma_0|+i(\eta_*-j)]}\right)$$

$$+\frac{\pi}{P}\sigma_*\mathrm{sign}(\sigma_*-\sigma_0)\left(\mathrm{Li}_2\left(e^{-\frac{\pi}{P}[|\sigma_*-\sigma_0|+i(\eta_*+j)]}\right) - \mathrm{Li}_2\left(e^{-\frac{\pi}{P}[|\sigma_*-\sigma_0|+i(\eta_*-j)]}\right)\right)\Big]. \tag{2.38}$$

---

[6]Taking the probe brane limit has a nice correspondence with the matrix model calculation, where the insertion of the Wilson loop does not modify the saddle point equations.

We note here that the Wilson loop expectation values are factorised; each of the above two summands depends explicitly on one of the two rank functions in our setup. This is unlike the expression for the holographic central charges (2.30)-(2.31), which also include an interaction term between the Fourier coefficients of the two rank functions. The reason is that, in the electrostatic picture, the holographic central charge accounts for the interaction between the two density of charges, hence the quadratic dependence on $\mathcal{R}(\eta)$. On the contrary, a Wilson loop is a probe that interacts with the two densities of charges individually, thus bearing a linear dependence on $\mathcal{R}(\eta)$.

To study the situation in which the two rank functions are far away from each other, we must send $\frac{\sigma_0}{P} \to \infty$. There are two ways of doing so: either directly, in which case we are left with the Wilson loop in the supergravity background dictated by $\mathcal{R}_1(\eta)$, or by keeping $\frac{\tilde{\sigma}}{P} = \frac{\sigma - \sigma_0}{P}$ fixed. This second limit zooms in near $\mathcal{R}_2(\eta)$, and we are left with the Wilson loop in the corresponding supergravity background.

The cautious reader might question the representation carried by the Wilson loops after the factorisation. It is slightly premature to answer this question at this stage, seeing as we have not yet identified the field theory interpretation of the double rank function setup. We will clarify this issue in Subsection 3.7, when we reproduce (2.38) from the gauge theory side.

## 2.7 On the reliability of the supergravity backgrounds

### 2.7.1 Validity of the AdS ansatz

In the case of *one* rank function, the backgrounds in (2.5) and (2.12) contain certain singularities. These singularities are localised around the sources (flavour branes), becoming sharply localised in the limit of long quivers with large ranks. In this case, the backgrounds are reliable in most of the spacetime. Closer to the sources, the description should be supplemented by the inclusion of the open string sector (realised on the sources). In other words, the string-sigma model must include the presence of boundaries.

The case of two (or more) rank functions separated by a distance $\sigma_0$ is qualitatively different. It can be noticed (see Figure 6 for an example in AdS$_4$) that the dilaton is singular in the region $0 < |\sigma| < |\sigma_0|$, indicating that the background is not reliable there. This situation ameliorates, when either $|\sigma_0| \to 0$ or $|\sigma_0| \to \infty$. The singularities are a consequence of the function $\Lambda(\sigma, \eta)$ defined in equations (2.5c) and (2.12c) vanishing. Hence, it is not only the dilaton that is problematic, but the whole background.

This behaviour can be interpreted in the following way: spaces like the ones studied here, containing an AdS$_d$ factor, are not reliable when the parameter $\sigma_0$ is switched on. The case of very large $\frac{\sigma_0}{P}$ on the other hand, improves the behaviour, keeping only the singularities localised on the sources for $\frac{\sigma_0}{P} \to \infty$.

We speculate that this is the reflection that the isometries of AdS$_{d+1}$ are too stringent for systems with finite $\frac{\sigma_0}{P}$. In other words, the conformal symmetry of the dual field theory must be broken in this case. In Section 3, we discuss the field theoretical interpretation of the deformation by a finite $\sigma_0$, finding that an RG flow must take place, recovering the conformal symmetry when $\frac{\sigma_0}{P} \to \infty$.

Some natural questions arise: what is the value of calculations done with backgrounds for which $\frac{\sigma_0}{P}$ is finite? Are the holographic central charge or the Wilson loops vev calculations meaningful?

It has been observed that for holographic systems with singular behaviour, certain observables become independent of the singularities. Indeed, the warp factors and fields in the background conspire to produce a finite (and physically meaningful) quantity. This is one of the hallmarks of the "good singularities", using the definitions developed in [103, 104]. This is the case for the holographic central charge computed in Subsection 2.4 and the Wilson loop

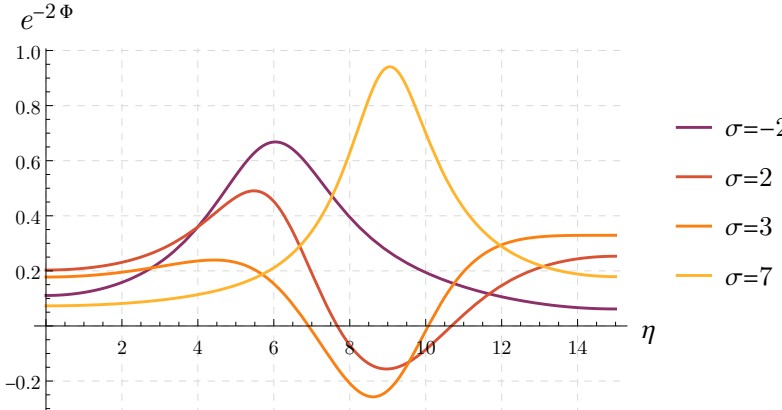

Figure 6: Plot of $e^{-2\Phi}$ at various values of $\sigma$ in the AdS$_4$ background. Here we set $P = 15$ and $\sigma_0 = 6$. The two quivers are given by two triangular rank functions (see [55, Sec.4.1]), $\mathcal{R}_1$ has a flavour group at $\eta = 6$ while $\mathcal{R}_2$ at $\eta = 9$. The presence of flavour branes is signaled by one bump below ($\sigma = -2$) and one bump above ($\sigma = 7$) the locations of the two rank functions in the $\sigma$-axis. In the region between them, the dilaton is not well-defined since its exponential can be negative.

vev of Subsection 2.5, whose results are reproduced by field theory computations in Section 3. We emphasise that there should exist probes of the backgrounds with two rank functions that will display the singular behaviour in the case of finite $\frac{\sigma_0}{P}$.

Of course, it would be very interesting to find new backgrounds that interpolate between the conformal field theories for very large and vanishing values of $\frac{\sigma_0}{P}$. This is a difficult problem that we leave for future investigation.

### 2.7.2 Factorisation

Our claim is that the solution factorises in the holographic dual of two decoupled SCFTs at $|\sigma_0| \to \infty$. In this limit, the supergravity background will be described by a single metric (2.5), which, depending on the precise way the limit is taken, is either the metric specified by the rank function $\mathcal{R}_1$ or the one specified by the rank function $\mathcal{R}_2$. Namely, sending $\sigma_0 \to \infty$ directly, one is left with the AdS$_{d+1}$ metric fixed by the rank function $\mathcal{R}_1$, as in [55,74]. However, first shifting $\tilde{\sigma} = \sigma - \sigma_0$ and then sending $\sigma_0 \to \infty$, the AdS$_{d+1}$ metric that remains is determined by the rank function $\mathcal{R}_2$, located at $\tilde{\sigma} = 0$. In this way, the AdS$_{d+1}$ background, which we is reliable as $\sigma_0 \to \infty$, detects the two decoupled SCFTs. This behaviour indeed matches holographically with the choice of vacuum of the SCFT, namely whether the singularity at the tip of the Coulomb branch of the gauge theory given by the rank function $\mathcal{R}_1$ or the one given by $\mathcal{R}_2$.

In conclusion, the electrostatic setup, which requires an AdS$_{d+1}$ ansatz, cannot be an accurate description at finite $\sigma_0$, which breaks conformal symmetry. It is nonetheless a valuable device that captures several key features of the RG flow of the dual field theory and, thanks to the cancellations explained above, it reliably computes physical observables such as $c_{hol}$.

## 2.8 Matrix models for electrostatic problems

In this subsection, we set up a simple electrostatic problem whose large $P$ limit directly matches the holographic central charge of the geometry containing the AdS$_6$ factor. The derivation is based on constructing a unitary matrix model for the electrostatic problem, as explained

in [105, Sec.2.7], without any knowledge of AdS/CFT. This model is distinct from, and enormously simpler than, the matrix model obtained from localization on $\mathbb{S}^d$.

We begin with the simplest setting: consider the strip

$$\{(\eta, \sigma) \, : 0 \le \eta \le P, \, -\infty < \sigma < \infty\} = [0, P] \times \mathbf{R},$$

and place $P-1$ unit charges on the interior points of the integral lattice along the compact $\eta$-direction at $\sigma = 0$. That is to say, the $j^{\text{th}}$ particle is placed at position $(\eta_j, 0)$, with $\eta_j$ describing fluctuations centered at $\eta_j = j$, $\forall j = 1, \dots, P-1$. The electrostatic potential $W_{\text{el}}$ is subject to the boundary conditions

$$W_{\text{el}}(\eta, \sigma \to \infty) = 0 = W_{\text{el}}(\eta, \sigma \to -\infty),$$
$$W_{\text{el}}(0, \sigma) = 0 = W_{\text{el}}(P, \sigma). \tag{2.39}$$

Using that the Coulomb potential is logarithmic in 1d, the Boltzmann factor $e^{-\beta E}$ due to the interaction among the $P-1$ charges confined at $\sigma = 0$ is determined by

$$E(\eta_1, \dots, \eta_{P-1}) = - \sum_{1 \le j \ne \ell \le P-1} \ln \left( \left| e^{i2\pi \eta_j / P} - e^{i2\pi \eta_\ell / P} \right| \frac{P}{2\pi} \right). \tag{2.40}$$

Furthermore, we introduce a background charge distribution, to compensate the repulsion among the charges. Without this neutralising background charge density, on an unbounded domain the particles would repel each other to $\eta_j \to \pm\infty$. The presence of boundaries prevents this scenario, but the equilibrium would simply be given by the charges being equi-spaced. To allow more interesting configurations, we add the background charge. The resulting potential energy of a charge at position $(\eta, 0)$ is generically written in Fourier expansions as

$$V_{\text{el}}(z) = \sum_{k=1}^{\infty} a_k \sin \left( \frac{k\pi \eta}{P} \right).$$

The canonical partition function for this system is obtained as a mild generalization of [105, Sec.2.7]. In units such that $\beta = 2$, it reads

$$\exp \left( \mathcal{F}_{\text{el}}^{\text{Dir}} \right) = \frac{1}{(P-1)!} \int_{[0,P]^{P-1}} \prod_{1 \le j < \ell \le P-1} \left| \left( e^{i2\pi \eta_j / P} - e^{i2\pi \eta_\ell / P} \right) \right|^2 \prod_{j=1}^{P-1} e^{-2V(z_j)} \left( \frac{2\pi}{P} \right) \mathrm{d}\eta_j. \tag{2.41}$$

We have expressed the partition function directly in terms of the free energy $\mathcal{F}_{\text{el}}^{\text{Dir}}$, and dropped an $\eta$-independent overall factor, that can be cancelled by a suitable choice of background charge. The integral (2.41) is a unitary matrix model, equivalent to integration over the unitary group with Haar measure,

$$\exp \left( \mathcal{F}_{\text{el}}^{\text{Dir}} \right) = \int_{U(P-1)} \mathrm{d}U \, \exp \left\{ \mathrm{Tr} V (U + U^\dagger) \right\}. \tag{2.42}$$

The large $P$ limit of (2.42) is calculated by Szegő's theorem [106]:

$$\lim_{P \to \infty} \mathcal{F}_{\text{el}}^{\text{Dir}} = \sum_{k=1}^{\infty} k a_k^2,$$

which appears to be proportional to the holographic central charge (2.20).

The relation with $c_{hol}$ is only formal at this stage, because we have not specified the Fourier coefficients $a_k$. First, let us notice that, in the large $P$ limit, the $P-1$ discrete charges are

replaced by an inhomogeneous density of charge $\mathcal{R}_{\text{el}}(\eta)$, which grows linearly in $P$. The electrostatic potential solves a continuous version of the previous problem, in which the large number of discrete charges is replaced by the continuous density. This leads to the Poisson equation

$$\partial_\eta^2 W_{\text{el}} + \partial_\eta^2 W_{\text{el}} + \frac{2\pi}{P}\mathcal{R}_{\text{el}}(\eta)\delta(\sigma) = 0\,,$$

which is precisely (2.10), upon identification

$$\mathcal{R}(\eta) = \frac{2\pi}{P}\mathcal{R}_{\text{el}}(\eta)\,. \tag{2.43}$$

Then, we determine the coefficients $a_k$ by requiring that, at equilibrium, the electrostatic force on the position $(\eta, 0)$ due to the density of charges $\mathcal{R}_{\text{el}}$ is compensated by the interaction with the background. The electrostatic potential on the position $\eta$ generated by the distribution of charges at all positions $\tilde\eta \neq \eta$, in this continuous version, is:

$$\Phi(\eta) = \frac{1}{P}\int_0^P \mathrm{d}\tilde\eta \ln\sin\left(\frac{\pi}{P}|\eta - \tilde\eta|\right)\mathcal{R}_{\text{el}}(\tilde\eta)\,.$$

The corresponding Coulomb force is $\partial_\eta\Phi(\eta)$. Equating it with the force $\partial_\eta V_{\text{el}}(\eta)$ results in:

$$\frac{1}{P}\int_0^P \mathrm{d}\tilde\eta \cot\left(\frac{\pi}{P}(\eta - \tilde\eta)\right)\mathcal{R}_{\text{el}}(\tilde\eta) = -2\sum_{k=1}^\infty k a_k \cos\left(\frac{k\pi\eta}{P}\right)\,.$$

Expanding the left-hand side of the equilibrium condition in Fourier modes, with the identification (2.43), integrating over $\tilde\eta$ and imposing the equality, we get

$$k a_k = \frac{P}{2\pi}R_k\,.$$

Thus, the coefficients $a_k$ we seek are exactly the ones defined in (2.9). This completes the match with the single rank function supergravity setup, concretely (2.10) and (2.20):

$$\lim_{P\to\infty}\mathcal{F}_{\text{el}}^{\text{Dir}} = \sum_{k=1}^\infty k a_k^2 = 2\pi^4 c_{hol}\,.$$

This purely electrostatic matrix model predicts the correct holographic central charge in the case of a single rank function. One may set up the analogous problem in the two rank functions case. Two sets of charges are placed along the $\eta$-direction at $\sigma = 0$ and $\sigma = \sigma_0$. We call their positions $\eta_j, \hat\eta_j$, respectively. The Boltzmann factor $e^{-\beta E}$ is given by

$$E = E(\eta_1, \ldots, \eta_{P-1}) + E(\hat\eta_1, \ldots, \hat\eta_{P-1}) + E_{\text{int}}(\eta_1, \ldots, \eta_{P-1}, \hat\eta_1, \ldots, \hat\eta_{P-1}; \sigma_0)\,,$$

where the first two pieces are as in (2.40) and [105, Eq.(2.72)]

$$E_{\text{int}} = -\sum_{j=1}^{P-1}\sum_{\ell=1}^{P-1}\ln\left[\frac{P}{\pi}\left|\sin\left(\frac{\pi}{P}(\eta_j - \hat\eta_\ell + i\sigma_0)\right)\right|\right]\,.$$

At $\sigma_0 \to 0$ the two pairs of eigenvalues coalesce and we get a larger matrix, containing the two sets of charges. At $\sigma_0 \to \infty$, the interaction term is exponentially suppressed and the matrix model breaks into two copies of (2.41). The two limiting pictures agree with the supergravity problem.

The purely electrostatic problem and the associated unitary matrix model allow for several generalizations. An immediate one is to have distinct number of charges at $\sigma = 0$ and $\sigma = \sigma_0$. An exhaustive exploration of the implications of these matrix models in holography is a mathematical problem that we leave for future work.

# 3 Part II: Matrix models

In this section, we discuss the large $N$ limit of the free energy on the QFT side. The field theories we are interested in are encoded in framed $A_{P-1}$ quivers, that is, linear quivers with $P-1$ gauge and flavour nodes, as in Figure 3. The supergravity solution is reliable in the regime $P \gg 1$, that we will consider below. Moreover, we focus on quivers that are balanced.

The long quiver limit of linear quivers was first solved by Uhlemann for 5d SCFTs [87] and later extended to 3d in [88,89]. Here we reformulate these results in a unified notation and rederive them in a formalism consistent with Section 2. More importantly, we extend the derivation away from the superconformal fixed point, allowing mass deformations. Along the way, we refine the notion of strong coupling limit for five-dimensional long quivers, in a way that is invariant under fibre/base duality in M-theory.

On a technical level, the main results are the reformulation of [87], uniform in $d$, and its extension along RG flows triggered by real masses. We compute the free energies, identify the physical meaning of the results and discuss related subtleties. The outcome is used to substantiate the AdS/CFT correspondence and to identify the deformation holographically dual to the problem studied in Subsection 2.5.

The outline of this section is as follows. Subsection 3.1 contains an elementary presentation of the RG flows we consider, from the perspective of the Coulomb branch of the QFT. In Subsection 3.2 we detail the general procedure, following and extending [87–89]. The free energies in 5d and 3d are derived in Subsection 3.3 in this more general setting. The case of two rank functions is discussed explicitly in Subsection 3.4, where we propose an alternative derivation. In Subsection 3.5 we discuss the implications of the RG flow on the QFT side, define an *effective free energy* and show the agreement between the supergravity calculations and the matrix model. The effective free energies satisfy the F-theorem, as detailed in Subsection 3.6. Wilson loops in antisymmetric representations are studied in Subsection 3.7.

The main results of this section are: the deformed free energies of long quivers in 5d and 3d (3.21) and (3.22), respectively; the explicit match with the holographic central charges in (3.32) and (3.33); the match of the Wilson loop vev in (3.41); the discussion on the effective free energy in Subsection 3.5.3; and finally the F-theorem in Subsection 3.6.

## 3.1 Coulomb branches and mass deformations

In this section, we study mass deformations of five- and three-dimensional linear quivers by evaluating their sphere partition functions in the long quiver and large $N$ limit. We begin with an overview of the mass deformations and how they resolve the singularity of the Coulomb branch of the SCFT. For definiteness, and only during the current subsection, we frame the discussion in the 5d setting. It can be readily exported to the 3d $\mathcal{N} = 4$ case, as we spell out in Subsection 3.1.3.

The goal of the current subsection is to recast known facts in the framework of the present work. This proves useful to shed light on the setup discussed holographically in Section 2, but the experienced reader may safely skip this subsection and refer to Figure 8 for a schematic summary.

### 3.1.1 Mass parameter space

The moduli space of massive deformations, called the parameter space of the theory, is spanned by the vacuum expectation value (vev) of the real scalars in the background vector multiplets for the global symmetry.

The fundamental hypermultiplets acquire a real mass by coupling them to a background vector multiplet for the flavour symmetry. A non-vanishing vev for the background vector

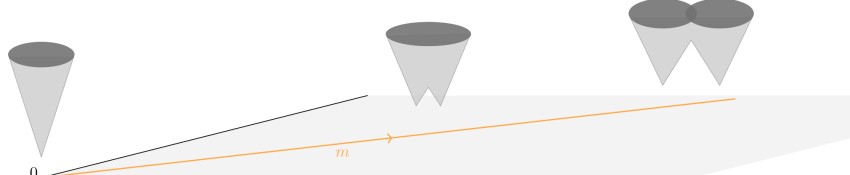

Figure 7: Schematic view of the Coulomb branch fibered over the mass parameter space. Moving along the RG flow triggered by the selected mass deformation (orange) resolves the Coulomb branch singularity. It breaks in two disjoint cones at the limiting point $m = \infty$.

multiplet thus activates an RG flow. The flows holographically dual to the ones proposed in Subsection 2.5 are controlled by a *single* mass parameter $m$: at every node $j = 1, \ldots, P-1$, the scalar in the background vector multiplet for the flavour symmetry algebra $\mathfrak{u}(F_j)$ is given a vev

$$\mathrm{diag}(\underbrace{0, \ldots, 0}_{F_{1,j}}, \underbrace{m, \ldots, m}_{F_{2,j}}).$$

This is usually referred to as leaving $F_{1,j}$ of the hypermultiplets massless, and giving a mass $m$ to the remaining $F_{2,j}$ hypermultiplets. As we will show in Subsections 3.2-3.3, the argument extends to a larger number of mass parameters without any conceptual difference.

In both 5d and 3d, the global symmetry of the SCFT contains the torus

$$\mathbb{T}_G = \prod_{j=1}^{P-1} \left[ U(1)_f^{F_j} \times U(1)_I \right] / U(1)_{\mathrm{diag}}^{(5-d)/2},$$

where the $U(1)_f$ factors come from the maximal torus of the flavour symmetry, and $U(1)_I$ acting on the instantons in 5d, or on the monopoles in 3d. There is always one such $U(1)_I$ per gauge node. Finally, $U(1)_{\mathrm{diag}}$ acts diagonally on the flavour symmetry, and the quotient by $U(1)_{\mathrm{diag}}^{(5-d)/2}$ conveniently encodes that the center of the flavour symmetry is gauged in 3d, where the gauge groups are unitary, but not in 5d.

The vevs of the scalars in the vector multiplets for the $U(1)_f$ are the standard mass parameters, while the vevs of the scalar in the $U(1)_I$ are, respectively, the inverse gauge couplings $1/g_{\mathrm{YM},j}^2$ in 5d and the FI parameters in 3d. For later reference, notice that, as opposed to four-dimensional Yang–Mills theory, the gauge couplings of 5d $\mathcal{N} = 1$ theories pertain to the mass parameter space, and shall be dealt with accordingly in our analysis, as opposed to most of the existing large $N$ studies.

### 3.1.2 Coulomb branch geometry

Consider a five-dimensional linear quiver as in Figure 3, denoted by $\mathcal{Q}$. The zero-mode of the real scalar $\vec{\sigma} \in \mathbf{R}^{|\vec{N}|}$ in the $|\vec{N}|$-dimensional Cartan subalgebra of the gauge group parametrises the Coulomb branch of $\mathcal{Q}$, CB[$\mathcal{Q}$] for short. CB[$\mathcal{Q}$] is fibered over the mass parameter space of the theory [107]. At the origin of the parameter space, CB[$\mathcal{Q}$] has a conical singularity at its origin, $\vec{\sigma} = 0$. Moving on the parameter space, the singularity is fully resolved at generic points, and partially resolved at positive-codimensional loci in the parameter space.

As explained above, we are mainly interested in the scenario in which one activates vevs for the background scalars controlled by a unique parameter $m$. The hypermultiplets are grouped in two families, becoming massless at separate points on CB[$\mathcal{Q}$], whose distance increases along the RG flow triggered by $m$. We are thus resolving the singularity of CB[$\mathcal{Q}$] in a minimal

and controlled way, by moving along a one-real dimensional locus on the parameter space. This idea is illustrated in Figure 7.

CB[$\mathcal{Q}$] is expected to break in two smaller singularities at the end of the RG flow, which we will identify with the Coulomb branches of two quivers $\mathcal{Q}_1, \mathcal{Q}_2$ with smaller gauge and flavour ranks. Both CB[$\mathcal{Q}_1$] and CB[$\mathcal{Q}_2$] have a singularity at their origin and another one at the point at infinity. The hypermultiplets that would become massless at infinity are effectively decoupled from the gauge theory.

Besides the geometry, another clear way to see the appearance of decoupled modes is from the realization of the quiver theories using brane webs [85]. Our choice of mass deformation corresponds to splitting the colour branes (D5 in 5d, D3 in 3d) into two separate stacks. The strings that connect branes within the $\alpha^{\text{th}}$ stack give rise to the modes in $\mathcal{Q}_\alpha$, $\alpha = 1, 2$, whilst the strings stretched between branes belonging to different stacks give rise to heavy modes, that eventually decouple.

Before moving on to the actual computation, we emphasise that the RG flow activated by the mass deformation we introduce is independent of the RG flows that deform the 5d SCFTs onto their gauge theory phases, as represented in Figure 8. In the language of Subsection 3.1.1, the two deformations correspond to moving along different directions in the mass parameter space, and can be dealt with independently.

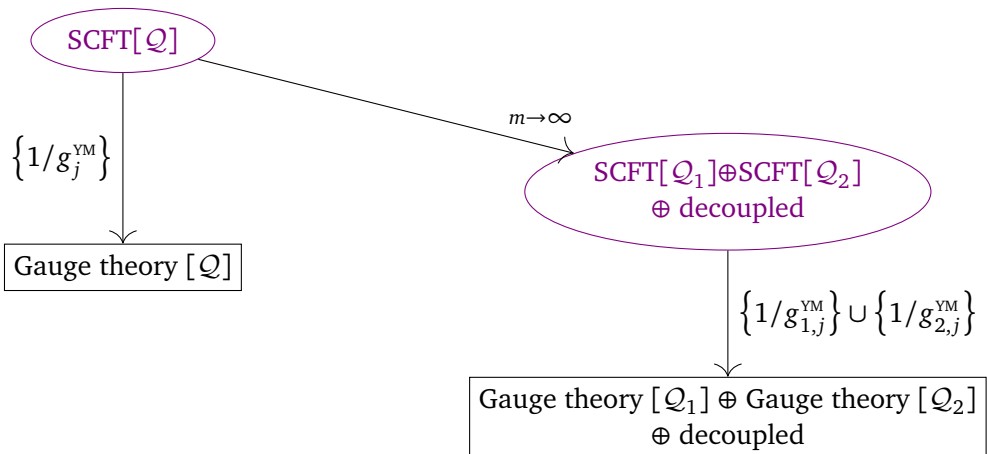

Figure 8: Illustration of the massive deformations and associated RG flows in our 5d setup. The diagonal arrow labelled by $m$ is the flow that matches with supergravity. Vertical arrows indicate RG flows from a SCFT to a gauge theory, the relevant operator triggering it being the supersymmetric Yang–Mills kinetic term.

### 3.1.3 Mass deformation in 3d $\mathcal{N} = 4$ theories

The discussion above, albeit phrased suitably for 5d $\mathcal{N} = 1$ quiver theories, applies to 3d $\mathcal{N} = 4$ quivers $\mathcal{Q}$ as well, with minor adjustments. In the three-dimensional setup the Coulomb branch CB[$\mathcal{Q}$] is a hyperKähler variety. The scalars in the vector multiplets, both gauge and background, form a triplet under the $SU(2)$ which rotates the complex structure of CB[$\mathcal{Q}$]. Fixing a given complex structure corresponds to select an $\mathcal{N} = 2$ subalgebra of the full $\mathcal{N} = 4$ supersymmetry and, under such choice, each $\mathcal{N} = 4$ vector multiplet decomposes into an $\mathcal{N} = 2$ vector multiplet, carrying a real scalar, and an $\mathcal{N} = 2$ chiral multiplet in the adjoint representation, carrying a complex scalar. Fixing an $\mathcal{N} = 2$ subalgebra is necessary for localisation, and only the real scalar enters the localisation expression, whilst the complex scalars are set to zero at the localisation locus. Consistently, the superpotential vanishes on the localization locus. See [108] for a detailed review.

The mass deformation we consider corresponds to couple a collection of hypermultiplets to an $\mathcal{N} = 4$ background vector multiplet in a way that preserves the full supersymmetry, exactly as they are coupled to the dynamical gauge vector multiplet. Upon localisation, only the real component of the triplet of background scalars enters the localised expressions, and we refer to it as the real mass. Of course, our choice would be rotated by a $SU(2)$ transformation on CB[$\mathcal{Q}$], but it would do so in exactly the same way as the dynamical scalars. In other words, the precise analogue of the 5d procedure is to activate a full $\mathcal{N} = 4$ background vector multiplet with its $SU(2)$-triplet of scalar fields, and we give a specifically chosen vev to it.

Finally let us mention that in $d = 3$ we may consider a deformation given by a triplet of scalars in a twisted vector multiplet, corresponding to an FI term. Such deformation would partially resolve the Higgs branch and lift the Coulomb branch, obstructing the mass deformation we are interested in. While in 3d $\mathcal{N} = 4$ there should exists one such deformation that mirrors the mass deformation we consider, we do not pursue it here, and focus solely on mass deformations, that geometrically resolve the Coulomb branch.

### 3.1.4 Summary of the mass deformation prescription

We have introduced the mass parameter space, reviewed the realisation of mass deformations via coupling to background vector multiplets, and the effect on the Coulomb branch geometry. We are now ready to distil our prescription into a pragmatic recipe.

(1) At every node $j$, select an arbitrary splitting of the number of flavours $F_j = \sum_{\alpha=1}^{\mathscr{F}} F_{\alpha,j}$.[7]

(2) At every $j$, couple the hypermultiplets to a background vector multiplet that gives a mass $m_\alpha$ (independent of $j$) to the collection $F_{\alpha,j}$.

(3) Impose a splitting of the gauge ranks $N_j = \sum_{\alpha=1}^{\mathscr{F}} N_{\alpha,j}$ by requiring the balancing condition for the reduced collection $\left\{ N_{\alpha,j}, F_{\alpha,j} \right\}_{j=1,\ldots,P-1}$ at every $\alpha$.

This prescription has a neat interpretation in Type IIB string theory, which we comment on in Appendix C.1.

Technically, we do not get to choose the Higgsing pattern, rather it is a consequence of how the Coulomb branch has been resolved. The third point of the recipe should be

(3′) Determine the splitting of the gauge ranks $N_j = \sum_{\alpha=1}^{\mathscr{F}} N_{\alpha,j}$ consistent with the Coulomb branch that survives at the end of the flow.

We claim that (3) and (3′) are equivalent. For the sake of completeness, we elaborate further on this facet below, in Subsection 3.4.3 and in Appendix C.2.3.

## 3.2 Long quivers and their large $N$ limit

### 3.2.1 Notation

We denote the gauge ranks $N_1, \ldots, N_{P-1}$ and the flavour ranks $F_1, \ldots, F_{P-1}$. We use a label $j = 1, \ldots, P-1$ running over the nodes of the quiver, and labels $a, b = 1, \ldots, N_j$ running over the indices in the Cartan of the gauge algebra $\mathfrak{u}(N_j)$ at the node $j$.

Following [55, 74] and Section 2, we define a rank function $\mathcal{R}(\eta)$, for a continuous index $0 \le \eta \le P$, such that $\mathcal{R}(\eta = j) = N_j$. Its Fourier expansion is

$$\mathcal{R}(\eta) = \sum_{k=1}^{\infty} R_k \sin\left( \frac{k\pi\eta}{P} \right).$$

---

[7] The number $\mathscr{F}$ must be independent of $j$, but this results in no loss of generality, because one may always split $F_j$ into a smaller number of summands by setting some of the $F_{\alpha,j}$ to zero.

In the following, we fix $N \in \mathbb{N}$. We will assume $N \gg 1$ later on. It is convenient to introduce a continuous label $0 \le z \le 1$,

$$z = \frac{\eta}{P},$$

and write

$$v(z) := \frac{1}{N} \mathcal{R}(Pz),$$

so that $N_j = N v(z)$. The Fourier expansion of the scaled rank function $v(z)$ is

$$v(z) = \sum_{k=1}^{\infty} \mathfrak{a}_k \sin(k\pi z),$$

with coefficients $\mathfrak{a}_k$ related to $a_k$ defined in (2.15) through $\mathfrak{a}_k = \frac{R_k}{N} = \frac{2\pi k}{NP} a_k$.

To set up the Veneziano limit, the flavour ranks must scale with $N$, so that $F_j = N\zeta_j$, with $\zeta_j$ held fixed in the large $N$ limit. The flavour ranks are collected into a flavour rank function $\zeta(z)$.

We work in units in which the radius of $\mathbb{S}^d$ is set to one. The sphere free energy $\mathcal{F}$ is defined as [99]

$$\mathcal{F} = (-1)^{\frac{d-1}{2}} \ln \mathcal{Z}_{\mathbb{S}^d} . \tag{3.1}$$

### 3.2.2 Mass deformations

Throughout this section, the fundamental hypermultiplets are grouped into $\mathscr{F}$ families of equal mass, such that $F_{\alpha,j}$ hypermultiplets have the same mass $m_\alpha$, $\forall \alpha = 1, \ldots, \mathscr{F}$. This corresponds to turning on the same mass for a fraction $\zeta_\alpha(z)$ of $\zeta(z)$, at every $0 \le z \le 1$. Clearly, for consistency we have

$$\sum_{\alpha=1}^{\mathscr{F}} \zeta_\alpha(z) = \zeta(z).$$

Let us stress that there are $\mathscr{F}$ different mass scales

$$\{ m_\alpha \ : \ \alpha = 1, \ldots, \mathscr{F} \},$$

with a variable number of hypermultiplets with each mass at each node. Hypermultiplets in the $\alpha^{\text{th}}$ family at every node have the same mass $m_\alpha$. Previously, we demanded that $\zeta_\alpha(z)$ remains finite in the large $N$ limit. In particular, $\mathscr{F}$ is held fixed. The large $\mathscr{F}$ limit might be interesting for future investigation.

Recall that we are dealing with balanced quivers, and that the balancing condition reads (see [87, 109] for the proof)[8]

$$\begin{aligned}
\zeta(z) &= -\frac{1}{P^2} \frac{\partial^2}{\partial z^2} v(z) \\
&= \frac{\pi^2}{P^2} \sum_{k=1}^{\infty} \mathfrak{a}_k k^2 \sin(k\pi z) .
\end{aligned}$$

Because the function $\zeta(z)$ splits into the sum of $\zeta_\alpha(z)$, we define the corresponding Fourier coefficients $\mathfrak{a}_{\alpha,k}$, such that

$$\zeta_\alpha(z) = \frac{\pi^2}{P^2} \sum_{k=1}^{\infty} \mathfrak{a}_{\alpha,k} k^2 \sin(k\pi z) , \qquad \forall \alpha = 1, \ldots, \mathscr{F} . \tag{3.2}$$

---

[8]Recall the discussion below (2.3) about sign conventions in the $\partial_z^2 v$ versus $\mathcal{R}''$ derivative.

The corresponding "reduced" rank functions are

$$\nu_\alpha(z) = \sum_{k=1}^\infty \mathfrak{a}_{\alpha,k} \sin(k\pi z), \qquad \forall \alpha = 1, \dots, \mathscr{F}.\tag{3.3}$$

### 3.2.3 Sphere partition functions

To unify the 5d and 3d notation, we work on the round $d$-dimensional sphere $\mathbb{S}^d$, $d \in \{3, 5\}$. The $d$-sphere partition function of a linear quiver with 8 supercharges is [110–114]

$$\mathcal{Z}_{\mathbb{S}^d} = \int_{\mathbb{R}^{|\vec{N}|}} d\vec{\sigma}\; Z_{\text{cl}}(\vec{\sigma}) Z_{\text{1-loop}}^{\text{vec}}(\vec{\sigma}) Z_{\text{1-loop}}^{\text{hyp}}(\vec{\sigma}, \vec{m}) Z_{\text{non-pert}}.\tag{3.4}$$

Here $\vec{\sigma}$ is the zero-mode of the real scalar in the vector multiplet, at the localization locus. We have used the shorthand notation $|\vec{N}| := \sum_{j=1}^P N_j$ and[9]

$$d\vec{\sigma} := \prod_{j=1}^{P-1} \prod_{a=1}^{N_j} d\sigma_{a,j} \times \left[\prod_{j=1}^{P-1} \delta\left(\sum_{a=1}^{N_j} \sigma_{a,j}\right)\right]^{(d-3)/2}$$

is the Lebesgue measure on the Cartan of the gauge algebra, isomorphic to $\mathbb{R}^{|\vec{N}|}$ in 3d and to $\mathbb{R}^{|\vec{N}|-P+1}$ in 5d. The term in square bracket enforces that the gauge nodes in 5d are $SU(N_j)$, while they are $U(N_j)$ in 3d. The integrand of (3.4) comprises the following terms:

- The classical contribution $Z_{\text{cl}} = e^{-S_{\text{YM}}-S_{\text{CS}}-S_{\text{FI}}}$, includes the Yang–Mills (YM), Chern–Simons (CS) and Fayet–Iliopoulos (FI) terms, evaluated at the localization locus. Hereafter we set the Chern–Simons couplings to zero, while

$$S_{\text{YM}} = \begin{cases} 0, & d = 3, \\ \sum_{j=1}^{P-1} \sum_{a=1}^{N_j} \frac{(2\pi)^3}{g_{\text{YM},j}^2} \sigma_{a,j}^2, & d = 5. \end{cases}$$

In flat space, the inverse 5d Yang–Mills coupling has the dimension of a mass parameter. For later convenience, we define the corresponding mass parameter at the node $j$ as

$$m_j^{\text{YM}} = \frac{(2\pi)^3}{g_{\text{YM},j}^2},$$

and collect all of them in the function $m^{\text{YM}}(z) = m_{zP}^{\text{YM}}$.

In $d = 3$ the gauge groups $U(N_j)$ have non-trivial fundamental group, and admit FI couplings. This possibility is precluded in $d = 5$, for the gauge group being simply connected. Therefore, we may contemplate the additional term

$$S_{\text{FI}} = \delta_{d,3} \sum_{j=1}^{P-1} \sum_{a=1}^{N_j} 2\pi i \xi_j \sigma_{a,j},$$

with $\xi_j \in \mathbf{R}$ the FI parameter associated to the $j^{\text{th}}$ gauge node. For simplicity, we set $S_{\text{FI}}$ to zero, but we claim that having a finite FI parameter would not change our results. We substantiate this claim at the end of Subsection 3.3.3.

---

[9]For ease of notation, we combine the Vandermonde factors coming from diagonalising the scalar zero-mode $\vec{\sigma}$ with the contribution of the vector multiplet.

- The one-loop contribution of the vector multiplet, $Z_{\text{1-loop}}^{\text{vec}}$. It takes the form

$$Z_{\text{1-loop}}^{\text{vec}}(\vec{\sigma}) = \prod_{j=1}^{P-1} \prod_{1 \le a \ne b \le N_j} \exp\left\{ -\mathsf{v}(\sigma_{a,j} - \sigma_{b,j}) \right\},$$

where $\mathsf{v}$ is a real-valued function of a single variable

$$\mathsf{v}(s) = \begin{cases} -\ln\left(2\sinh(\pi s)\right), & d = 3, \\ -\ln\left(2\sinh(\pi s)\right) - \frac{1}{2}f(is), & d = 5. \end{cases}$$

The function $f(is)$ in 5d is a real even function of $s$ [111, 112]:

$$f(is) \overset{\text{reg.}}{=} \sum_{n=1}^{\infty} n^2 \ln\left(1 + \frac{s^2}{n^2}\right)$$

$$\Rightarrow f(is) = \frac{\pi}{3}s^3 - s^2 \ln\left(1 - e^{-2\pi s}\right) + \frac{s}{\pi}\text{Li}_2\left(e^{-2\pi s}\right) + \frac{1}{2\pi^2}\text{Li}_3\left(e^{-2\pi s}\right) - \frac{\zeta(3)}{2\pi^2},$$

the first line meaning that $f(is)$ is defined to be the $\zeta$-function regularization of the divergent right-hand side, explicitly given in the second line.

- The one-loop contribution of the hypermultiplets. For the quivers we consider, the hypermultiplets are of two types: fundamental and bifundamental. The integrand factorises accordingly:

$$Z_{\text{1-loop}}^{\text{hyp}}(\vec{\sigma}, \vec{m}) = Z^{\text{fund}}(\vec{\sigma}, m) Z^{\text{bif}}(\vec{\sigma}).$$

With the assumption above on the masses, we have

$$Z^{\text{fund}}(\vec{\sigma}, \vec{m}) = \prod_{j=1}^{P-1} \prod_{a=1}^{N_j} \prod_{\alpha=1}^{\mathscr{F}} \exp\left\{ -F_{\alpha,j}\mathsf{h}(\sigma_{a,j} - m_\alpha) \right\},$$

$$Z^{\text{bif}}(\vec{\sigma}) = \prod_{j=1}^{P-2} \prod_{a=1}^{N_j} \prod_{b=1}^{N_{j+1}} \exp\left\{ -\mathsf{h}(\sigma_{a,j} - \sigma_{b,j+1}) \right\},$$

where $\mathsf{h}$ is a real-valued function of a single variable

$$\mathsf{h}(s) = \begin{cases} \ln\left(2\cosh(\pi s)\right), & d = 3, \\ -\ln\left(2\cosh(\pi s)\right) + \frac{1}{4}\left[f\left(\frac{1}{2} + is\right) + f\left(\frac{1}{2} - is\right)\right], & d = 5. \end{cases}$$

- The non-perturbative contribution $Z_{\text{non-pert}}$ is identically 1 in 3d [110], and accounts for codimension-4 field configurations in 5d [111, 112]. The mass of 5d instantons is $\propto 1/g_{\text{YM}}^2$, thus they are non-perturbatively suppressed away from the superconformal point, but can become massless in the conformal limit. As argued below, we will enforce a procedure such that, schematically,

$$Z_{\text{non-pert}} = 1 + e^{-P(\cdots)},$$

where the dots are a strictly positive number as long as $g_{\text{YM},j}^2 > 0$. In the following, we will work with finite gauge coupling in 5d, so that the SCFT flows to a gauge theory and the picture described so far applies. Then, $Z_{\text{non-pert}}$ is safely neglected in the regime $P \gg 1$. The strong coupling limit is taken at the end of the computation.

Convergence of the perturbative $\mathbb{S}^d$ partition function, i.e. setting $Z_{\text{non-pert}} \to 1$, imposes $F_j \geq 2N_j - N_{j-1} - N_{j+1}$ in $d = 3$ and $F_j \leq 2N_j - N_{j-1} - N_{j+1}$ in $d = 5$, making balanced quivers a preferred choice.

Before taking the large $N$ limit, a remark is in order. Turning on massive deformations of large $N$ field theories, there can be phase transitions (typically of third order) when such mass parameters cross a given threshold. It was proven in [109] that balanced linear quivers do not have such phase transitions.

### 3.2.4 Setting up the large $N$ limit

The first step to take the large $N$ limit is to rewrite the partition function in the form

$$\mathcal{Z}_{\mathbb{S}^d} = \int_{\mathbb{R}^{|\vec{N}|}} \mathrm{d}\vec{\sigma} \; e^{-S_{\text{eff}}(\vec{\sigma}, \vec{m})} Z_{\text{non-pert}} \,,$$

where, with the above notation,

$$\begin{aligned}
S_{\text{eff}} = {} & \sum_{j=1}^{P-1} \sum_{a=1}^{N_j} \left[ \delta_{d,5} m_j^{\text{YM}} \sigma_{a,j}^2 + \sum_{\alpha=1}^{\mathscr{F}} F_{\alpha,j} \mathsf{h}(\sigma_{a,j} - m_\alpha) \right] \\
& + \sum_{j=1}^{P-1} \sum_{a=1}^{N_j} \left[ \left( \sum_{b \neq a} \mathsf{v}(\sigma_{a,j} - \sigma_{b,j}) \right) + \left( \sum_{b=1}^{N_{j+1}} \mathsf{h}(\sigma_{a,j} - \sigma_{b,j+1}) \right) \right] + \mathfrak{B} \,.
\end{aligned} \tag{3.5}$$

The last term $\mathfrak{B}$ generically includes terms from the boundary of the quiver. Here we are over-counting the contribution from a bifundamental hypermultiplet at the last node, thus $\mathfrak{B}$ serves to correct this extra counting. It follows from power counting, and is shown explicitly in [87], that it is subleading in the large $P$ limit, thus we refrain from writing down its explicit form.

It is customary to introduce the eigenvalue density at the $j^{\text{th}}$ node:

$$\rho_j(\sigma) = \frac{1}{N} \sum_{a=1}^{N_j} \delta(\sigma_{a,j} - \sigma) \,.$$

Two caveats: first, notice that now $\sigma$ is a continuous real variable, related to, but different from the $|\vec{N}|$-dimensional real scalar $\vec{\sigma}$. Second, we are normalising all the densities $\rho_j$ with a unique (and so far, arbitrary) integer $N$. This implies

$$\int_{\mathbf{R}} \mathrm{d}\sigma \rho_j(\sigma) = \frac{N_j}{N} \,.$$

With the above definitions at hand, the sums over $j$ and $a$ can be replaced by integrals:

$$\sum_{a=1}^{N_j} f(\sigma_{a,j}) \rightsquigarrow N \int \mathrm{d}\sigma \rho_j(\sigma) f(\sigma) \,; \qquad \sum_{j=1}^{P-1} f_j \rightsquigarrow P \int_0^1 \mathrm{d}z f(z) \,,$$

for arbitrary expressions $f_j$. This allows to pack the eigenvalue densities at each node into a single function of two variables

$$\rho(z, \sigma) : [0,1] \times \mathbf{R} \to \mathbf{R}_{\geq 0} \,,$$

normalised as

$$\int_{\mathbf{R}} \mathrm{d}\sigma \rho(z, \sigma) = \nu(z) \,, \qquad \forall 0 \leq z \leq 1 \,.$$

Putting all together, (3.5) becomes

$$S_{\text{eff}} = PN^2 \int_0^1 dz \int d\sigma \rho(z,\sigma) \left[ \delta_{d,5} \frac{m^{\text{YM}}(z)}{N} \sigma^2 + \sum_{\alpha=1}^{\mathscr{F}} \zeta_\alpha(z) h(\sigma - m_\alpha) \right. \tag{3.6a}$$

$$+ \int_{\phi \neq \sigma} d\phi \rho(z,\phi) v(\sigma - \phi) \tag{3.6b}$$

$$\left. + \frac{1}{2} \int d\phi \left( \rho(z+\delta z, \phi) + \rho(z - \delta z, \phi) \right) h(\sigma - \phi) \right] + \mathfrak{B} . \tag{3.6c}$$

In the last line, we have used the symmetry of the last term to slightly rewrite it [87], with the understanding that, if $z = \frac{j}{P}$, then $z \pm \delta z = \frac{j \pm 1}{P}$. The boundary piece $\mathfrak{B}$ now contains half term from $j = P - 1$ and half from $j = 0$.

### 3.2.5 Long quiver limit

The long quiver limit, $P \gg 1$, was pioneered in [87]. Here we briefly sketch their derivation, with a few variations to deal with mass deformations as well as to treat $m^{\text{YM}}(z)$ in accordance with its geometric engineering.

First, we add and subtract $\rho(z,\phi) h(\sigma - \phi)$ in (3.6c):

$$\frac{1}{2} \left( \rho(z+\delta z, \phi) + \rho(z - \delta z, \phi) \right) h(\sigma - \phi)$$

$$= \frac{1}{2} \left[ \left( \rho(z+\delta z, \phi) - \rho(z, \phi) \right) - \left( \rho(z, \phi) - \rho(z - \delta z, \phi) \right) \right] h(\sigma - \phi) + \rho(z, \phi) h(\sigma - \phi) .$$

We combine the last summand with (3.6b) and define

$$\mathsf{F}_0(s) := \left[ \mathsf{v}(s) \cdot \mathbb{1}_{s \neq 0} + \mathsf{h}(s) \right] / (d-2)^2 ,$$

where $\mathbb{1}_{s \neq 0}$ vanishes at $s = 0$ and is 1 otherwise. Then, we observe that, in the large $P$ limit [87]

$$\left( \rho(z+\delta z, \phi) - \rho(z, \phi) \right) - \left( \rho(z, \phi) - \rho(z - \delta z, \phi) \right) \to \frac{1}{P^2} \frac{\partial}{\partial z^2} \rho(z, \phi) . \tag{3.7}$$

Second, we assume that $\sigma$ grows with $P$. Since the mass parameters are realised as background fields for the flavour symmetry, it is natural to put them on an equal footing and assume the same scaling with $P$:

$$\sigma = (d-2) P^\chi x , \quad m_\alpha = (d-2) P^\chi \mu_\alpha ,$$

with $x, \mu$ independent of $P$. The exponent $\chi > 0$ is determined momentarily by self-consistency of the large $P$ limit. If we were to find $\chi \leq 0$, we would have to reconsider our scaling ansatz.

The corresponding rescaled eigenvalue density is

$$\varrho(z, x) dx = \rho(z, \sigma) d\sigma .$$

Recall from Subsection 3.1.1 that, in 5d, the inverse gauge coupling has the dimensions of a mass parameter. It corresponds to the vev of the background scalar for the $U(1)_I$ global symmetry. Thus, we study the scaling

$$m^{\text{YM}}(z) = P^\chi \varkappa(z) ,$$

for some fixed function $\chi(z)$. Let us emphasise this point: we *do not* consider a 't Hooft scaling. Instead, we treat all the real scalars (dynamical and background) equally. In 5d, this is the most natural standpoint, as explained in Subsection 3.1.1. Moreover, on the Coulomb branch, the one-loop corrected gauge coupling is shifted proportionally to $|\sigma|$, which further advocates for scaling $m^{\text{YM}}$ precisely as $\sigma$ does.

This choice of scaling, distinct from the 't Hooft one, is also consistent with the M-theory realization of the quivers. Indeed, vevs of the scalars, regardless of dynamical or background, all are computed from the volumes of certain two-cycles inside a Calabi–Yau threefold [83]. The only difference between the inverse gauge couplings and the mass parameters, is that the two-cycles giving rise to the former live in the base of a fibration, while the two-cycles for the latter live in the fibre. With our prescription, as opposed to the 't Hooft limit considered in the existing literature, the volumes of all the two-cycles scale equally.[10] In particular, our procedure preserves the fibre/base duality [107, 115–117], whenever the fibration mentioned in the M-theory setup enjoys such duality.

In this approach, the Yang–Mills term is subleading in $N$ but dominant in $P$. The two limits do not commute, and their order of limits is thus relevant (as already pointed out in [109]).

*i)* We first take the large $N$ limit at fixed gauge coupling. The Yang–Mills term is subleading in the effective action. Non-perturbative contributions cannot be neglected at this stage.

*ii)* The large $P$ limit is taken afterwards. This suppresses the non-perturbative contributions, and also treats all massive fields and parameters democratically.

Introducing the functions [87, 88]

$$\mathsf{f}_0(x) = \begin{cases} \frac{\pi}{4}\delta(x), & d = 3, \\ -\frac{27\pi}{8}|x|, & d = 5, \end{cases} \qquad \mathsf{f}_{\text{h}}(x) = \begin{cases} \pi|x|, & d = 3, \\ -\frac{9\pi}{2}|x|^3, & d = 5, \end{cases} \tag{3.8}$$

the large argument behaviour of $\mathsf{F}_0$ and $\mathsf{h}$ is

$$\mathsf{F}_0(P^{\chi}x) \approx P^{\chi(d-4)}\mathsf{f}_0(x), \qquad \mathsf{h}(P^{\chi}x) \approx P^{\chi(d-2)}\mathsf{f}_{\text{h}}(x).$$

These definitions, together with (3.7), allow us to rewrite the effective action (3.6) as:

$$S_{\text{eff}} = P^{1+\chi(d-4)}N^2 \int_0^1 \mathrm{d}z \int \mathrm{d}x \varrho(z,x) \left[ P^{2\chi} \sum_{\alpha=1}^{\mathscr{F}} \zeta_\alpha(z)\mathsf{f}_{\text{h}}(x-\mu) \right. \tag{3.9}$$
$$\left. + \int \mathrm{d}y \left( \varrho(z,y)\mathsf{f}_0(x-y) + P^{2\chi-2}\frac{1}{2}\partial_z^2\varrho(z,y)\mathsf{f}_{\text{h}}(x-y) \right) \right].$$

Subleading contributions in $N$ and $P$ have been neglected. In the first line, recall from (3.2) that $\zeta_\alpha(z)$ involves a factor $1/P^2$. In order to reach an equilibrium configuration, at least two of the terms in (3.9) must be of the same order in $P$ and compete. This requirement imposes

$$2\chi - 2 = 0 \implies \chi = 1.$$

Therefore $S_{\text{eff}}$, and hence $\mathcal{F}$, bear overall factors $N^2 P^{d-3}$.

### 3.2.6 Saddle point equation

At leading order in $N$ and $P$,

$$\ln \mathcal{Z}_{\mathbb{S}^d} \approx \ln \int \mathrm{d}\sigma e^{-S_{\text{eff}}}.$$

---

[10]LSa thanks Michele Del Zotto for suggestions on this point.

From (3.9), the integrand is suppressed both in $N$ and $P$, meaning that the leading order contribution comes from the saddle point configuration of $S_{\text{eff}}$. We thus need to minimise it over the space of probability densities $\varrho(z,x)$.

Taking the saddle point equation for (3.9) and acting onto it with $\left(\frac{\partial}{\partial x}\right)^{d-2}$, we finally get

$$\frac{1}{4}\partial_x^2 \varrho(z,x) + \partial_z^2 \varrho(z,x) + \sum_{\alpha=1}^{\mathscr{F}} P^2 \zeta_\alpha(z)\delta(x-\mu_\alpha) = 0\,, \qquad (3.10)$$

both in 3d and 5d. The saddle point equation (3.10) is a Poisson equation, and it is a modification of [87, Eq.(2.36)] by

$$\zeta(z)\delta(x) \rightsquigarrow \sum_{\alpha=1}^{\mathscr{F}} \zeta_\alpha(z)\delta(x-\mu_\alpha)\,,$$

meaning that inserting mass deformations has a controlled effect on the long quiver limit.

By linearity, (3.10) admits a solution

$$\varrho(z,x) = \sum_{\alpha=1}^{\mathscr{F}} \varrho_\alpha(z,x)\,, \qquad (3.11)$$

and, generalizing [55,74], we get

$$\varrho_\alpha(z,x) = \pi \sum_{k=1}^{\infty} k\mathfrak{a}_{\alpha,k} \sin(k\pi z)e^{-2\pi k|x-\mu_\alpha|}\,. \qquad (3.12)$$

### 3.2.7 Match with the supergravity solution

We emphasise that the saddle point equation (3.10) agrees precisely with the Poisson equation (2.10) derived in supergravity. To show this, we start with the case of vanishing masses. We make the identifications

$$z = \eta/P\,, \qquad x = \sigma/(2P) \qquad (3.13)$$

(here $\sigma$ is the holographic variable). It is convenient to switch to the normalisation of [87], by introducing an eigenvalue density $\hat{\varrho}(\eta,\sigma) = N\varrho(z,x)$ normalised to $N$. Analogously, denote $F(\eta) = N\zeta(Pz)$ the flavour rank function without Veneziano normalisation. Then, (3.10) becomes

$$\partial_\eta^2 \hat{\varrho} + \partial_\sigma^2 \hat{\varrho} + 2PF(\eta)\delta(\sigma) = 0\,.$$

Recall that $F(\eta) = \mathcal{R}''(\eta)$, in the conventions of (2.3) for the $'$-derivative, which yields opposite sign with respect to the $\partial_z$-derivative. Dividing by $2P$ we get

$$\partial_\eta^2\left(\frac{\hat{\varrho}}{2P}\right) + \partial_\sigma^2\left(\frac{\hat{\varrho}}{2P}\right) - \partial_\eta^2 \mathcal{R}(\eta)\delta(\sigma) = 0\,.$$

Acting with the $'$-derivative twice on the supergravity equation (2.10) and identifying

$$\hat{W}'' = \frac{\hat{\varrho}}{2P}\,, \qquad (3.14)$$

where the differentiation only involves the $\eta$-dependence, we find perfect agreement with the saddle point equation derived in QFT.

The argument extends immediately away from the superconformal case. The presence of masses simply splits the flavour and gauge rank functions, and we identify the positions $\sigma_\alpha$

in supergravity with $2P\mu_\alpha$ in QFT. For instance, for $\mathscr{F} = 2$, $\mu_1 = 0, \mu_2 = \mu$, the identifications above yield the supergravity equation (2.26) from the saddle point equation (3.10).

It is instructive to rewrite the QFT effective action (3.9) in the supergravity language, substituting (3.13)-(3.14). One finds:

$$
S_{\text{eff}} = P^{\chi(d-4)}N \int_0^P d\eta \int_{-\infty}^{+\infty} d\sigma \, \partial_\eta^2 \hat{W}(\sigma, \eta) \left\{ -P^{2\chi} \sum_{\alpha=1}^{\mathscr{F}} \partial_\eta^2 \mathcal{R}(\eta) \frac{f_h(\sigma - \sigma_\alpha)}{(2P)^{\chi(d-2)}} \right.
$$
$$
\left. + N \int_{-\infty}^{+\infty} d\tilde{\sigma} \left[ \partial_\eta^2 \hat{W}(\tilde{\sigma}, \eta) \frac{f_0(\sigma - \tilde{\sigma})}{(2P)^{\chi(d-4)}} + P^{2\chi} \frac{1}{2} \partial_\eta^4 \hat{W}(\tilde{\sigma}, \eta) \frac{f_h(x - y)}{(2P)^{\chi(d-2)}} \right] \right\}.
$$

The powers of $P$ cancel out in this expression, regardless of the value of $\chi$, while each power of $N$ accompanies a $\sigma$-integral. Simplifying the expression, one arrives at

$$
S_{\text{eff}} = \frac{N}{2^{\chi(d-4)}} \int_0^P d\eta \int_{-\infty}^{+\infty} d\sigma \, \partial_\eta^2 \hat{W}(\sigma, \eta) \cdot \partial_\eta^2 \left\{ -\sum_{\alpha=1}^{\mathscr{F}} \mathcal{R}(\eta) \cdot \frac{1}{2^{2\chi}} f_h(\sigma - \sigma_\alpha) \right.
$$
$$
\left. + N \int_{-\infty}^{+\infty} d\tilde{\sigma} \left[ \hat{W}(\tilde{\sigma}, \eta) f_0(\sigma - \tilde{\sigma}) + \partial_\eta^2 \hat{W}(\tilde{\sigma}, \eta) \cdot \frac{1}{2^{2\chi+1}} f_h(x - y) \right] \right\}.
$$

## 3.3 Free energies in massive deformations of 5d and 3d SCFTs

In this subsection, we compute the free energy for arbitrary $\mathscr{F}$ and present the general solution in 5d and 3d. The SCFT result of [87, 89] is recovered upon setting all masses to zero.

### 3.3.1 Free energies in 5d and 3d

The free energy $\mathcal{F}$ is defined in (3.1). It is computed at leading order in $N, P$ plugging our solution (3.11)-(3.12) into $S_{\text{eff}}$ and evaluating it on-shell.

The derivation is identical to [87, Sec.3] and [89, Sec.3], upon adjusting to our conventions and replacing

$$
\varrho(z, x) \rightsquigarrow \sum_{\alpha=1}^{\mathscr{F}} \varrho_\alpha(z, x),
$$
$$
\zeta(z) f_h(x) \rightsquigarrow \sum_{\alpha=1}^{\mathscr{F}} \zeta_\alpha(z) f_h(x - \mu_\alpha).
$$

Using this and (3.9), we arrive at the result

$$
\mathcal{F} = \frac{(-1)^{\frac{d-3}{2}}}{2} N^2 P^{d-3} \int_0^1 dz \int_{-\infty}^{+\infty} dx \left( \sum_{\alpha=1}^{\mathscr{F}} \varrho_\alpha(z, x) \right) \left( P^2 \sum_{\beta=1}^{\mathscr{F}} \zeta_\beta(z) f_h(x - \mu_\beta) \right)
$$
$$
= \frac{(-1)^{\frac{d-3}{2}}}{2} N^2 P^{d-3} \sum_{\alpha,\beta=1}^{\mathscr{F}} \int_0^1 dz \, P^2 \zeta_\beta(z) \int_{-\infty}^{+\infty} dx \, \varrho_\alpha(z, x) f_h(x - \mu_\beta). \tag{3.15}
$$

Importantly, by a straightforward generalization of the derivation in [87, Sec.3] and denoting $S_{\text{fund}}$ the contribution of the fundamental hypermultiplets to the action, we arrive at the identity

$$
S_{\text{eff}}\Big|_{\text{on shell}} = \frac{1}{2} S_{\text{fund}}\Big|_{\text{on shell}},
$$

which is in fact a general property of matrix models in the planar limit, following from the equilibrium equation (see e.g. [118]).

Formula (3.15) yields the leading contribution to the free energy. One may also use the eigenvalue densities (3.12) to evaluate the subleading contributions, such as the boundary terms $\mathfrak{B}|_{\text{on shell}}$.

Separating the cases $\alpha = \beta$ and $\alpha \neq \beta$, and shifting variables, we rewrite (3.15) in the form

$$\mathcal{F} = \sum_{\alpha=1}^{\mathscr{F}} \mathcal{F}_\alpha + \sum_{\alpha \neq \beta} \mathcal{F}_{\alpha,\beta}^{\text{int}} \,. \tag{3.16}$$

Here $\mathcal{F}_\alpha$ is the free energy of a 5d or 3d balanced, linear quiver $\mathcal{Q}_\alpha$ with reduced gauge rank function $\nu_\alpha(z)$ and flavour rank function $\zeta_\alpha(z)$. From the explicit form of $\varrho_\alpha$ in (3.12) and $\zeta_\alpha$ in (3.2), each individual contribution reads

$$\frac{(-1)^{\frac{d-3}{2}}}{N^2 P^{d-3}} \mathcal{F}_\alpha = \frac{1}{2} \int_0^1 dz \left( \pi^2 \sum_{\ell=1}^\infty \mathfrak{a}_{\alpha,\ell} \ell^2 \sin(\ell \pi z) \right) \tag{3.17a}$$

$$\int_{-\infty}^{+\infty} dx \left( \pi \sum_{k=1}^\infty \mathfrak{a}_{\alpha,k} k \sin(k\pi z) e^{-2\pi k |x-\mu_\alpha|} \right) f_{\text{h}}(x - \mu_\alpha)$$

$$= \frac{\pi^3}{2} \sum_{\ell,k=1}^\infty \mathfrak{a}_{\beta,\ell} \mathfrak{a}_{\alpha,k} k \ell^2 \int_0^1 dz \sin(\ell \pi z) \sin(k \pi z) \int_{-\infty}^{+\infty} dx \, e^{-2\pi k |x|} f_{\text{h}}(x) \tag{3.17b}$$

$$= \frac{\pi^3}{4} \sum_{k=1}^\infty \mathfrak{a}_{\alpha,k}^2 k^3 \int_{-\infty}^{+\infty} dx \, e^{-2\pi k |x|} f_{\text{h}}(x) \,. \tag{3.17c}$$

The terms in the second sum in (3.16) encode the pairwise coupling of the quivers $\mathcal{Q}_\alpha$ and $\mathcal{Q}_\beta$. Its appearance encapsulates the main technical novelty of the present analysis. Note that the quivers $\mathcal{Q}_\alpha$ only interact pairwise, and the free energy configuration may be drawn as a complete graph of $\mathscr{F}$ vertices, with vertex set $\{\mathcal{Q}_\alpha\}$. For example:

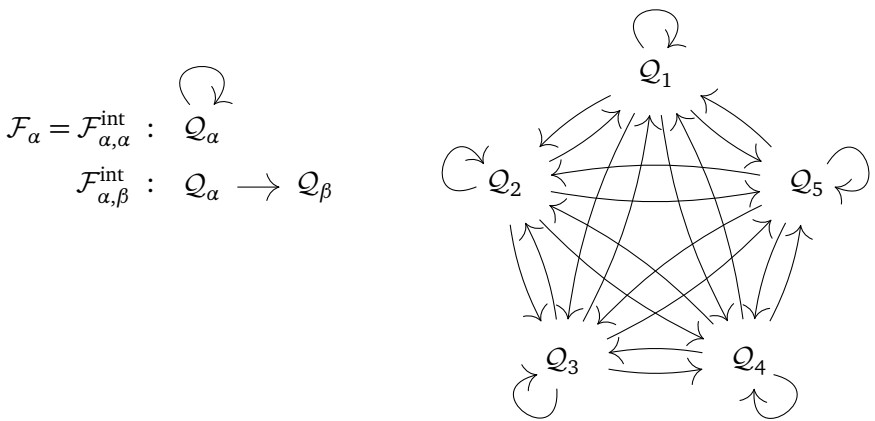

From the Type IIB string theory viewpoint, the arrows in this picture encode the free energy of strings stretched between stacks of D$d$ colour branes. When the stacks are pairwise separated, the string excitations become massive. An example is provided in Figure 11 in Appendix C.2.

To evaluate $\mathcal{F}_{\alpha,\beta}^{\text{int}}$, exploiting again (3.12) and (3.2), we have

$$\frac{(-1)^{\frac{d-3}{2}}}{N^2 P^{d-3}} \mathcal{F}_{\alpha,\beta}^{\text{int}} = \frac{1}{2} \int_0^1 dz \left( \pi^2 \sum_{\ell=1}^{\infty} \mathfrak{a}_{\beta,\ell} \ell^2 \sin(\ell \pi z) \right) \tag{3.18a}$$

$$\int_{-\infty}^{+\infty} dx \left( \pi \sum_{k=1}^{\infty} \mathfrak{a}_{\alpha,k} k \sin(k\pi z) e^{-2\pi k|x-\mu_\alpha|} \right) f_{\text{h}}(x - \mu_\beta)$$

$$= \frac{\pi^3}{2} \sum_{\ell,k=1}^{\infty} \mathfrak{a}_{\beta,\ell} \mathfrak{a}_{\alpha,k} k \ell^2 \int_0^1 dz \sin(\ell \pi z) \sin(k\pi z) \int_{-\infty}^{+\infty} dx e^{-2\pi k|x|} f_{\text{h}}(x - \mu_{\beta\alpha}) \tag{3.18b}$$

$$= \frac{\pi^3}{4} \sum_{k=1}^{\infty} \mathfrak{a}_{\alpha,k} \mathfrak{a}_{\beta,k} k^3 \mathcal{I}_n(\mu_{\beta\alpha}). \tag{3.18c}$$

In (3.18b) we have introduced the notation $\mu_{\beta\alpha} := \mu_\beta - \mu_\alpha$, and we have defined

$$\mathcal{I}_k(\mu) := \int_{-\infty}^{+\infty} dx \; e^{-2\pi k|x|} f_{\text{h}}(x - \mu) \tag{3.19}$$

in (3.18c), whose explicit form differs in 5d and 3d.

From (3.16)-(3.18c) we already learn an important lesson: because the coefficients $\mathfrak{a}_k$ grow linearly in $P$, we predict the scaling:

$$\mathcal{F} \propto (-1)^{\frac{d-1}{2}} N^2 P^{d-1},$$

which interpolates between and agrees with the scaling found in 5d [87] and 3d [89]. This, however, will be the behaviour of a quiver with a *generic* distribution of flavour ranks. Concrete examples, as for instance the $T_N$ theory, will show different scaling — see also Appendix B.

### 3.3.2 Free energy in 5d

To determine the free energy in 5d, it only remains to evaluate the integral (3.19). Plugging $f_{\text{h}}(x)|_{d=5}$ from (3.8), we get

$$\mathcal{I}_k(\mu)|_{d=5} = -\frac{9\pi}{2} \int_{-\infty}^{+\infty} dx e^{-2\pi k|x|} |x - \mu|^3$$

$$= -\frac{27}{8\pi^3 k^4} \left( e^{-2\pi k|\mu|} + 2\pi k|\mu| + \frac{4}{3}(\pi k|\mu|)^3 \right). \tag{3.20}$$

Inserting this expression in the general formula (3.15) yields the final result

$$\mathcal{F}|_{d=5} = \sum_{\alpha=1}^{\mathcal{F}} \underbrace{\frac{27}{32} N^2 P^2 \sum_{k=1}^{\infty} \frac{1}{k} \mathfrak{a}_{\alpha,k}^2}_{\mathcal{F}_\alpha|_{d=5}} \tag{3.21}$$

$$+ 2 \sum_{1 \le \alpha < \beta \le \mathcal{F}} \underbrace{\frac{27}{32} N^2 P^2 \sum_{k=1}^{\infty} \frac{1}{k} \mathfrak{a}_{\alpha,k} \mathfrak{a}_{\beta,k} \left( e^{-2\pi k|\mu_{\beta\alpha}|} + 2\pi k|\mu_{\beta\alpha}| + \frac{4}{3}(\pi k|\mu_{\beta\alpha}|)^3 \right)}_{\mathcal{F}_{\alpha,\beta}^{\text{int}}|_{d=5}}.$$

### 3.3.3 Free energy in 3d

In 3d, the integral (3.19) is simpler. Recall from (3.8) that $f_h(x) = \pi|x|$. Then

$$
\begin{aligned}
\mathcal{I}_k(\mu)|_{d=3} &= \pi \int_{-\infty}^{+\infty} dx e^{-2\pi k|x|}|x-\mu| \\
&= \frac{1}{2\pi k^2}\left(e^{-2\pi k|\mu|} + 2\pi k|\mu|\right).
\end{aligned}
$$

Plugging this expression in formula (3.15) we find the general result

$$
\begin{aligned}
\mathcal{F}|_{d=3} = \sum_{\alpha=1}^{\mathscr{F}} \underbrace{\frac{\pi^2}{8}N^2 \sum_{k=1}^{\infty} k\mathfrak{a}_{\alpha,k}^2}_{\mathcal{F}_\alpha|_{d=3}} \\
+ 2\sum_{1\leq\alpha<\beta\leq\mathscr{F}} \underbrace{\frac{\pi^2}{8}N^2 \sum_{k=1}^{\infty} k\mathfrak{a}_{\alpha,k}\mathfrak{a}_{\beta,k}\left(e^{-2\pi k|\mu_{\beta\alpha}|} + 2\pi k|\mu_{\beta\alpha}|\right)}_{\mathcal{F}_{\alpha,\beta}^{\text{int}}\big|_{d=3}}.
\end{aligned}
$$

(3.22)

At $d=3$, it is worth commenting on the possibility of introducing FI parameters $\xi_j$, which, in the long quiver limit, would give rise to a function $\xi(z)$. Remember that they would introduce a linear dependence on $\sigma$ in the effective action. These contributions would be subleading in $N$ and drop out in the large $N$ limit. One may retain the dependence on the FI parameters by imposing a 't Hooft scaling on them, but this would artificially break the symmetric role between FI parameters and mass parameters in 3d $\mathcal{N}=4$.

In conclusion, turning on FI parameters and treating them consistently with 3d $\mathcal{N}=4$ mirror symmetry in the long quiver limit explained in Subsection 3.2.5 does not alter the eigenvalue density nor the free energy, at leading order in $N$. Imposing a scaling on the FI parameters to keep track of them, even if inconsistent with mirror symmetry, would not change the eigenvalue density $\varrho(z,x)$, but may give an extra contribution to the free energy at leading order. Denoting $\hat{\xi}(z)$ the scaled version of $\xi(z)$, this additional contribution is the first moment of the measure $\varrho(z,x)dx$,

$$
S_{\text{FI}}\Big|_{\text{on shell}} = N^2 \int_0^1 dz\,\hat{\xi}(z) \int_{-\infty}^{+\infty} x\varrho(z,x)dx = 0,
$$

which vanishes because $\varrho(z,x)$ is an even function of $x$ for every $0 < z < 1$. Therefore, the FI parameter does not change the leading order value of $\mathcal{F}$ in the long quiver limit of balanced 3d $\mathcal{N}=4$ quivers. This is not inconsistent with mirror symmetry, since the mirror of a balanced linear quiver is not balanced (unless there is only one flavour node) [55], and our argument for the vanishing of the FI contribution only holds for *long* and *balanced* quivers.[11]

Another feature of theories with four or more supercharges in $d=3$ is the holomorphic dependence on the mass [119], which allows for a shift $m \mapsto m + i\varepsilon$ with $m \in \mathbf{R}$ and $|\varepsilon| < \frac{1}{2}$. Thus the imaginary part $\varepsilon$ must be kept fixed in the long quiver limit, while the real part $m$ scales linearly with $P$. We use the trigonometric identity

$$
\cosh(\pi(\sigma - m - i\varepsilon)) = \cos(\pi\varepsilon)\cosh(\pi(\sigma - m)) - i\sin(\pi\varepsilon)\sinh(\pi(\sigma - m)),
$$

---

[11]The only exception to this statement is the $T[SU(N)]$ theory. In that case, $P$ and $N$ are not independent and the whole scaling argument must me revisited, see [13, 88]. Using that the mirror map gives $\xi_j$ as a difference of the mass parameters, the mirror of the mass deformation we consider would correspond to a single $\xi_j \neq 0$ which grows linearly in $N$.

and take the logarithm on both sides. We observe that the complexification of the real mass parameter does not alter the real part of the effective action. In the long quiver limit, it produces an imaginary shift of the function h by $-\pi\varepsilon\,\text{sign}(x)$. This imaginary part remains finite in the long quiver limit and drops out of the computations. In conclusion, the holomorphic dependence on $m$ persists trivially in the long quivers, and the complexification $m \mapsto m + i\varepsilon$ does not change $\mathcal{F}|_{d=3}$ at leading order. This was expected, because the leading contribution comes from regions of the integration domain with $\sigma \propto P$, whereas $|\varepsilon| < \frac{1}{2}$.

### 3.3.4 Holographic match in 5d and 3d SCFTs

Let us take the SCFT limit of our result, with $\mu_\alpha \to 0$, and show the agreement with the supergravity result of Section 2.4. We do it first in 5d, by setting all masses to vanish in (3.21). The polynomial terms in $|\mu|$ vanish, and the exponential dependence disappears. Altogether, we have:

$$
\begin{aligned}
\lim_{\vec{\mu} \to 0} \mathcal{F}|_{d=5} &= \frac{27}{32} N^2 P^2 \left[ \sum_{\alpha=1}^{\mathscr{F}} \sum_{k=1}^{\infty} \frac{1}{k} \mathfrak{a}_{\alpha,k}^2 + \sum_{1 \le \alpha < \beta \le \mathscr{F}} \sum_{k=1}^{\infty} \frac{2}{k} \mathfrak{a}_{\alpha,k} \mathfrak{a}_{\beta,k} \right] \\
&= \frac{27}{32} N^2 P^2 \sum_{k=1}^{\infty} \frac{1}{k} \left( \sum_{\alpha=1}^{\mathscr{F}} \mathfrak{a}_{\alpha,k} \right)^2 \\
&= \frac{27}{32} N^2 P^2 \sum_{k=1}^{\infty} \frac{1}{k} \mathfrak{a}_k^2 \,,
\end{aligned}
$$

which is the free energy of the quiver with "reduced" rank function is $\nu(z)$, whose Fourier coefficients are $\mathfrak{a}_k = \frac{1}{N} R_k$. Comparing with (2.20), we obtain the relation:

$$
d = 5 \; : \qquad \mathcal{F} = \frac{27\pi^6}{4} c_{hol} \,. \tag{3.23}
$$

We find agreement between the free energy and the holographic central charge, up to an arbitrary numerical factor in the definition of $c_{hol}$. The overall coefficient of 27 is recurring in 5d, see for instance [120, Eq.(5.11)], and appears to be due to the universal relation $\mathcal{F} = \omega^3 \mathcal{F}_{\text{univ.}}$. Here $\omega$ is related to the equivariant parameters on a squashed $\mathbb{S}^5$, with $\omega = 3$ on the round sphere.[12]

In 3d, turning off all mass deformations in (3.22), the linear terms in $\mathcal{F}_{\alpha,\beta}^{\text{int}}$ vanish, and the exponential terms go to 1. We get:

$$
\begin{aligned}
\lim_{\vec{\mu} \to 0} \mathcal{F}|_{d=3} &= \frac{\pi^2}{8} N^2 \left[ \sum_{\alpha=1}^{\mathscr{F}} \sum_{k=1}^{\infty} k \mathfrak{a}_{\alpha,k}^2 + \sum_{1 \le \alpha < \beta \le \mathscr{F}} \sum_{k=1}^{\infty} 2k \mathfrak{a}_{\alpha,k} \mathfrak{a}_{\beta,k} \right] \\
&= \frac{\pi^2}{8} N^2 \sum_{k=1}^{\infty} k \left( \sum_{\alpha=1}^{\mathscr{F}} \mathfrak{a}_{\alpha,k} \right)^2 \\
&= \frac{\pi^2}{8} N^2 \sum_{k=1}^{\infty} k \mathfrak{a}_k^2 \,,
\end{aligned}
$$

thus recovering the reduced rank function $\nu(z)$, akin to the 5d case. Comparing with (2.23), we find:

$$
d = 3 \; : \qquad \mathcal{F} = 4\pi c_{hol} \,, \tag{3.24}
$$

so that the free energy matches the holographic central charge, up to an overall normalisation.

---

[12] $\mathcal{F}$ in (3.1) equals $\mathcal{F}_{\vec{\omega}}$ of [87, Eq.(2.23)]. In [74], $c_{hol}$ was compared with $\mathcal{F}_{\text{univ.}} = 3^{-3} \mathcal{F}_{\vec{\omega}}$ read off from of [87]. Accounting for the factor $\omega^3 = 27$, our eq. (3.23) agrees with [74, Eq.(3.9)].

### 3.4 Two interacting linear quivers

In this subsection we discuss the case $\mathscr{F} = 2$ and relate it to the holographic result of Subsection 2.5. Without loss of generality, we set $m_1 = 0, m_2 = m$.

The final result for the free energy can be read off from the general procedure of Subsection 3.2. However, we will rederive it using a different approach. We will reformulate the large $N$ limit in a way which probes the two singularities of the Coulomb branch of $\mathcal{Q}$. The analysis of Subsection 3.1, and especially the factorisation of the Coulomb branch into two reduced branches plus decoupled massive zero-modes, will be manifest in the ensuing procedure.

Despite detailing it only for the case of two ranks functions, the procedure we show below extends to an arbitrary number $\mathscr{F}$ of mass scales and rank functions.

#### 3.4.1 Change of variables to probe the singularities

The data of the problem include the flavour rank functions

$$\zeta_\alpha(z) = \frac{\pi^2}{P^2} \sum_{k=1}^{\infty} \mathfrak{a}_{\alpha,k} k^2 \sin(k\pi z), \qquad \alpha = 1, 2,$$

from which one extracts the reduced gauge rank functions $\nu_\alpha(z)$, as in (3.3). We then define the gauge ranks $N_{\alpha,j} = N \nu_\alpha(j/P)$, $\alpha = 1, 2$. By construction, $N_{1,j} + N_{2,j} = N_j$. To probe the Coulomb branch singularities at 0 and $m$, we leave the variables

$$\sigma_{a,j}, \quad j = 1, \ldots, P-1, \ a = 1, \ldots, N_{1,j}$$

untouched, and define the new variables

$$\phi_{\dot{a},j} = \sigma_{\dot{a}+N_{1,j},j} - m, \quad j = 1, \ldots, P-1, \ \dot{a} = 1, \ldots, N_{2,j}.$$

Undotted indices run over the gauge ranks $N_{1,j}$ while dotted indices run over the gauge ranks $N_{2,j}$. With these new variables, the contributions to the sphere partition function become:

- from the vector multiplet,

$$Z_{1\text{-loop}}^{\text{vec}} = \prod_{j=1}^{P-1} \left[ \prod_{1 \leq a \neq b \leq N_{1,j}} \exp\left\{-\mathsf{v}(\sigma_{a,j} - \sigma_{b,j})\right\} \right] \times \left[ \prod_{1 \leq \dot{a} \neq \dot{b} \leq N_{2,j}} \exp\left\{-\mathsf{v}(\phi_{\dot{a},j} - \phi_{\dot{b},j})\right\} \right]$$

$$\times \left[ \prod_{a=1}^{N_j} \prod_{\dot{b}=1}^{N_j} \exp\left\{-\mathsf{v}(\sigma_{a,j} - \phi_{\dot{b},j} + m)\right\} \right],$$

- from the fundamental hypermultiplets

$$Z^{\text{fund}} = \prod_{j=1}^{P-1} \left[ \prod_{a=1}^{N_{1,j}} \exp\left\{-F_{1,j}\mathsf{h}(\sigma_{a,j})\right\} \right] \times \left[ \prod_{\dot{a}=1}^{N_{2,j}} \exp\left\{-F_{2,j}\mathsf{h}(\phi_{\dot{a},j})\right\} \right]$$

$$\times \left[ \prod_{a=1}^{N_{1,j}} \exp\left\{-F_{2,j}\mathsf{h}(\sigma_{a,j} - m)\right\} \right] \times \left[ \prod_{\dot{a}=1}^{N_{2,j}} \exp\left\{-F_{1,j}\mathsf{h}(\phi_{\dot{a},j} + m)\right\} \right],$$

and from the bifundamental hypermultiplets,

$$Z^{\text{bif}} = \prod_{j=1}^{P-2} \left[ \prod_{a=1}^{N_{1,j}} \prod_{b=1}^{N_{1,j+1}} \exp\left\{-\mathsf{h}(\sigma_{a,j} - \sigma_{b,j+1})\right\} \right] \times \left[ \prod_{\dot{a}=1}^{N_{2,j}} \prod_{\dot{b}=1}^{N_{2,j+1}} \exp\left\{-\mathsf{h}(\phi_{\dot{a},j} - \phi_{\dot{b},j+1})\right\} \right]$$

$$\times \left[ \prod_{a=1}^{N_{1,j}} \prod_{\dot{b}=1}^{N_{2,j+1}} \exp\left\{-\mathsf{h}(\sigma_{a,j} - \phi_{\dot{b},j+1} + m)\right\} \right] \times \left[ \prod_{\dot{a}=1}^{N_{2,j}} \prod_{b=1}^{N_{1,j+1}} \exp\left\{-\mathsf{h}(\phi_{\dot{a},j} - \sigma_{b,j+1} - m)\right\} \right].$$

In the new variables, we readily find that the effective action splits according to:

$$S_{\text{eff}} = S_{\text{eff},1} + S_{\text{eff},2} + S_{\text{int}}(m),$$

with $S_{\text{eff},\alpha}$ the effective action for the *superconformal* quiver $\mathcal{Q}_\alpha$. Each such effective action does not depend on the mass and is of the form analyzed in [87, 89]. The dependence on $m$ is entirely captured by the interaction among the two quivers in $S_{\text{int}}$.

The Hanany–Witten brane setup is easily read off from the sphere partition in the adjusted variables. Examples are drawn in Figure 10 in Appendix C.1 and in Figure 11 in Appendix C.2.

### 3.4.2 Change of variables, Coulomb branches and holography

The variables $\vec{\sigma}, \vec{\phi}$ are adjusted coordinates for the Coulomb branches

$\vec{\sigma}$ : $\text{CB}[\mathcal{Q}_1]$ of the quiver $\mathcal{Q}_1$, with reduced gauge and flavour rank functions $\nu_1$ and $\zeta_1$, and

$\vec{\phi}$ : $\text{CB}[\mathcal{Q}_2]$ of the quiver $\mathcal{Q}_2$, with reduced gauge and flavour rank functions $\nu_2$ and $\zeta_2$.

In particular, the singularity expected at the origin of the Coulomb branch of $\mathcal{Q}_2$ is indeed located at $\vec{\phi} = 0$. See [92] for related analysis.

This identification however is not exact at finite $m$. We have learnt from Subsection 3.1.2 that there is a region, between the two conical singularities of $\text{CB}[\mathcal{Q}_1]$ and $\text{CB}[\mathcal{Q}_2]$, in which a more complicated geometry arises. The matrix model neatly sees this effect after the change of variables. For $\sigma_{a,j} < 0$ and $\phi_{\dot{a},j} > 0$ there are no massless modes and the approximate description as two disjoint Coulomb branches is accurate. However, in the region $0 < \sigma_{a,j} < m$ and $-m < \phi_{\dot{a},j} < 0$ there are additional massless states. These are read off from the zeros of the argument in the functions $\mathsf{v}, \mathsf{h}$. In the limits $m \to 0$ and $m \to \infty$ we recover the conformal situation, with massless states only at the origin.

This matrix model analysis is in perfect agreement with the supergravity discussion of Subsection 2.7. It was noted there that the region where the $\text{AdS}_{d+1}$ solution is not reliable is precisely $0 < \sigma < \sigma_0$.

### 3.4.3 Change of variables, Coulomb branch singularities and balanced quivers

The change of variables provides an insightful rewriting of the integrand in the sphere partition function, as it corresponds to zoom in close to a region of $\text{CB}[\mathcal{Q}]$. However, being just a change of variables, it yields the same value for every choice of $N_{2,j}$. The point we want to make is that, among all possible ways to rewrite the integrand, i.e. all possible choices of the integers $N_{2,j}$, one will be suited to describe the quivers that survive at the end of the RG flow. Similar methods have been employed in [92].

The idea is to zoom in close to all possible Coulomb branch singularities and compare their suppression factor as a function on $m$. The least suppressed one will dominate in the IR limit $m \to \infty$. Schematically, let us denote by

$$\mathcal{F}_\alpha\left[\{N_{2,j}\}\right], \qquad \alpha = 1, 2$$

the free energies of the two quivers read off from the change of variables, namely $\mathcal{Q}_1$ with Coulomb branch parameter $\vec{\sigma}$ and $\mathcal{Q}_2$ with Coulomb branch parameter $\vec{\phi}$, for a concrete choice of gauge ranks $\{N_{2,j}\}$. We have seen that these two terms are independent of $m$. Besides, let us denote $\mathcal{F}_{\text{dec.}}[\{N_{2,j}\}](m)$ the free energy coming from the mass dependent contributions. We then have

$$\mathcal{F} = \mathcal{F}_{\text{dec.}}[\{N_{2,j}\}](m) + \sum_{\alpha=1}^{2} \mathcal{F}_\alpha[\{N_{2,j}\}].$$

The quantity $\mathcal{F}_{\text{dec.}}$ is carefully computed below, but for the moment, it suffices to notice that it will damp the sphere partition function, proportionally to the number of fields that acquire a large mass along the RG flow. In a nutshell, as $|m| \to \infty$,

$$(-1)^{\frac{d-1}{2}} \mathcal{F}_{\text{dec.}}[\{N_{2,j}\}](m) \approx -|f_{\text{h}}(m)|\left(\# \text{ of massive modes for the given } \{N_{2,j}\}\right).$$

We signs and absolute values in this formula serve to emphasise that this factor suppressed $\mathcal{Z}_{\mathbb{S}^d}$. We conclude that, in order to isolate and read off the quivers $\mathcal{Q}_\alpha$ that appear in the IR, the choice of collection $\{N_{2,j}\}$ is determined by minimizing such number. This statement resonates with [92, Sec.5].

Thanks to the fact that the quiver we begin with is balanced, we can prove by a counting argument that the minimum is precisely given by the collection $\{N_{2,j}\}$ such that the IR quivers $\mathcal{Q}_\alpha$ are balanced. We report the explicit calculation in Appendix C.2.3 for the simplest example of SQCD, but the argument holds for every balanced quiver.

We stress that this counting argument is general, but to find the explicit solution we have relied on the fact that the UV quiver is balanced. Without a relation between gauge and flavour ranks, it would be harder in practice to figure out the correct splitting of the quiver.

### 3.4.4 Long quiver limit after change of variables

We now apply the large $N$ and large $P$ limits to the quiver, after the change of variables. We define the eigenvalue densities

$$\rho_{1,j}(\sigma) = \frac{1}{N} \sum_{a=1}^{N_{1,j}} \delta(\sigma - \sigma_{a,j}),$$

$$\rho_{2,j}(\phi) = \frac{1}{N} \sum_{\dot{a}=1}^{N_{2,j}} \delta(\phi - \phi_{\dot{a},j}),$$

corresponding to the two quivers $\mathcal{Q}_1$ and $\mathcal{Q}_2$, whose adjusted variables are $\sigma$ and $\phi$ respectively. From the reasoning of Subsection 3.2, we introduce the scaled variables

$$\sigma = (d-2)Px, \qquad \phi = (d-2)Py, \qquad m = (d-2)P\mu,$$

and the scaled eigenvalue densities $\varrho_1(z,x), \varrho_2(z,y)$. Mimicking the long quiver limit of Subsection 3.2, now with two distinct eigenvalue densities, the terms $S_{\text{eff},\alpha}$ are standard, while for the interaction term we find

$$S_{\text{int}} = S_0 + S_{\text{der}} + S_{\text{fund},1} + S_{\text{fund},2}, \tag{3.25}$$

where

$$S_0 = 2P^{(d-3)}N^2 \int_0^1 \mathrm{d}z \int \mathrm{d}x \varrho_1(z,x) \int \mathrm{d}y \varrho_2(z,y) \mathrm{f}_0(x-y-\mu),$$

$$S_{\mathrm{der}} = \frac{1}{2}P^{(d-3)}N^2 \int_0^1 \mathrm{d}z \int \mathrm{d}x \int \mathrm{d}y \left[ \varrho_1(z,x)\partial_z^2 \varrho_2(z,y) + \varrho_2(z,y)\partial_z^2 \varrho_1(z,x) \right] \mathrm{f}_\mathrm{h}(x-y-\mu),$$

$$S_{\mathrm{fund},1} = N^2 P^{(d-1)} \int_0^1 \mathrm{d}z \zeta_2(z) \int \mathrm{d}x \varrho_1(z,x) \mathrm{f}_\mathrm{h}(x-\mu),$$

$$S_{\mathrm{fund},2} = N^2 P^{(d-1)} \int_0^1 \mathrm{d}z \zeta_1(z) \int \mathrm{d}y \varrho_2(z,y) \mathrm{f}_\mathrm{h}(y+\mu).$$

To solve the equilibrium problem, we must minimise the resulting effective action, now with respect to the two sets of variables $\vec{\sigma}$ and $\vec{\phi}$. We first minimise and then act on the two resulting equations with $\frac{\partial^{d-2}}{\partial x^{d-2}}$ and $\frac{\partial^{d-2}}{\partial y^{d-2}}$, respectively. In this way, we arrive at the pair of saddle point equations:

$$\frac{1}{4}\partial_x^2 \varrho_1(z,x) + \partial_z^2 \varrho_1(z,x) + P^2 \zeta_1(z)\delta(x) + \frac{1}{4}\partial_x^2 \varrho_2(z,x-\mu) +$$
$$\partial_z^2 \varrho_2(z,x-\mu) + P^2 \zeta_2(z)\delta(x-\mu) = 0, \tag{3.26a}$$

$$\frac{1}{4}\partial_y^2 \varrho_1(z,y) + \partial_z^2 \varrho_1(z,y) + P^2 \zeta_2(z)\delta(y) + \frac{1}{4}\partial_x^2 \varrho_1(z,y+\mu) +$$
$$\partial_z^2 \varrho_1(z,y+\mu) + P^2 \zeta_1(z)\delta(y+\mu) = 0. \tag{3.26b}$$

The pair of saddle point equations (3.26) has several expected properties. First, (3.26a) and (3.26b) are related through exchanging the labels $1 \leftrightarrow 2$ and replacing $x = y - \mu$. Second, if we define the eigenvalue density $\varrho(z,x) = \varrho_1(z,x) + \varrho_2(z,x-\mu)$, (3.26a) becomes the saddle point equation (3.10) derived without change of variables. Third, (3.26a) is the sum of a term which would yield the saddle point equation for $\mathcal{Q}_1$ alone, plus an extra term coming from $S_{\mathrm{int}}$ and involving $\varrho_2(z,x-\mu)$, in which the $\mu$-dependence is entirely contained (and likewise for (3.26b) and $\mathcal{Q}_2$).

The solution to (3.26) is again given by (3.12).

### 3.4.5  Free energy for two interacting linear quivers

Equipped with the eigenvalue densities $\varrho_1(z,x), \varrho_2(z,y)$ we can compute the free energy at leading order in $N$ and $P$ through

$$\mathcal{F} = (-1)^{\frac{d-3}{2}} \left( S_{\mathrm{eff},1} + S_{\mathrm{eff},2} + S_{\mathrm{int}} \right) \Big|_{\mathrm{on\ shell}}.$$

The terms $S_{\mathrm{eff},\alpha}$ simply contribute a factor $\pm \mathcal{F}_\alpha$, that is, the free energy for the linear quiver $\mathcal{Q}_\alpha$. The novel contributions are:

$$S_{\mathrm{int}}\Big|_{\mathrm{on\ shell}} = \left( S_0 + S_{\mathrm{der}} + S_{\mathrm{fund},1} + S_{\mathrm{fund},2} \right)\Big|_{\mathrm{on\ shell}}$$
$$=: (-1)^{\frac{d-3}{2}} \left[ \mathcal{F}_0 + \mathcal{F}_{\mathrm{der}} + \mathcal{F}_{\mathrm{fund},1} + \mathcal{F}_{\mathrm{fund},2} \right],$$

with the first equality due to (3.25). The fundamental hypermultiplets contribute:

$$
\frac{(-1)^{\frac{d-3}{2}}}{N^2 P^{(d-3)}} \mathcal{F}_{\text{fund},1} = \int_0^1 dz P^2 \zeta_2(z) dx \varrho_1(z,x) f_{\text{h}}(x-\mu)
$$

$$
= \int_0^1 dz \left( \pi^2 \sum_{k=1}^{\infty} \mathfrak{a}_{2,k} k^2 \sin(k\pi z) \right) \int dx \left( \pi \sum_{\ell=1}^{\infty} \mathfrak{a}_{1,\ell} \ell \sin(\ell\pi z) e^{-2\pi\ell|x|} \right) f_{\text{h}}(x-\mu)
$$

$$
= \frac{\pi^3}{2} \sum_{k=1}^{\infty} \mathfrak{a}_{1,k} \mathfrak{a}_{2,k} k^3 \mathcal{I}_k(\mu),
$$

with the integral $\mathcal{I}_k(\mu)$ defined in (3.19). The contribution from $\mathcal{F}_{\text{fund},2}$ is identical, due to the symmetries $1 \leftrightarrow 2$ and $\mu \leftrightarrow -\mu$ in the last line. Therefore

$$
(-1)^{\frac{d-3}{2}} \sum_{\alpha=1}^{2} \mathcal{F}_{\text{fund},\alpha} = N^2 P^{(d-3)} \pi^3 \sum_{k=1}^{\infty} \mathfrak{a}_{1,k} \mathfrak{a}_{2,k} k^3 \mathcal{I}_k(\mu).
$$

The next contribution is

$$
\frac{(-1)^{\frac{d-3}{2}}}{N^2 P^{(d-3)}} \mathcal{F}_0 = 2 \int_0^1 dz \int dx \varrho_1(z,x) \int dy \varrho_2(z,y) f_0(x-y-\mu)
$$

$$
= 2 \int_0^1 dz \int dx \left( \pi \sum_{k=1}^{\infty} \mathfrak{a}_{1,k} k \sin(k\pi z) e^{-2\pi k|x|} \right)
$$

$$
\times \int dy \left( \pi \sum_{\ell=1}^{\infty} \mathfrak{a}_{2,\ell} \ell \sin(\ell\pi z) e^{-2\pi\ell|y|} \right) f_0(x-y-\mu)
$$

$$
= \pi^2 \sum_{k=1}^{\infty} \mathfrak{a}_{1,k} \mathfrak{a}_{2,k} k^2 \hat{\mathcal{I}}_k^{(0)}(\mu),
$$

where in the last line we have defined

$$
\hat{\mathcal{I}}_k^{(0)}(\mu) := \int_{-\infty}^{+\infty} dx \int_{-\infty}^{+\infty} dy \, e^{-2\pi k(|x|+|y|)} f_0(x-y-\mu). \tag{3.27}
$$

Likewise, for the last contribution to the free energy we get

$$
\frac{(-1)^{\frac{d-3}{2}}}{N^2 P^{(d-3)}} \mathcal{F}_{\text{der}} = \frac{1}{2} \int_0^1 dz \int dx \int dy \left[ \varrho_1(z,x) \partial_z^2 \varrho_2(z,y) + \varrho_2(z,y) \partial_z^2 \varrho_1(z,x) \right] f_{\text{h}}(x-y-\mu)
$$

$$
= \frac{\pi^2}{2} \int_0^1 dz \int dx \int dy \sum_{k=1}^{\infty} \mathfrak{a}_{1,k} k \sin(k\pi z) e^{-2\pi k|x|}
$$

$$
\times \sum_{\ell=1}^{\infty} \mathfrak{a}_{2,\ell} \ell \sin(\ell\pi z) e^{-2\pi\ell|y|} \left[ -\pi^2\ell^2 - \pi^2 k^2 \right] f_{\text{h}}(x-y-\mu)
$$

$$
= -\frac{\pi^4}{2} \sum_{k=1}^{\infty} \mathfrak{a}_{1,k} \mathfrak{a}_{2,k} k^4 \hat{\mathcal{I}}_k^{(\text{der})}(\mu),
$$

with

$$
\hat{\mathcal{I}}_k^{(\text{der})}(\mu) := \int_{-\infty}^{+\infty} dx \int_{-\infty}^{+\infty} dy \, e^{-2\pi k(|x|+|y|)} f_{\text{h}}(x-y-\mu). \tag{3.28}
$$

The final answer for the free energy is obtained solving the integrals (3.19), (3.27) and (3.28), whose explicit form depends on the dimension $d$, through the functions (3.8).

### 3.4.6 Free energy for two interacting linear quivers in 5d

$\mathcal{I}_k(\mu)$ in 5d has already been computed in (3.20):

$$\mathcal{I}_k(\mu)|_{d=5} = -\frac{27}{8\pi^3 k^4}\left(e^{-2\pi k|\mu|} + 2\pi k|\mu| + \frac{1}{6}(2\pi k|\mu|)^3\right).$$

Using (3.8) at $d=5$, the integrals $\hat{\mathcal{I}}_k^{(0)}(\mu), \hat{\mathcal{I}}_k^{(\mathrm{der})}(\mu)$ are evaluated as:

$$\begin{aligned}
\hat{\mathcal{I}}_k^{(0)}(\mu) &= -\frac{27}{8}\pi\int_{-\infty}^{+\infty}\mathrm{d}x\int_{-\infty}^{+\infty}\mathrm{d}y\; e^{-2\pi k(|x|+|y|)}|x-y-\mu|\\
&= -\frac{27}{32\pi^2 k^3}\left[3e^{-2\pi k|\mu|} + (2\pi k|\mu|)\left(2 + e^{-2\pi k|\mu|}\right)\right],
\end{aligned}$$

and

$$\begin{aligned}
\hat{\mathcal{I}}_k^{(\mathrm{der})}(\mu) &= -\frac{9}{2}\pi\int_{-\infty}^{+\infty}\mathrm{d}x\int_{-\infty}^{+\infty}\mathrm{d}y\; e^{-2\pi k(|x|+|y|)}|x-y-\mu|^3\\
&= -\frac{27}{16\pi^4 k^5}\left[5e^{-2\pi k|\mu|} + (2\pi k|\mu|)\left(4 + e^{-2\pi k|\mu|}\right) + \frac{1}{3}\cdot(2\pi k|\mu|)^3\right].
\end{aligned}$$

Plugging these expressions in the formula for $\mathcal{F}$, we observe from direct computation that they satisfy

$$\mathcal{F}_{\mathrm{fund},1} + \mathcal{F}_{\mathrm{fund},2} + \mathcal{F}_0 + \mathcal{F}_{\mathrm{der}} = -\frac{1}{2}\left[\mathcal{F}_{\mathrm{fund},1} + \mathcal{F}_{\mathrm{fund},2}\right].$$

The latter is indeed a generic property, proved relying on the saddle point condition, which was utilised in (3.15). We thus compute

$$\begin{aligned}
\left[\mathcal{F}_{\mathrm{fund},1} + \mathcal{F}_{\mathrm{fund},2}\right]\Big|_{d=5} &= -\pi^3 N^2 P^2 \sum_{k=1}^{\infty}\mathfrak{a}_{1,k}\mathfrak{a}_{2,k}k^3\mathcal{I}_k(\mu)\bigg|_{d=5}\\
&= \frac{27}{8}N^2 P^2\sum_{k=1}^{\infty}\frac{1}{k}\mathfrak{a}_{1,k}\mathfrak{a}_{2,k}\left(e^{-2\pi k|\mu|} + 2\pi k|\mu| + \frac{4}{3}(\pi k|\mu|)^3\right).
\end{aligned}$$

The final expression for the free energy is

$$\mathcal{F} = \mathcal{F}_1 + \mathcal{F}_2 + \frac{27}{16}N^2 P^2\sum_{k=1}^{\infty}\frac{1}{k}\mathfrak{a}_{1,k}\mathfrak{a}_{2,k}\left(e^{-2\pi k|\mu|} + 2\pi k|\mu| + \frac{4}{3}(\pi k|\mu|)^3\right), \tag{3.29}$$

in agreement with (3.21). At $\mu \to 0$ we recover the UV free energy, whereas at $|\mu| \to \infty$ we find the factorized expression $\mathcal{F}_1 + \mathcal{F}_2$ for the free energy of a pair of quivers, plus the (divergent) contribution of massive decoupled fields, that is removed by local, supersymmetric counterterms in the usual way. We discuss the IR behaviour of $\mathcal{F}$ in more detail in Subsections 3.5.3-3.6.

### 3.4.7 Free energy for two interacting linear quivers in 3d

In 3d, we have

$$\mathcal{I}_k(\mu)|_{d=3} = \frac{1}{2\pi k^2}\left(e^{-2\pi k|\mu|} + 2\pi k|\mu|\right),$$

and

$$\hat{\mathcal{I}}_k^{(0)}(\mu)\Big|_{d=3} = \int_{-\infty}^{+\infty} dx \ e^{-2\pi k|x|} \int_{-\infty}^{+\infty} dy \ e^{-2\pi k|y|} \frac{\pi}{4}\delta(x-y-\mu)$$
$$= \frac{\pi}{4} \int_{-\infty}^{+\infty} dx \ e^{-2\pi k(|x|+|x-\mu|)}$$
$$= \frac{e^{-2\pi k|\mu|}}{8k}(1+2\pi k|\mu|) \ .$$

The last integral we need is

$$\hat{\mathcal{I}}_k^{(der)}(\mu)\Big|_{d=3} = \int_{-\infty}^{+\infty} dx \int_{-\infty}^{+\infty} dy \ e^{-2\pi k(|x|+|y|)}\pi|x-y-\mu|$$
$$= -\frac{2}{27}\hat{\mathcal{I}}_k^{(0)}(\mu)\Big|_{d=5}$$
$$= \frac{1}{8\pi^3 k^3}\left[3e^{-2\pi k|\mu|} + (2\pi k|\mu|)\left(2+e^{-2\pi k|\mu|}\right)\right] \ .$$

Therefore, in 3d we have the contributions

$$\sum_{\alpha=1}^{2} \mathcal{F}_{\text{fund},\alpha} = \frac{1}{2}N^2\pi^2 \sum_{k=1}^{\infty} \mathfrak{a}_{1,k}\mathfrak{a}_{2,k}k\left(e^{-2\pi k|\mu|} + 2\pi k|\mu|\right) ,$$

$$\mathcal{F}_0 = \frac{1}{8}N^2\pi^2 \sum_{k=1}^{\infty} \mathfrak{a}_{1,k}\mathfrak{a}_{2,k}k\left(e^{-2\pi k|\mu|} + e^{-2\pi k|\mu|}2\pi k|\mu|\right) ,$$

$$\mathcal{F}_{\text{der}} = -\frac{1}{8}N^2\pi^2 \sum_{k=1}^{\infty} \mathfrak{a}_{1,k}\mathfrak{a}_{2,k}k\left[e^{-2\pi k|\mu|}(3 + 2\pi k|\mu|) + 2(2\pi k|\mu|)\right] .$$

Summing all of them together we arrive at

$$\mathcal{F} = \mathcal{F}_1 + \mathcal{F}_2 + \frac{\pi^2}{4}N^2 \sum_{k=1}^{\infty} \mathfrak{a}_{1,k}\mathfrak{a}_{2,k}k\left[e^{-2\pi k|\mu|} + 2\pi k|\mu|\right] . \tag{3.30}$$

This expression agrees with the derivation *without* change of variables, eq. (3.22). Again, at $|\mu|$ we isolate the contribution $\mathcal{F}_1 + \mathcal{F}_2$ of a pair of decoupled linear quivers, plus the contribution of massive, decoupled free fields. The latter is scheme-dependent and is cancelled by a supersymmetric counterterm [99], see Subsection 3.6.

### 3.5 Holographic match

Let us discuss the match of the free energy, computed in QFT, with the holographic central charge computed in supergravity. Clearly, the free energy (3.30) does not agree with the holographic central charge (2.31). But this was to be expected, as we now explain.

The holographic flows we considered in Subsection 2.5 describe flows between two SCFTs: a UV CFT at $|\mu| = 0$ and an IR CFT at $|\mu| = \infty$,[13] see Figure 8. Here, UV and IR refer to the parameter $\mu$, not to a gauge coupling. More precisely, $\mu$ appears in the coefficient of a relevant operator in the long quiver. By conformal invariance, the radius $r$ of $\mathbb{S}^d$ does not enter the SCFT computation, but enters in the combination $r\mu$ when $\mu \neq 0$, effectively introducing a Wilsonian cutoff $\propto 1/r$. We have made the customary redefinition $r\mu \to \mu$ in the matrix model, with $\mu$

---

[13]For balanced quivers we find that the endpoints $\mu = \pm\infty$ of the RG flow coincide.

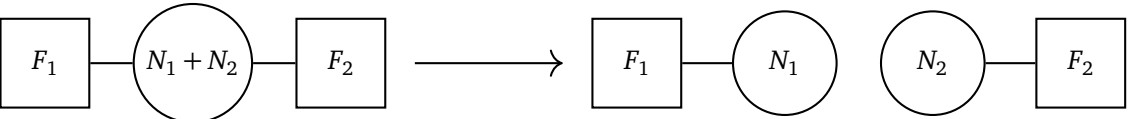

Figure 9: The mass deformation breaks the flavour group (square node) explicitly, and modifies the Coulomb branch geometry, which eventually splits in two cones (cf. Subsection 3.1.2). With our highly non-generic choice of deformation, this induces a Higgsing of the gauge group (circular node) $U(N_1 + N_2) \rightarrow U(N_1) \times U(N_2)$. The process, indicated by the arrow, is explained in Subsection 3.4.3.

being dimensionless. Moving $\mu$ in the range $0 \leq \mu \leq \infty$ describes the RG flow between two SCFTs, with UV and IR referring to the Wilsonian approach just mentioned.

The QFT picture, however, is slightly richer than the flow just sketched, and this fact is captured by $\mathcal{F}$, as we now proceed to explain.

The free energy and holographic central charge agree at the UV fixed point $\mu = 0$, see Subsection 3.3.4. However, for $0 < |\mu| < \infty$, the free energy computes a deformation of the UV SCFT, which includes more fields than the IR SCFT. This comes about because the breaking of gauge and flavour groups, represented in Figure 9, has a twofold effect:

($i$) to reduce the number of hypermultiplet modes at each node from $(N_{1,j} + N_{2,j})(F_{1,j} + F_{2,j})$ to $N_{1,j}F_{1,j} + N_{2,j}F_{2,j}$;

($ii$) to lift the Higgsed W-bosons.

In addition, for a linear quiver,

($iii$) the Higgsing procedure also lifts the bifundamental hypermultiplets of $U(N_{1,j}) \times U(N_{2,j\pm1})$.

The fields involved in these three effects are massive and decouple in the $|\mu| \rightarrow \infty$ limit (that is, at the very end of the RG flow, they become free fields at infinite distance from the interacting IR CFT). Therefore, the correct statement is that, when the mass deformation of interest is turned on, the UV SCFT flows to the pair of factorised SCFTs, stacked with decoupled heavy fields. Schematically:

$$\text{UV:} \qquad\qquad\qquad\qquad\qquad \text{IR:}$$
$$\mathcal{Q} \quad \xrightarrow{\;|\mu|\rightarrow\infty\;} \quad \mathcal{Q}_1 \sqcup \mathcal{Q}_2 \sqcup \text{decoupled}.$$

In conclusion, the holographic central charge is computed in supergravity and only captures the features of the interacting SCFT at the end of the RG flow. In contrast, the free energy is computed from a UV perspective. In order to compare these two quantities, we must subtract the contributions elucidated in the points ($i$) to ($iii$).

Let $\mathcal{F}_{\text{dec.}}$ denote the contributions of fields that decouple at the end of the RG flow, properly defined in Appendix C. Enforcing the subtraction, hence discarding $\mathcal{F}_{\text{dec.}}$, we define the *effective free energy*

$$\mathcal{F}_{\text{EFT}} = \mathcal{F} - \mathcal{F}_{\text{dec.}}. \tag{3.31}$$

Intriguingly, we find that the effective free energy matches with the holographic central charge, possibly *up to a local counterterm* involving only the background vector multiplets. Furthermore, (3.31) interpolates precisely between its UV value and the free energy of two decoupled quivers $\mathcal{Q}_1, \mathcal{Q}_2$. Namely:

$$\lim_{|\mu|\rightarrow 0} \mathcal{F}_{\text{EFT}} = \mathcal{F}[\mathcal{Q}], \qquad\qquad \lim_{|\mu|\rightarrow\infty} \mathcal{F}_{\text{EFT}} = \mathcal{F}[\mathcal{Q}_1] + \mathcal{F}[\mathcal{Q}_2],$$

which is the precise QFT analogue of the behaviour (2.32)-(2.33) of the holographic central charge.

In summary, along the flow parametrised by $\mu$, we adopt an effective field theory (EFT) approach, in which we separate the contributions of the fields that would decouple in the IR limit $\mu \to \infty$. This EFT method, embodied in the definition (3.31) and in the computations in Appendix C, leads to the perfect holographic match all along the RG flow, up to a background Chern–Simons counterterm in 5d. We elaborate on this last point in Subsection 3.5.3.

As a byproduct of our analysis, we show in Subsection 3.6 that the quantity (3.31) satisfies a strong version of the F-theorem, both in 3d and 5d.

### 3.5.1 Matching the effective free energy in 5d

The computation of the contribution $\mathcal{F}_{\text{dec.}}$ from the massive fields outlined in points $(i), (ii), (iii)$ is performed in Appendix C.

In 5d, the analysis of the regions of the 5d Coulomb branch onto which fundamental hyper-multiplets and W-bosons acquire a large mass and the resolution of the ensuing integral leads to expression (C.9). The effective free energy (3.31) is then readily evaluated by subtracting (C.9) from the 5d $\mathcal{F}$ in eq. (3.29). The result is:

$$\mathcal{F}_{\text{EFT}}\Big|_{d=5} = \mathcal{F}_1 + \mathcal{F}_2 + \frac{27}{16}N^2 P^2 \sum_{k=1}^{\infty} \mathfrak{a}_{1,k}\mathfrak{a}_{2,k}\frac{1}{k}e^{-2\pi k|\mu|}\left(1 + 2\pi k|\mu| + \frac{4}{3}(\pi k|\mu|)^3\right). \quad (3.32)$$

This answer *almost* agrees with the holographic central charge (2.30), except for the extra cubic dependence on $\mu$. This term is an effective Chern–Simons coupling for the background vector multiplet, and can be removed by adding a local counterterm only involving background fields. Similar background Chern–Simons terms allow the match of the rank-$N$ $E_n$ theories with its holographic dual [36]. A more thorough discussion is deferred to Subsection 3.5.3.

The actual relation between (3.32) and (2.30) is

$$d = 5 \; : \qquad \mathcal{F}_{\text{EFT}} = \frac{27\pi^6}{4}c_{hol},$$

up to the background Chern–Simons counterterm. This is exactly the relation (3.23) found in the UV SCFT, and $\mathcal{F}_{\text{EFT}}$ satisfies it for all $\mu$.

### 3.5.2 Matching the effective free energy in 3d

In 3d, $\mathcal{F}_{\text{dec.}}$ is given in (C.10). Combining it with (3.30), we obtain

$$\mathcal{F}_{\text{EFT}}\Big|_{d=3} = \mathcal{F}_1 + \mathcal{F}_2 + \frac{\pi^2}{4}N^2 \sum_{k=1}^{\infty} \mathfrak{a}_{1,k}\mathfrak{a}_{2,k}ke^{-2\pi k|\mu|}\left(1 + 2\pi k|\mu|\right). \quad (3.33)$$

In particular, it is manifest from (3.33) that $\mathcal{F}_{\text{EFT}}$ has a finite $|\mu| \to \infty$ limit, describing a pair of factorised linear quivers. Moreover, recalling the identification of parameters $\sigma_0 = 2P\mu$ and $N\mathfrak{a}_k = R_k$, one can compare with (2.31) and find

$$d = 3 \; : \qquad \mathcal{F}_{\text{EFT}} = 4\pi c_{hol},$$

which is exactly the relation (3.24) found in the UV SCFT. No counterterm is needed in 3d $\mathcal{N} = 4$, in perfect agreement with the lack of admissible such background couplings that preserve the amount of supersymmetry.

### 3.5.3 Chern–Simons terms for background vector multiplets

The main lesson of the present subsection is that, to match with the holographic central charge, we need to adopt an EFT approach and define an effective free energy (3.31). This is the free energy to which we subtract all the fields that will eventually decouple at $|\mu| \to \infty$, which one expects not to be captured by the holographic central charge.

The outcome of our analysis is a perfect match in 3d $\mathcal{N} = 4$, but a mismatching term, cubic in the background scalar $\mu$, appears in 5d $\mathcal{N} = 1$. The term in $\mathcal{F}$ that leads to a mismatch in $\mathcal{F}_{\text{EFT}}$ is

$$\delta \mathcal{F} = \frac{9}{4} \pi^3 \mu^3 N^2 P^2 \sum_{k=1}^{\infty} k^2 \mathfrak{a}_{1,k} \mathfrak{a}_{2,k} \tag{3.34a}$$

$$= \frac{\pi}{3} m^3 \sum_{j=1}^{P-1} \frac{1}{2} \mathcal{R}_1(j) F_{2,j} . \tag{3.34b}$$

To pass to the second line, we have used $N \mathfrak{a}_{\alpha,k} = R_{\alpha,k}$, $m^3 = (3P\mu)^3$ and the identities

$$2P \sum_{j=1}^{P-1} \mathcal{R}_1(j) F_{2,j} = 2P \int_0^P d\eta \, \mathcal{R}_1(\eta) \mathcal{R}_2''(\eta)$$

$$= \frac{2}{P} \int_0^P d\eta \sum_{k=1}^{\infty} R_{1,k} \sin\left(\frac{\pi k \eta}{P}\right) \sum_{\ell=1}^{\infty} \pi^2 \ell^2 R_{2,k} \sin\left(\frac{\pi \ell \eta}{P}\right)$$

$$= \pi^2 \sum_{k=1}^{\infty} k^2 R_{1,k} R_{2,k} .$$

At this point, we recall the basic feature of our setup: at every $j$, the corresponding scalar in the background vector multiplet is given a vev $M_j = (\underbrace{0, \ldots, 0}_{F_{1,j}}, \underbrace{m, \ldots, m}_{F_{2,j}})$. Therefore (3.34b) is

$$\delta \mathcal{F} = \sum_{j=1}^{P-1} \frac{\pi}{3} k_{f,j}^{\text{eff}} \, \text{Tr}[\text{diag}(\underbrace{0, \ldots, 0}_{F_{1,j}}, \underbrace{m, \ldots, m}_{F_{2,j}})^3], \tag{3.35}$$

with $k_{f,j}^{\text{eff}}$ read off from (3.34b). For each flavour group $U(F_j)$ it is

$$k_{f,j}^{\text{eff}} = \frac{1}{2} \mathcal{R}_1(j) \text{sgn}(\mu),$$

using the fact that relaxing the assumption $\mu > 0$ simply amounts to replace $\mu^3$ with $|\mu^3|$. Written in the form (3.35), it is manifest that $\delta \mathcal{F}$ can be cancelled by a local counterterm only involving background fields. The counterterm is a supersymmetric Chern–Simons term for the background vector multiplets of the flavour groups $U(F_j)$.

To see this, remember that a 5d supersymmetric Chern–Simons term with coupling $k$, involving a gauge field $A$ with curvature $F_A$, has the schematic form

$$S_{\text{CS}}|_{d=5} = \frac{k}{24\pi^2} \int_{\mathbb{S}^5} \text{Tr} A \wedge F_A \wedge F_A + \text{ SUSY completion} .$$

We are interested in the case in which $A$ is a background gauge field for the flavour symmetry, thus it is valued in $\mathfrak{u}(F_j)$. Let $M_j$ be the scalar field in the $\mathfrak{u}(F_j)$ background vector multiplet. Using localisation, such Chern–Simons term reduces to [111, 112]

$$\mathcal{F}_{\text{CS},f,j} = \frac{k_{f,j}}{24\pi^2} \text{Tr} \left(2\pi M_j\right)^3 = \frac{\pi}{3} k_{f,j} \text{Tr} \left(M_j^3\right),$$

where we use the subscript $f$ to emphasize that it involves the flavour symmetry group, not a gauge group. Specifying $M_j = (\underbrace{0,\ldots,0}_{F_{1,j}}, \underbrace{m,\ldots,m}_{F_{2,j}})$, we prove the claim that (3.35) can be cancelled by a local, supersymmetric Chern–Simons term for background vector multiplets. The effective Chern–Simons couplings needed are $k_{f,j} = -k_{f,j}^{\text{eff}}$. The conclusions depend on which of the two scenarios holds:

- $\mathcal{R}_1(j) = 0 \bmod 2 \ \forall j$, which means $k_{f,j}^{\text{eff}} \in \mathbb{Z} \ \forall j$;

- $\mathcal{R}_1(j) \neq 0 \bmod 2$, for which $k_{f,j}^{\text{eff}} \in \frac{1}{2} + \mathbb{Z}$.

When $\mathcal{R}_1(j) = 0 \bmod 2$, we can remove the $j^{\text{th}}$ summand in $\delta\mathcal{F}$, as per (3.34b), by introducing a Chern–Simons term for the background vector multiplet, with coefficient $k_{f,j} = -k_{f,j}^{\text{eff}} \in \mathbb{Z}$. If this can be done $\forall j = 1,\ldots,P-1$, in other words, when $\mathcal{R}_1(j) = 0 \bmod 2$ $\forall j$, $\mathcal{F}_{\text{EFT}}|_{d=5}$ matches with $c_{hol}|_{d=5}$ upon specifying a local, unitary, supersymmetric countert­erm only involving background fields, which moreover is invariant under large background gauge transformations. As explained in [121] for the similar case of 3d $\mathcal{N} = 2$ theories, we have the freedom to add such terms to the action. In fact, in 3d it is sometimes necessary to specify such counterterms in order to match the sphere partition functions of dual theo­ries [121]. The exact same argument holds here, except that our duality is AdS/CFT, instead of a duality of QFTs.

When $\mathcal{R}_1(j) \neq 0 \bmod 2$, however, $k_{f,j}^{\text{eff}}$ is *fractional*. We can remove its integer part by a Chern–Simons counterterm with integer level, as just described. Nevertheless, the fractional part $\kappa_{f^3}$, defined as

$$k_{f,j}^{\text{eff}} = \kappa_{f^3} + \mathbb{Z}, \qquad \kappa_{f^3} = \frac{1}{2},$$

is physical, which is to say, it cannot be removed preserving all the symmetries of the system (including conformal and supersymmetry) and also gauge invariance under background gauge transformations [121].

These fractional background Chern–Simons terms have first been observed and studied in 5d SCFTs in [122]. As explained therein, $\kappa_{f^3}$ equals the coefficient of a contact term in the flavour current three-point function. It is the analogue of the well-known mixed gauge-parity anomaly in odd dimensions [123,124], with the difference that the present anomaly only involves flavour symmetries.

We may introduce a Chern–Simons term for the background vector multiplet, with frac­tional coefficient $\kappa_{f^3}$. This counterterm cancels $\delta F$, as well as the contact terms in the flavour current three-point functions, at the expense of invariance under background gauge trans­formations. We choose to do so, and note that this regularisation agrees with [107]. A back­ground Chern–Simons term, similar to the one we add here, is needed to match holographically the rank-$N$ $E_n$ theories [36].

## 3.6 On the F-theorem

Let us say a few words about how our results support the F-theorem [98,99]. To begin, let us be precise with the statement of the conjecture. Usually, it is written as

$$\mathcal{F}|_{\text{UV}} > \mathcal{F}|_{\text{IR}}, \tag{3.36}$$

but this inequality must be supplemented by the condition that $\mathcal{F}$ is stationary at the fixed points. Still, (3.36) would be false if $\mathcal{F}$ is taken at its face value, namely from the matrix model in a theory which is not conformal. Elementary counterexamples are given in [99, App.] and in Appendix C.2.5.

Let us elaborate on this fact. On odd-dimensional spheres of radius $r$, the partition function has the form [99, 125]

$$(-1)^{(d-1)/2} \ln \mathcal{Z}_{\mathbb{S}^d} = \sum_{n=0}^{(d-1)/2} C_n r^{2n+1} + \mathcal{F}_\star \,,$$

for some coefficients $C_n$. The quantity $\mathcal{F}_\star$, independent of massive parameters, is free from ambiguities. The coefficients $C_n$ in the polynomial in odd powers of $r$, however, can be shifted by local gravitational counterterms, only involving the metric and the curvature.[14] For this reason, the quantity that must enter the F-theorem is $\mathcal{F}_\star$.

In the present context, the radius $r$, which we set to 1, enters $\mathcal{F}$ through the combination $\mu r$. Subtracting the ambiguous $\mu$-dependent pieces from (3.29) and (3.30) we show the validity of the F-theorem in $d = 5$ and $d = 3$, respectively, for the RG flows considered.

We will now argue for a stronger version of the F-theorem. We want to modify the matrix model free energy $\mathcal{F}$ in such a way that the result (i) decreases along an RG flow between two CFTs, (ii) interpolates between the UV and IR values of $\mathcal{F}_\star$, and (iii) is stationary at the CFT points. One main lesson of the present section is that decoupled fields must be carefully taken into account, because $\mathcal{F}$ will keep track of them. From the matrix model quantity $\mathcal{F}$ at arbitrary $\mu$, we subtract the contribution of the heavy fields, cf. the example in Appendix C.2.5. It turns out that this subtraction regulates $\mathcal{F}$ so that the remaining part is the universal constant $\mathcal{F}_\star$. It is worthwhile to mention that our technique is consistent with the reguralisation procedure for sphere partition functions of non-conformal theories discussed in [99].

The *effective* free energy thus obtained is an intrinsic observable of the IR interacting CFT, blind to the decoupled massive fields. This quantity satisfies the strong version of the F-theorem, as shown in Appendix D. In conclusion, considering $\mathcal{F}_{\text{EFT}}$ to quantitatively explore the RG flows is further motivated by the F-theorem.

## 3.7 Wilson loops in QFT

In this subsection, we compute the expectation value of a Wilson loop operator using the matrix model formalism discussed above and compare the result with the holographic computation of Subsection 2.6.

The vev of a $\frac{1}{2}$-BPS Wilson loop in a representation $R$ of the $j_*^{\text{th}}$ gauge node in a quiver gauge theory of the type in Figure 3 is given by

$$\left\langle \mathcal{W}_R^{(j_*)} \right\rangle = \frac{1}{\mathcal{Z}_{\mathbb{S}^d}} \int \mathrm{d}\vec{\sigma} \, e^{-S_{\text{eff}}} \frac{1}{\dim R} \mathrm{tr}_R \, e^{2\pi\sigma_{j_*}} \,.$$

Note that $j_*$ here corresponds precisely to $\eta_*$ of Subsection 2.6.

For all those representations such that the insertion of the loop operator yields a subleading contribution at large $N$, its expectation value is obtained by evaluating its classical expression on the saddle point solution. Before starting any computation, we observe that the large $N$ Wilson loop vev is linear in the eigenvalue density $\varrho(z, x)$, as opposed to $\mathcal{F}$, which is quadratic. This means that the Wilson loop vev in $\mathcal{Q}$ splits from the onset into a sum of Wilson loops in the quivers $\mathcal{Q}_\alpha$.

We are primarily interested in Wilson loops in the rank-$\ell$ antisymmetric representation of the gauge group, which satisfies the subleading assumption. These supersymmetric Wilson loops in long quivers were studied previously in [100, 101, 126].[15] To this end we introduce

---

[14]Another way to put it, is that $\mathcal{F}_\star$ is obtained in a regularisation scheme that cancels divergences and preserves conformal invariance. The non-conformal terms are scheme-dependent [122, 125].

[15]See [109, 127, 128] for the single-node case.

the parameter

$$\kappa = \frac{\ell}{N_{j_*}},\tag{3.37}$$

which is an effectively continuous parameter $0 \le \kappa < 1$ in the large $N$ limit. Following Uhlemann [100], we also need an auxiliary function $b(\kappa,z)$ such that

$$\kappa = \int_{b(\kappa,z)}^{\infty} \mathrm{d}x\, \varrho(z,x).\tag{3.38}$$

In other words, $b(\kappa,z)$ is a point in the support of $\varrho(z,\cdot)$ such that a fraction $\kappa$ of the eigenvalues lies on its right and $1-\kappa$ on its left.

The expectation value of the Wilson loop is then obtained by evaluating the integral

$$\ln\langle \mathcal{W}_{\wedge^\ell}\rangle = 2\pi(d-2)PN \int_0^\kappa \mathrm{d}\tilde{\kappa}\, b(\tilde{\kappa},z_*),$$

with $z_* = j_*/P$ on the right-hand side. In practice, it is convenient to invert the relation (3.38) and express $\kappa(b)$ as a function of a free parameter $b(z)$, with $z$ fixed from the beginning of the computation. Using $b(0,z) = \infty$, we have

$$\ln\langle \mathcal{W}_{\wedge}(b)\rangle = 2\pi(d-2)PN \int_{b(z)}^{\infty} \kappa'(b) b\,\mathrm{d}b.\tag{3.39}$$

Let us evaluate (3.38) on the solution (3.12), which gives us the function $\kappa(b)$:

$$\kappa(b) = \pi \sum_{\alpha=1}^{\mathscr{F}} \sum_{k=1}^{\infty} k\mathfrak{a}_{\alpha,k} \sin(\pi k z_*) \int_b^{\infty} e^{-2\pi k|x-\mu_\alpha|}\mathrm{d}x\tag{3.40a}$$

$$= \frac{1}{2} \sum_{\alpha=1}^{\mathscr{F}} \sum_{k=1}^{\infty} \mathfrak{a}_{\alpha,k} \sin(\pi k z_*)\, e^{-2\pi k|b-\mu_\alpha|}\tag{3.40b}$$

$$= \frac{1}{2\pi^2} \frac{P}{N} \sum_{\alpha=1}^{\mathscr{F}} \sum_{j=1}^{P-1} F_{\alpha,j} \mathrm{Re}\left[\mathrm{Li}_2\left(e^{-2\pi|b-\mu_\alpha|+i\pi(z_j-z_*)}\right) - \mathrm{Li}_2\left(e^{-2\pi|b-\mu_\alpha|+i\pi(z_j+z_*)}\right)\right],\tag{3.40c}$$

with $z_j = j/(P-1)$ and $\mathrm{Li}_2$ the polylogarithm of order 2. The equality between (3.40b) and (3.40c) stems from the dilogarithm identity $\mathrm{Li}_2(u) = \sum_{n=1}^{\infty} \frac{u^n}{n^2}$ and

$$\frac{P}{N} \sum_{j=1}^{P-1} F_{\alpha,j} \mathrm{Re}\left[\mathrm{Li}_2\left(e^{-2\pi|b-\mu_\alpha|+i\pi(z_j-z_*)}\right) - \mathrm{Li}_2\left(e^{-2\pi|b-\mu_\alpha|+i\pi(z_j+z_*)}\right)\right]$$

$$= \int_0^1 \mathrm{d}z P^2 \zeta_\alpha(z) \sum_{n=1}^{\infty} \frac{e^{-2\pi n|b-\mu_\alpha|}}{n^2}\left[\cos(\pi(z-z_*)) - \cos(\pi(z-z_*))\right]$$

$$= \pi^2 \int_0^1 \mathrm{d}z \left[\sum_{k=1}^{\infty} \mathfrak{a}_{\alpha,k} k^2 \sin(\pi k z)\right]\left[\sum_{n=1}^{\infty} \frac{e^{-2\pi n|b-\mu_\alpha|}}{n^2} 2\sin(\pi z_*)\sin(\pi z)\right]$$

$$= \pi^2 \sum_{k=1}^{\infty} \mathfrak{a}_{\alpha,k} \sin(k\pi z_*) e^{-2\pi k|b-\mu_\alpha|},$$

where in the second step we have used (3.2).

A word of caution is due about (3.40b), which requires a small technical detour. The reader interested in the final result can safely skip the next paragraph.

To get (3.40b) we have assumed that $b \geq \max_\alpha \mu_\alpha$. This seems a meaningless assumption if we are willing to explore the full RG flow. At the conformal point, this amounts to require $b \geq 0$. The converse regime $b < 0$ is recovered by replacing $\kappa \mapsto 1 - \kappa$ everywhere, which has the physical meaning of acting with charge conjugation on the Wilson loop. The corresponding term in (3.40b) would have $e^{-2\pi|b|}$ replaced by $-2 + e^{-2\pi|b|}$. More generally, the mass scales $\mu_\alpha$ partition the $b$-parameter space into chambers. Whenever $b$ crosses an interface separating the chamber $b > \mu_\alpha$ from $b < \mu_\alpha$, the corresponding integral in (3.40b) jumps $e^{-2\pi|b|} \mapsto -2 + e^{-2\pi|b|}$. However, formula (3.39) only depends on $\kappa'(b)$, thus the difference between the distinct chambers drops out of the computation of the Wilson loop vev. That is, the jumps across the interfaces affect $\kappa(b)$ but not the Wilson loop vev. With this caveat in mind, we can safely fix a chamber arbitrarily and evaluate $\kappa(b)$ as in (3.40b). It would be interesting to understand better if this mechanism of jumps across the interfaces has any physical implications, potentially related to how the screening of the loop operator by massive BPS particles varies with $\kappa$.

Back to the Wilson loop computation. Finally, plugging (3.40c) into (3.39) leads us to

$$
\begin{aligned}
\frac{\ln \langle \mathcal{W}_\wedge (b) \rangle}{(d-2)} = {} & \frac{P^2}{2\pi^2} \sum_{\alpha=1}^{\mathscr{F}} \sum_{j=1}^{P-1} F_{\alpha,j} \mathrm{Re} \Big[ \mathrm{Li}_3 \left( e^{-2\pi|b-\mu_\alpha|+i\pi(z_j-z_*)} \right) - \mathrm{Li}_3 \left( e^{-2\pi|b-\mu_\alpha|+i\pi(z_j+z_*)} \right) \\
& + 2\pi b \, \mathrm{sign}(b - \mu_\alpha) \left( \mathrm{Li}_2 \left( e^{-2\pi|b-\mu_\alpha|+i\pi(z_j-z_*)} \right) - \mathrm{Li}_2 \left( e^{-2\pi|b-\mu_\alpha|+i\pi(z_j+z_*)} \right) \right) \Big] .
\end{aligned}
\tag{3.41}
$$

Recall that $z_*$ is fixed by the choice of where to insert the Wilson loop along the quiver, and $P z_*$ equals $\eta_*$ of Subsection 2.6. The specialization of expression (3.41) to $\mathscr{F} = 2$, with $\mu_1 = 0, \mu_2 = \frac{\sigma_0}{2P}$ is in perfect agreement with the holographic computation (2.38). We regard the match of the holographic and QFT computation of the Wilson loop expectation value as strong evidence of the interpretation of the holographic setup with two ranks functions as a mass deformation in the dual CFT.

We close this section by returning to a puzzle that we encountered at the end of Subsection 2.6. In the supergravity analysis, it was unclear which representation of the gauge group the factorised Wilson loops transform in. The matrix model perspective automatically resolves this issue in the following way. The factorised form of the expression for the eigenvalue density $\varrho(z, x)$ in (3.11), suggests that we can take $\kappa$ in the left hand side of (3.38) to have a similar decomposition, namely that

$$
\kappa = \sum_{\alpha=1}^{\mathscr{F}} \kappa_\alpha ,
\tag{3.42}
$$

which in turn leads to

$$
\ell = \sum_{\alpha=1}^{\mathscr{F}} \ell_\alpha , \qquad \ell_\alpha = 0, 1, \ldots, N_{\alpha,j} - 1 ,
\tag{3.43}
$$

by virtue of (3.37). This is to be interpreted as follows. One starts with an insertion of a Wilson loop in the $\wedge^\ell$ representation of the $j^{\text{th}}$ gauge node. Next, we turn on the mass deformations, and zoom in on the point in the Coulomb branch where the gauge group is broken to a product of $\mathscr{F}$ factors. The original Wilson loop is now factorised into $\mathscr{F}$ daughter loops, each transforming in the $\wedge^{\ell_\alpha}$ of the $j^{\text{th}}$ gauge group of the $\alpha^{\text{th}}$ resulting quiver.

The Wilson loop vev splits into the sum of $\mathscr{F}$ terms, and we have $\mathscr{F}$ different ways to take the IR limit. Fixing an index $\beta$, we set $\mu_\beta = 0$ and send $|\mu_\alpha| \infty$ for all $\alpha \neq \beta$. This is equivalent to send $|\mu_\alpha - b| \infty$ for all $\alpha \neq \beta$ but keeping $|\mu_\beta - b| \infty$ fixed. All the contributions to (3.41) are exponentially suppressed except for the term $\alpha = \beta$ in the sum, leading to the Wilson loop vev corresponding to the quiver with rank function $\mathcal{R}_\beta$. This parallels the behaviour found in supergravity.

# 4 Conclusions

Let us start with a brief summary of the contents of this work.

In Part I, Section 2 of this paper we studied the holographic set-up. The duals to 5d $\mathcal{N} = 1$ SCFTs contain a geometry of the form $AdS_6 \times \mathbb{S}^2 \times \Sigma$, with warp factors that only depend on the coordinates $(\sigma, \eta)$ on $\Sigma$. The Ramond and Neveu–Schwarz fields respect these isometries, which in turn are identified with the $SO(2,5) \times SU(2)$ bosonic part of the superconformal group of 5d $\mathcal{N} = 1$ SCFTs. A very analogous picture is displayed by the $AdS_4 \times \mathbb{S}_1^2 \times \mathbb{S}_2^2 \times \Sigma$ backgrounds dual to $\mathcal{N} = 4$ SCFTs in 3d.

The problem of finding the warp factors and other functions in the infinite families of backgrounds is reduced to a Laplace equation for a suitably defined potential function, together with adequate boundary conditions. This is the *same* Laplace equation in both 5d and 3d cases. Charges associated to NS5 branes, colour and flavour branes are calculated. They can be put in correspondence with Hanany–Witten setups associated with the field theories.

The picture provided by the Laplace problem, namely a density of charge in between conducting planes, see Figure 4, suggests various generalisations. The one studied in this paper is that for which the charge density, corresponding to a gauge rank function in the field theory, is split into two new rank functions. This new electrostatic problem was studied, the free energy (referred here as holographic central charge) and Wilson loops in antisymmetric representations were calculated using the holographic perspective.

In Part II, Section 3 of this paper, we discussed the matrix models associated with the field theory duals to the AdS backgrounds of Part I. The free energy of the matrix model coincides, after a careful regularisation procedure, with the holographic result of Part I. The matrix model calculation of the antisymmetric Wilson loop expectation value agrees with the holographic calculation as well. We also showed that the holographic Laplace problem is reproduced in the matrix model, the saddle point equation being the second derivative of the Poisson equation for the same electrostatic problem. Full details of every calculation are provided, either in the main body of the paper or in dedicated appendices. These results put the holographic correspondence proposed in this work on a firm basis.

The matrix model study of the situation leading to the modified holographic Laplace equation gives a clean picture of its field theory interpretation. Namely, a mass deformation for a large number of matter fields. This leads to many linear quivers with interactions between them. A pictorial representation of these RG flows, controlled by a single mass parameter, was given in the introduction in Figures 1 and 2.

One advantage of our rank function-based formalism is that the dependence on the specific gauge theory realisation of a CFT is eventually washed away from the supersymmetric physical observables. The final answer is invariant under field theory dualities. Along the way, we introduced a large $N$ limit distinct from the 't Hooft limit, which preserves the properties of the geometric engineering of the 5d theories. That is, in $d = 5$, we do not scale the inverse gauge coupling with $N$, as in the usual 't Hooft large $N$ limit, but rather treat it in the same way as the masses and Coulomb branch parameters. Given that 5d field theory dualities and their M-theory realisations often exchange masses and inverse gauge couplings, our way of taking the large $N$ limit is the most natural in this context.

One perhaps unsatisfactory aspect is that, to match with the computation in supergravity away from the superconformal points, we had to adopt an effective field theory approach. This was motivated both from the point of view of holography and of the F-theorem. This step, however, relied on a technical examination of the integration domain for the fields that become heavy and eventually decouple. A derivation more robust against lack of information on the UV theory would be desirable, and we leave it as an open problem.

The sphere free energy in odd dimensions is a candidate $c$-function [98, 99]. That is, it is

expected to be stationary at the RG fixed points, with its value in the IR lower than its value in the UV. One important open question for 5d SCFTs, is a field theoretic derivation of the 5d F-theorem, see [109, 122, 129] for evidence in that direction. We have evaluated the sphere free energy for a wide class of 5d SCFTs, in the long quiver limit. One important outcome of our analysis, which will be useful for future investigations, is that the 5d F-theorem must be stated and tested using the effective free energy we introduced.

The findings of this paper suggest several generalisations and avenues for future research. One immediate application would be to study similar mass deformations from the holographic perspective in the AdS$_5$ and AdS$_3$ backgrounds with eight supercharges, where the techniques in Section 2.5 carry over.

Moreover, it would be very interesting to construct reliable backgrounds that can follow the QFT along the RG flow, from vanishing masses $\mu \to 0$ to very large masses. Such backgrounds should contain two 'throats' consisting of AdS$_d$ and spheres to suitably realise the R-symmetry. Away from the throats, we expect a factor $\mathbf{R}^{1,d-2}$ and a nonlinear combination of the $(r, \eta, \sigma)$-directions. It seems natural to speculate with backgrounds of this type, as one should be able to zoom-in different regions, known to be reliable. Probably this would look like a generalisation of the many stack-brane solutions, out of the near horizon. This more involved family of backgrounds should be non-singular and provide expressions for the holographic central charge analogous to (2.30)-(2.31).

Another interesting application of our formalism is to explore different setups for the electrostatic problem. Indeed, it not only gives an immediate way of identifying holographically dual theories, but it also provides an intuitive description of the SCFT. For example, by imposing periodic boundary conditions one might be able to describe circular linear quivers, while more sophisticated charge distributions may give rise to star-shaped quivers. Another open problem is the construction of a holographic dual of unbalanced quivers. It would be interesting to explore our holographic proposal in these scenarios.

The last open question we want to mention is the study of vortex loops in the 3d theories, in the Veneziano limit. Mirror symmetry maps a Wilson loop to a vortex loop [130] and the results of Subsection 3.7 will serve as a benchmark for such a prospect study. In this paper we have considered mass deformations, without FI parameters, leaving the study of the mirror quivers with only FI deformations turned on for future work. Moreover, the localisation of the ellipsoid partition function for 3d gauge theories with vortex loop insertions was recently considered in [131], which could be relevant to pursue this direction.

### Acknowledgments:

The contents and presentation of this work much benefited from extensive discussion with various colleagues. We would like to specially thank: Lorenzo Coccia, Diego Correa, Stefano Cremonesi, Wei Cui, Michele Del Zotto, Diego Rodriguez-Gomez, Mauricio Romo, Masazumi Honda, Yolanda Lozano, Ali Mollabashi, Matteo Sacchi, Guillermo Silva, Daniel Thompson, Miguel Tierz, Alessandro Tomasiello, Christoph Uhlemann, Yifan Wang.

**Funding information**   AL and CN are supported by STFC grant ST/T000813/1. LSch is supported by The Royal Society through grant RGF\R1\180087 Generalised Dualities, Resurgence and Integrability. AL has also received funding from the European Research Council (ERC) under the European Union's Horizon 2020 research and innovation programme (grant agreement No 804305). The authors have applied to a Creative Commons Attribution (CC BY) licence.

# A Potentials, harmonic and special functions

In this appendix we give a concrete formula for the potential

$$\hat{W}(\sigma,\eta) = \sum_{k=1}^{\infty} a_k \sin\left(\frac{k\pi\eta}{P}\right) e^{-\frac{k\pi|\sigma|}{P}}, \qquad a_k = \frac{P}{2k\pi} R_k,$$

where the Fourier coefficients $R_k$ were calculated in (2.18) and take the form

$$R_k = \frac{2}{k\pi}\left(N_0 + (-1)^{k+1} N_P\right) + \frac{2P}{\pi^2 k^2} \sum_{j=1}^{P-1} F_j \sin\left(\frac{k\pi j}{P}\right).$$

To efficiently write the potential, let us introduce some machinery. Define the following functions $G_s^c, G_s^s$ of two variables for positive integers $s$:

$$G_s^c(\sigma,\eta) = \sum_{k=1}^{\infty} \frac{1}{k^s} \cos(\pi k\eta) e^{-\pi k\sigma},$$

$$G_s^s(\sigma,\eta) = \sum_{k=1}^{\infty} \frac{1}{k^s} \sin(\pi k\eta) e^{-\pi k\sigma}.$$

We can rewrite these in terms of polylogarithms:

$$
\begin{aligned}
G_s^c(\sigma,\eta) &= \operatorname{Re} \operatorname{Li}_s\left(e^{\pi(-\sigma+i\eta)}\right), \\
G_s^s(\sigma,\eta) &= \operatorname{Im} \operatorname{Li}_s\left(e^{\pi(-\sigma+i\eta)}\right).
\end{aligned}
\tag{A.1}
$$

They satisfy the following relations:

$$
\begin{aligned}
\partial_\eta G_s^c(\sigma,\eta) &= -\pi G_{s-1}^s(\sigma,\eta), \\
\partial_\eta G_s^s(\sigma,\eta) &= \pi G_{s-1}^c(\sigma,\eta), \\
\partial_\sigma G_s^c(\sigma,\eta) &= -\pi G_{s-1}^c(\sigma,\eta), \\
\partial_\sigma G_s^s(\sigma,\eta) &= -\pi G_{s-1}^s(\sigma,\eta).
\end{aligned}
\tag{A.2}
$$

As a consequence, $G_s^c$ and $G_s^c$ are harmonic functions, i.e. they solve the Laplace equation

$$\left(\partial_\eta^2 + \partial_\sigma^2\right) G_s^{c,s} = 0 \qquad \forall s \in \mathbb{Z}_{>0}.$$

By construction, $G_s^c$ and $G_s^c$ are, respectively, even and odd periodic function of $\eta$. Moreover, using simple trigonometric identities, we can compute the sum

$$\sum_{k=1}^{\infty} \frac{1}{k^s} \sin(\pi k n) \sin(\pi k\ell) e^{-\frac{k\pi\sigma}{P}} = \frac{1}{2}\left[G_s^c\left(\frac{\sigma}{P}, n-\ell\right) - G_s^c\left(\frac{\sigma}{P}, n+\ell\right)\right].
\tag{A.3}$$

We can now use eq. (A.3) to write down the potential

$$
\begin{aligned}
\hat{W}(\sigma,\eta) = \frac{P}{\pi^2}&\left[N_0 G_2^s\left(\frac{|\sigma|}{P}, \frac{\eta}{P}\right) - N_P G_2^s\left(\frac{|\sigma|}{P}, \frac{\eta+P}{P}\right)\right] \\
+ \frac{P^2}{2\pi^3}&\sum_{j=1}^{P-1} F_j\left[G_3^c\left(\frac{|\sigma|}{P}, \frac{\eta-j}{P}\right) - G_3^c\left(\frac{|\sigma|}{P}, \frac{\eta+j}{P}\right)\right].
\end{aligned}
\tag{A.4}
$$

By using (A.2) and the boundary condition

$$\lim_{\epsilon\to 0}\left(\partial_\sigma \hat{W}(\sigma=+\epsilon,\eta) - \partial_\sigma \hat{W}(\sigma=-\epsilon,\eta)\right) = -\mathcal{R}(\eta),$$

we can also write the rank function in this formalism, namely

$$\mathcal{R}(\eta) = \frac{2}{\pi}\left[N_0 G_1^s\left(0, \frac{\eta}{P}\right) - N_P G_1^s\left(0, \frac{\eta + P}{P}\right)\right]$$
$$+ \frac{P^2}{2\pi^3}\sum_{j=1}^{P-1} F_j\left[G_2^c\left(0, \frac{\eta - j}{P}\right) - G_2^c\left(0, \frac{\eta + j}{P}\right)\right].$$

That this function is in fact piecewise linear can be see from the fact that

$$G_1^s(0, \eta) = \operatorname{Im}\operatorname{Li}_1(e^{i\pi\eta}) = -\operatorname{Im}\log(1 - e^{i\pi\eta}) = \frac{\pi}{2}(1 - \eta), \quad 0 < \eta < 2. \tag{A.5}$$

In addition, $\forall 0 < \alpha < 1$ we have

$$\frac{1}{\pi^2}\left[G_2^c(0, \eta - \alpha) - G_2^c(0, \eta + \alpha)\right] = \begin{cases} (1 - \alpha)\eta, & 0 < \eta < \alpha, \\ (1 - \eta)\alpha, & \alpha < \eta < 2 - \alpha, \\ (1 - \alpha)(\eta - 2), & 2 - \alpha < \eta < 2, \end{cases}$$

because

$$G_2^c(0, \eta) = \left(\frac{\pi}{2}\right)^2(\eta - 1)^2 - \frac{\pi^2}{12}, \qquad 0 < \eta < 2.$$

To see this latter statement, $G_2^c(0, \eta)$ can be fixed by using $\partial_\eta G_2^c(0, \eta) = -\pi G_1^s(0, \eta)$ and (A.5). The integration constant is fixed by demanding that $G_2^c(0, 0) = \operatorname{Li}_2(1) = \zeta(2) = \frac{\pi^2}{6}$.

With the identifications worked out in Subsection 3.2.7, the unnormalised eigenvalue density $\hat{\varrho}(z, x)$ in this formalism reads

$$\hat{\varrho}(z, x) = 2\left[N_P G_2^s\left(\frac{|\sigma|}{P}, \frac{\eta}{P} + 1\right) - N_0 G_2^s\left(\frac{|\sigma|}{P}, \frac{\eta}{P}\right)\right]$$
$$+ \frac{P}{\pi^2}\sum_{j=1}^{P-1} F_j\left[G_1^c\left(\frac{|\sigma|}{P}, \frac{\eta + j}{P}\right) - G_1^c\left(\frac{|\sigma|}{P}, \frac{\eta - j}{P}\right)\right].$$

As a consistency check, with the substitutions (3.13), $\varrho(z, x)$ it is expressed as

$$\varrho(z, x) = 2\left[\frac{N_P}{N}G_2^s(2x, z + 1) - \frac{N_0}{N}G_2^s(2x, z)\right]$$
$$+ \frac{P^2}{\pi^2}\int_0^1 d\tilde{z}\,\zeta(\tilde{z})\left[G_1^c(2x, z + \tilde{z}) - G_1^c(2x, z - \tilde{z})\right].$$

Plugging (3.2) in the second line and integrating over $\tilde{z}$, the last formula reproduces (3.12).

It also possible to cast the holographic central charge in this language. Namely eq. (2.21) can be rewritten as

$$c_{hol}\big|_{d=5} = \frac{P^2}{4\pi^8}(2N_0^2 + 2N_P^2 + 3N_0 N_P)\zeta(3)$$
$$+ \frac{P^3}{\pi^9}\sum_{j=1}^{P-1} F_j\left[N_0 G_4^c\left(0, \frac{j\pi}{P}\right) + N_P G_4^s\left(0, \frac{(P - j)\pi}{P}\right)\right]$$
$$- \frac{P^4}{4\pi^{10}}\sum_{j=1}^{P-1}\sum_{k=1}^{P-1} F_l F_k\left[G_5^c\left(0, \frac{\pi(k + j)}{P}\right) - G_5^c\left(0, \frac{\pi(k - j)}{P}\right)\right],$$

where we can also understand the origin of the $\zeta$-function as $G_3^c(0,0) = \mathrm{Li}_3(1) = \zeta(3)$ and $G_3^c(0,\pi) = \mathrm{Li}_3(-1) = -\frac{3\zeta(3)}{4}$. Likewise, (2.25) is

$$
\begin{aligned}
c_{hol}\big|_{d=3} = & \frac{1}{8\pi}(N_0^2 + N_P^2)\gamma_E + \frac{1}{8\pi}N_0 N_P \log(2) \\
& + \frac{P}{4\pi^2}\sum_{j=1}^{P-1} F_j\left[ N_0 G_2^s\left(0, \frac{l\pi}{P}\right) + N_P G_2^s\left(0, \frac{(P-l)\pi}{P}\right)\right] \\
& - \frac{P^2}{16\pi^3}\sum_{j=1}^{P-1}\sum_{k=1}^{P-1} F_j F_k\left[ G_3^c\left(0, \frac{\pi(k+j)}{P}\right) - G_3^c\left(0, \frac{\pi(k-j)}{P}\right)\right].
\end{aligned}
$$

Here the offset-offset terms can be traced back to $G_1^c(0,\pi) = \mathrm{Li}_1(-1) = \log(2)$ and a regularisation of $G_1^c(0,0) = \mathrm{Li}_1(1) = \zeta(1)$.

## A.1 Potentials and special functions in odd dimensions

First, let us rewrite the expressions (A.4) and apply them to the potentials for the $\mathrm{AdS}_6$ and $\mathrm{AdS}_4$ geometries separately. We will write in terms of the coordinate $\xi = e^{-\frac{\pi}{P}[|\sigma|-i\eta]}$. For the supergravity background dual to the 5d QFT,

$$
\hat{W}_5(\sigma,\eta) = \frac{P}{\pi^2}\mathrm{Im}\left[N_0 \mathrm{Li}_2(\xi) - N_P \mathrm{Li}_2(-\xi)\right] + \frac{P^2}{2\pi^3}\sum_{j=1}^{P-1} F_j \mathrm{Re}\left[\mathrm{Li}_3\left(\xi e^{-\frac{i\pi J}{P}}\right) - \mathrm{Li}_3\left(\xi e^{\frac{i\pi J}{P}}\right)\right].
$$

In the background dual to the 3d QFT, we have $\hat{V}_3(\sigma,\eta)$ given by

$$
\begin{aligned}
\hat{V}_3(\sigma,\eta) = & -\frac{N_0}{2\pi}\log[(1-\xi)(1-\bar{\xi})] + \frac{N_P}{2\pi}\log[(1+\xi)(1+\bar{\xi})] \\
& + \frac{P}{2\pi^2}\sum_{j=1}^{P-1} F_j \mathrm{Im}\left[\mathrm{Li}_2\left(\xi e^{\frac{i\pi J}{P}}\right) - \mathrm{Li}_2\left(\xi e^{-\frac{i\pi J}{P}}\right)\right].
\end{aligned}
$$

Finally for the holographic central charge in the case of two rank functions $\mathcal{R}_1(\eta)$ and $\mathcal{R}_2(\eta)$ we found expressions in five dimensions and three dimensions that read,

$$
c_{hol}\big|_{d=5}[\mathcal{R}_1,\mathcal{R}_2] = \frac{P^2}{8\pi^6}\sum_{k=1}^{\infty}\frac{1}{k}\left[ R_{1,k}^2 + R_{2,k}^2 + 2R_{1,k}R_{2,k}e^{-\frac{k\pi\sigma_0}{P}}\left(1 + \frac{k\pi\sigma_0}{P}\right)\right],
$$

$$
c_{hol}\big|_{d=3}[\mathcal{R}_1,\mathcal{R}_2] = \frac{\pi}{32}\sum_{k=1}^{\infty}k\left[ R_{1,k}^2 + R_{2,k}^2 + 2R_{1,k}R_{2,k}e^{-\frac{k\pi\sigma_0}{P}}\left(1 + \frac{k\pi\sigma_0}{P}\right)\right].
$$

The two rank functions have Fourier coefficients

$$
R_{\alpha,k} = \frac{2}{k\pi}\left(N_{\alpha,0} + (-1)^{k+1}N_{\alpha,P}\right) + \frac{2P}{\pi^2 k^2}\sum_{j=1}^{P-1} F_{\alpha,j}\sin\left(\frac{k\pi j}{P}\right). \tag{A.6}
$$

The contributions of the squared terms are just two copies the one we have already calculated, see (2.21) and (2.25) for the 5d and 3d cases, respectively. Obviously, one copy with $N_{1,0}, N_{1,P}, F_{1,j}$ and another with $N_{2,0}, N_{2,P}, F_{2,j}$. What interests us is the crossed terms

$$
c_{hol}^{int}\big|_{d=5}[\mathcal{R}_1,\mathcal{R}_2] = \frac{P^2}{8\pi^6}\sum_{k=1}^{\infty}\frac{1}{k}\left[ 2R_{1,k}R_{2,k}e^{-\frac{k\pi\sigma_0}{P}}\left(1 + \frac{k\pi\sigma_0}{P}\right)\right],
$$

$$
c_{hol}^{int}\big|_{d=3}[\mathcal{R}_1,\mathcal{R}_2] = \frac{\pi}{32}\sum_{k=1}^{\infty}k\left[ 2R_{1,k}R_{2,k}e^{-\frac{k\pi\sigma_0}{P}}\left(1 + \frac{k\pi\sigma_0}{P}\right)\right].
$$

Using the Fourier expansion in (A.6) we find

$$c_{hol}^{int}[\mathcal{R}_1, \mathcal{R}_2] = \sum_{I=1}^{3} T_I \,,$$

where in $d = 5$ we have

$$T_1\Big|_{d=5} = \frac{P^2}{8}(N_{1,0}N_{2,0} + N_{1,P}N_{2,P})\Big[\text{Li}_3\left(e^{-\frac{\pi\sigma_0}{P}}\right) - \text{Li}_3\left(-e^{-\frac{\pi\sigma_0}{P}}\right) + \frac{\pi\sigma_0}{P}\text{Li}_2\left(e^{-\frac{\pi\sigma_0}{P}}\right)$$
$$-\frac{\pi\sigma_0}{P}\text{Li}_2\left(-e^{-\frac{\pi\sigma_0}{P}}\right)\Big]\,,$$

$$T_2\Big|_{d=5} = \frac{P^3}{\pi^9}\sum_{j=1}^{P-1}(F_{2,j}N_{1,0} + F_{1,j}N_{2,0})\text{Im}\Big[\text{Li}_4\left(e^{-\frac{\pi(\sigma_0-ij)}{P}}\right) + \frac{\pi\sigma_0}{P}\text{Li}_3\left(e^{-\frac{\pi(\sigma_0-ij)}{P}}\right)\Big]$$
$$+\frac{P^3}{\pi^9}\sum_{j=1}^{P-1}(F_{2,j}N_{1,P} + F_{1,j}N_{2,P})\text{Im}\Big[\text{Li}_4\left(-e^{-\frac{\pi(\sigma_0+ij)}{P}}\right) + \frac{\pi\sigma_0}{P}\text{Li}_3\left(-e^{-\frac{\pi(\sigma_0+ij)}{P}}\right)\Big]\,,$$

$$T_3\Big|_{d=5} = \frac{P^4}{2\pi^{10}}\sum_{j=1}^{P-1}\sum_{\ell=1}^{P-1}F_{1,j}F_{2,\ell}\text{Re}\Big[\text{Li}_5\left(e^{-\frac{\pi(\sigma_0+i(j-\ell))}{P}}\right) - \text{Li}_5(e^{-\frac{\pi(\sigma_0+i(j+\ell))}{P}})$$
$$+\frac{\pi\sigma_0}{P}\text{Li}_4\left(e^{-\frac{\pi(\sigma_0+i(j-\ell))}{P}}\right) - \frac{\pi\sigma_0}{P}\text{Li}_4\left(e^{-\frac{\pi(\sigma_0+i(j+\ell))}{P}}\right)\Big]\,,$$

and in $d = 3$ we have

$$T_1\Big|_{d=3} = \frac{1}{4\pi}(N_{1,0}N_{2,0} + N_{1,P}N_{2,P})\Big[\text{Li}_1\left(e^{-\frac{\pi\sigma_0}{P}}\right) - \text{Li}_1\left(-e^{-\frac{\pi\sigma_0}{P}}\right) + \frac{\pi\sigma_0}{P}\text{Li}_0\left(e^{-\frac{\pi\sigma_0}{P}}\right)$$
$$-\frac{\pi\sigma_0}{P}\text{Li}_0\left(-e^{-\frac{\pi\sigma_0}{P}}\right)\Big]\,,$$

$$T_2\Big|_{d=3} = \frac{P}{4}\sum_{j=1}^{P-1}(F_{2,j}N_{1,0} + F_{1,j}N_{2,0})\text{Im}\Big[\text{Li}_2\left(e^{-\frac{\pi(\sigma_0-ij)}{P}}\right) + \frac{\pi\sigma_0}{P}\text{Li}_1\left(e^{-\frac{\pi(\sigma_0-ij)}{P}}\right)\Big]$$
$$+\frac{P}{4}\sum_{j=1}^{P-1}(F_{2,j}N_{1,P} + F_{1,j}N_{2,P})\text{Im}\Big[\text{Li}_2\left(-e^{-\frac{\pi(\sigma_0+ij)}{P}}\right) + \frac{\pi\sigma_0}{P}\text{Li}_1\left(-e^{-\frac{\pi(\sigma_0+ij)}{P}}\right)\Big]\,,$$

$$T_3\Big|_{d=3} = \frac{P^2}{8\pi^3}\sum_{j=1}^{P-1}\sum_{\ell=1}^{P-1}F_{1,j}F_{2,\ell}\text{Re}\Big[\text{Li}_3\left(e^{-\frac{\pi(\sigma_0+i(j-\ell))}{P}}\right) - \text{Li}_3(e^{-\frac{\pi(\sigma_0+i(j+\ell))}{P}})$$
$$+\frac{\pi\sigma_0}{P}\text{Li}_2\left(e^{-\frac{\pi(\sigma_0+i(j-\ell))}{P}}\right) - \frac{\pi\sigma_0}{P}\text{Li}_2\left(e^{-\frac{\pi(\sigma_0+i(j+\ell))}{P}}\right)\Big]\,.$$

Except for the overall coefficient and the ensuing scaling with $P$, the terms $T_I$ are uniform across odd dimensions and assume the form $\pm\Big[\text{Li}_{d-3+I}(\pm\cdot) + \frac{\pi\sigma_0}{P}\text{Li}_{d-4+I}(\pm\cdot)\Big]$.

## B Five-dimensional quivers in M-theory

In this appendix we collect observations comparing our results with the geometric engineering of the five-dimensional linear quivers in M-theory.

To begin, let us remind that the 5d linear quivers we deal with arise as deformations of 5d SCFTs. The latter can be engineered in M-theory via compactification on toric Calabi–Yau threefold ($CY_3$) singularities [83] (see also [84, 107, 116, 132, 133] for a minimal sample of related works). Prototypical examples are the homogeneous quiver, also known as the $+_{N,P}$

quiver:

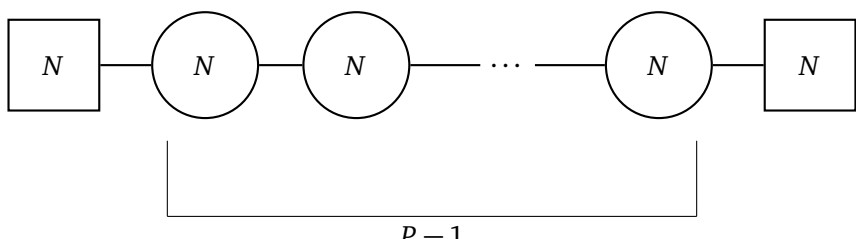

and the $T_N$ quiver:

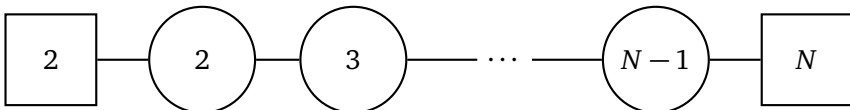

where all the gauge nodes are $SU(N_j)$. The corresponding toric diagrams are:

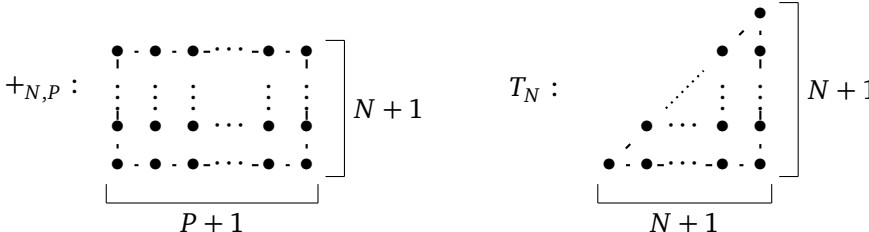

Comparing the M-theory setup with Section 3 gives us the following insights.

- The fact that the rank function $\mathcal{R}(\eta)$ is piecewise linear is reminiscent of tropical geometry, which famously appears in the study of toric varieties. The precise connection is that the graph of $\mathcal{R}(\eta)$, closed up along the $\eta$-axis, gives the perimeter of the toric polygon of $CY_3$. The area below the graph of $\mathcal{R}(\eta)$, which computes the total rank of the gauge theory, is a topological invariant,

$$\int_0^P d\eta \, [\mathcal{R}(\eta) - 1] = \frac{1}{2}\chi(CY_3).$$

This identity stems from the fact that $\chi(CY_3)$ is twice the area of the toric polygon for $CY_3$. Recall from (2.11c) that, on the Type IIB string theory side, $\int_0^P d\eta \, [\mathcal{R}(\eta) - 1]$ is the total number of D5 branes with the centre of mass removed, i.e. the actual rank of the 5d gauge theory.

- One effect of S-duality is to rotate the toric diagram of $CY_3$ by 90°. We have predicted a scaling of the free energy $\mathcal{F} \propto N^2 P^4$, which may not be invariant under this transformation. Comparing with the explicit results in [74] for $+_{N,P}$, $T_N$ and other theories, we observe in all examples that

$$\mathcal{F} \propto [\chi(CY_3)]^2$$

is manifestly invariant under rotations of the toric diagram. Therefore, the QFT corrects the naive scaling and matches with the M-theory expectations.

- Throughout Section 3, we dismissed the distinction between $SU(N_j)$ and $U(N_j)$ gauge nodes at large $N$, arguing that it is a subleading contribution (supported by explicit calculations in [109]). This allowed us to effectively work with unitary gauge groups in 5d.

Nevertheless, 5d linear quivers with $U(N_j)$ gauge nodes were geometrically engineered in [134]. One intriguing outcome of that work is a geometric prescription, valid for the balanced quiver we study, to decouple the central $U(1)_j \subset U(N_j)$ factors and recover special unitary gauge nodes.

From the translation of the M-theory setup to Type IIB string theory, done in [134, Sec.7], the picture that emerges is very reminiscent of the analysis we have carried out in the main text. It would be extremely interesting to come up with a refined probe of the large $N$ limit capable to discern between unitary and special unitary gauge groups. Having such a finer probe, we may compare the electrostatic problem in Figure 4 and its generalizations with [134, Sec.7]. In particular, if one could argue that the electrostatic problem gives an AdS$_6$ solution dual to a long quiver with unitary gauge nodes, this may hint at the existence of an SCFT from which these gauge theories descend.

- Related to the previous item, assuming unitary gauge nodes in the 5d quiver, we can introduce FI parameters accordingly. Viewing an $SU(N)$ gauge node as a $U(N)$ with a gauged centre of mass, reducing $U(N) \to SU(N)$ dictates to promote the FI parameter to a dynamical field and integrate over it. At the level of $\mathbb{S}^5$ partition function, this statement boils down to the Fourier transform identity

$$\int_{-\infty}^{+\infty} d\xi \, e^{2\pi\xi \sum_a \sigma_a} = \delta\left(\sum_a \sigma_a\right).$$

This idea was used in [109, App.A2] to show how the matrix model cleanly captures the M-theoretical mechanism of [134].

## C  Integrating out massive fields in long quivers

In this appendix we discuss the effect of integrating out massive fields on the sphere free energy. The idea is that, in the limit $m \to \infty$, $\mathcal{F}$ splits into the free energy of a theory will less fields, plus the contribution of decoupled, heavy fields. The discussion below is a quantitative test of the analysis in Subsection 3.1, using partition functions.

### C.1  Premise: mass deformation from branes

The aim of this appendix is to complement the description of the mass deformation explained in Subsection 3.1. The choice of mass deformation that corresponds to our prescription is encapsulated into a splitting of the rank function $\mathcal{R}(\eta) = \sum_{\alpha=1}^{\mathscr{F}} \mathcal{R}_\alpha(\eta)$, which induces a specific breaking pattern of flavour and gauge groups.

The choice of mass deformation is more transparent in the Hanany–Witten brane setup in Type IIB string theory. For clarity, we formulate the discussion in the case of 3d $\mathcal{N} = 4$ and for $\mathscr{F} = 2$. A portion of the brane system that leads to a linear quiver is depicted in Figure 10, to which we refer for detailed description.

A naive mass deformation would consist in sliding $F_{2,j}$ flavour branes, so that they are not intersected by the colour branes. From the perspective of the 3d gauge theory, that mass deformation would give mass to $N_j F_{2,j}$ hypermultiplet modes at each node, leaving all other modes massless. However, this is not the only brane configuration consistent with sliding $F_{2,j}$ flavour branes. One can in fact select arbitrary numbers $N_{2,j}$ and demand that $N_{1,j} F_{2,j} + N_{2,j} F_{1,j}$ hypermultiplet modes become massive. This corresponds to drag $N_{2,j}$ colour branes vertically together with the flavour branes, as shown in Figure 10. In this way, the $F_{2,j} N_{2,j}$ hypermultiplet modes arising from strings stretched between the D3 and D5 branes in the same stack remain

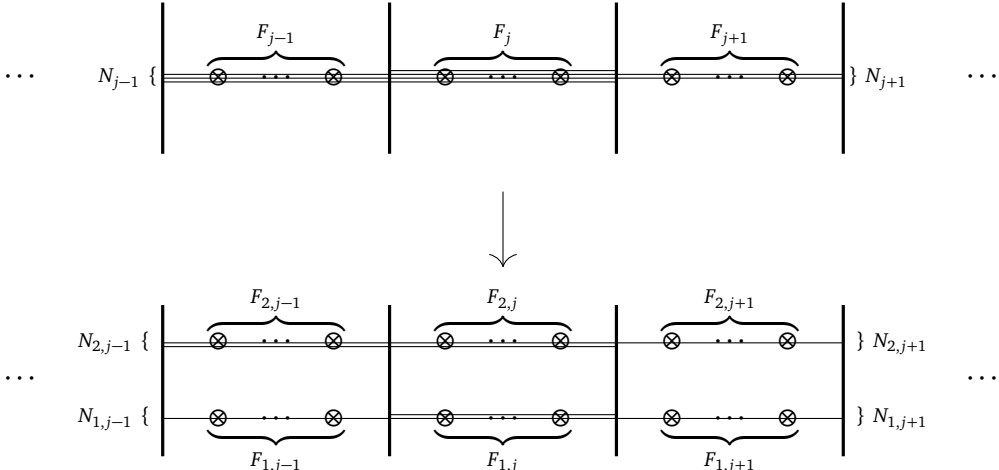

Figure 10: A portion of the Hanany–Witten brane setup for a balanced 3d $\mathcal{N} = 4$ linear quiver. Black vertical lines represent NS5 branes, horizontal lines are D3 branes, $\otimes$ are D5 branes. In the $j^{\text{th}}$ interval between two consecutive NS5 branes, there are $N_j$ D3 branes (colour branes) and $F_j$ D5 branes (flavour branes). The numbers $F_j$ are fixed by the balancing condition. Above, all the D3 brane segments intersect all the D5 branes in the same interval. Below, we have specified our mass deformation, as well as a point on the Coulomb branch of the gauge theory. The D5 branes have been split into two stacks, vertically separated, whose distance determines the mass parameter $m$. We have also prescribed the numbers $N_{2,j}$ of D3 brane segments that are moved vertically together with the D5 branes, thus specifying a point on the Coulomb branch of the theory.

massless. As a consequence, there are strings stretched between D3 branes in the different stacks, that are vertically separated. In the gauge theory, these D3-D3 strings give rise to massive W-bosons, reflecting a breaking of the gauge group. All the possible configurations of this type are to be taken into account, as they describe different points on the Coulomb branch of the theory.

## C.2 Warm-up example: 3d SQCD

To present the idea, let us begin with a very simple example: SQCD in 3d with gauge group $U(N)$ and flavour symmetry algebra $\mathfrak{su}(F)$. The balancing condition simply reads $F = 2N$, but we need not impose it. Related discussion on this 3d SQCD example is in [135].[16]

We assume $F_1$ out of the $F$ hypermultiplets have mass $m_1$ and the remaining $F_2 = F - F_1$ hypermultiplets have mass $m_2$. The traceless condition on the flavour symmetry imposes $F_1 m_1 + F_2 m_2 = 0$. We also write $N = N_1 + N_2$. To lighten the expressions, we adopt the (by now, standard) shorthand $\text{sh}(s) := \sinh(\pi s)$ and $\text{ch}(s) := \cosh(\pi s)$.

The three-sphere partition function of this model is

$$\mathcal{Z}_{\mathbb{S}^3} = \frac{1}{N!} \int_{\mathbf{R}^N} \mathrm{d}\vec{\sigma}' \frac{\prod_{1 \leq a < b \leq N} \left(2\text{sh}(\sigma'_a - \sigma'_b)\right)^2}{\prod_{a=1}^{N} \left(2\text{ch}(\sigma'_a - m_1)\right)^{F_1} \left(2\text{ch}(\sigma'_a - m_2)\right)^{F_2}}. \tag{C.1}$$

---

[16]For useful exact evaluations of the sphere partition function of various theories, see [136–139].

We now use the change of variables

$$
\begin{aligned}
\sigma_a &= \sigma'_a - m_1, & a &= 1, \dots, N_1, \\
\phi_{\dot a} &= \sigma'_{\dot a + N_1} - m_2, & \dot a &= 1, \dots, N_2,
\end{aligned}
\tag{C.2}
$$

and, defining $m := m_1 - m_2$, write

$$
\mathcal{Z}_{\mathbb{S}^3} = \frac{1}{N_1!} \int_{\mathbf{R}^{N_1}} \mathrm{d}\vec\sigma \, \frac{\prod_{1 \le a < b \le N_1} \left(2\mathrm{sh}(\sigma_a - \sigma_b)\right)^2}{\prod_{a=1}^{N_1} \left(2\mathrm{ch}(\sigma_a)\right)^{F_1}} \cdot \frac{1}{N_2!} \int_{\mathbf{R}^{N_2}} \mathrm{d}\vec\phi \, \frac{\prod_{1 \le \dot a < \dot b \le N_2} \left(2\mathrm{sh}(\phi_{\dot a} - \phi_{\dot b})\right)^2}{\prod_{\dot a=1}^{N_2} \left(2\mathrm{ch}(\phi_{\dot a})\right)^{F_2}}
\tag{C.3a}
$$

$$
\times \frac{\prod_{a=1}^{N_1} \prod_{\dot b=1}^{N_2} \left(2\mathrm{sh}(\sigma_a - \phi_{\dot b} + m)\right)^2}{\prod_{a=1}^{N_1} \left(2\mathrm{ch}(\sigma_a + m)\right)^{F_2} \prod_{\dot a=1}^{N_2} \left(2\mathrm{ch}(\phi_{\dot a} - m)\right)^{F_1}} .
\tag{C.3b}
$$

The breaking of the Weyl group $S_N \to S_{N_1} \times S_{N_2}$ has brought in a factor $\binom{N_1 + N_2}{N_2}$, which corresponds to the number of permutations that shuffle (C.2) without changing (C.3), leading to the overall factor

$$
\frac{1}{(N_1 + N_2)!} \binom{N_1 + N_2}{N_2} = \frac{1}{N_1!} \cdot \frac{1}{N_2!},
$$

which correctly accounts for the permutations of the $\sigma_a$ and the $\phi_{\dot a}$ separately.

Let us now unpack expression (C.3) (see also [92] for related considerations).

- Keeping only (C.3a), we would get the product of two disconnected, *conformal* quivers:

  - SQCD with gauge group $U(N_1)$ and $F_1$ massless hypermultiplets (with adjusted variable $\vec\sigma$),

  - and SQCD with gauge group $U(N_2)$ and $F_2$ massless hypermultiplets (with adjusted variable $\vec\phi$).

- The interaction is entirely in (C.3b), and is $m$-dependent. In it we recognise:

  - the left term in the denominator of (C.3b) is the one-loop determinant of $F_2$ hypermultiplets of real mass $-m$, charged under $U(N_1)$;

  - the right term in the denominator of (C.3b) the one-loop determinant of $F_1$ hypermultiplets of real mass $-m$, charged under $U(N_2)$;

  - the numerator of (C.3b) is the part of the vector multiplet one-loop determinant coming from the W-bosons associated to the $U(N_1 + N_2)$ roots $\pm\alpha_{a, \dot b + N_1}$. The standard masses $\pm(\sigma_a - \phi_{\dot b})$ due to the theory being on the Coulomb branch receive an additional shift $\pm m$.

If we require the two resulting theories to be balanced,

$$
F_1 = 2N_1, \qquad F_2 = 2N_2,
\tag{C.4}
$$

this gives us a preferred choice of the integers $N_1, N_2$, and the initial quiver is automatically balanced. We will show below that (C.4) indeed correctly describes the IR theory.

The Hanany–Witten brane system read off from the matrix model (C.3) is drawn in Figure 11. The identifications are as follows:

- The purple strings in the lower part of Figure 11 give rise to the left part of the integrand in (C.3a), while the purple strings in the upper part of Figure 11 give rise to the right part of the integrand in (C.3a);

- The gray, curly strings in Figure 11 give rise to (C.3b):

  — the gray string in the left part of Figure 11 gives rise to (one of the factors in) the left term in the denominator of (C.3b);

  — the gray string in the right part of Figure 11 gives rise to (one of the factors in) the right term in the denominator of (C.3b);

  — the gray string in the middle of Figure 11 gives rise to the numerator of (C.3b).

Analogous Hanany–Witten setups of NS5, D$d$ and D$(d + 2)$ branes can be read off from the most general matrix models for linear quivers we discuss in the main text.

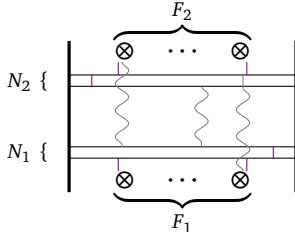

Figure 11: Hanany–Witten brane setup corresponding to the matrix model (C.3). Black vertical lines represent NS5 branes, horizontal lines are D3 branes, $\otimes$ are D5 branes. Purple vertical lines represent a sample of the strings giving rise to light modes, contributing to (C.3a). Gray, vertical, curly lines are strings that give rise to heavy modes, contributing to (C.3b). The picture extends straightforwardly to all the mass-deformed linear quivers considered in the present work. In the conventions of (3.16), the purple strings contribute to $\mathcal{F}_\alpha$ and the gray strings to $\mathcal{F}_{\alpha,\beta}^{\text{int}}$.

### C.2.1 Large mass: Coulomb branch considerations

The Coulomb branch of SQCD under consideration, with the masses turned on, has two singularities: at $m_1$ and $m_2$. In the presentation (C.1), $\vec{\sigma}'$ is probing such Coulomb branch. The two singularities are spread apart if $m = m_1 - m_2 \to \infty$. In terms of the adjusted variables $\vec{\sigma}, \vec{\phi}$, these two singularities are mapped to 0 and $\mp m$. However, there cannot be exact factorisation of the partition function in the limit $m \to \infty$, as evident from counting Higgs branch dimensions. The initial theory has $(N_1 + N_2)(F_1 + F_2)$ hypermultiplet zero-modes, but the two disconnected SQCD theories in (C.3a) only sum up to $N_1 F_1 + N_2 F_2$ hypermultiplet modes. The missing $N_1 F_2 + N_2 F_1$ are precisely the ones appearing in (C.3b).

### C.2.2 Large mass: free energy considerations

From the integration in (C.3), we write

$$\mathcal{Z}_{\mathbb{S}^3}[N_2] = \frac{1}{N_1!} \int_{\mathbf{R}^{N_1}} d\vec{\sigma} \; e^{-S_{\text{eff},1}(\vec{\sigma})} \frac{1}{N_2!} \int_{\mathbf{R}^{N_2}} d\vec{\phi} \; e^{-S_{\text{eff},2}(\vec{\phi})} \cdot e^{-S_{\text{int}}(\vec{\sigma},\vec{\phi},m)}, \qquad (C.5)$$

where we use the notation $\mathcal{Z}_{\mathbb{S}^3}[N_2]$ to stress that we are making a specific choice of splitting $N = N_1 + N_2$. The interaction term $S_{\text{int}}$ is the logarithm of (C.3b). We schematically write the integrals as

$$\int_{\mathbf{R}^{N_1}} d\vec{\sigma} \int_{\mathbf{R}^{N_2}} d\vec{\phi} = \int_{[-m,m]^{N_1}} d\vec{\sigma} \int_{[-m,m]^{N_2}} d\vec{\phi} + \cdots,$$

where the dots include a sum of integrals whose domains involve at least one $m < |\sigma_a|, |\phi_{\dot{a}}| < \infty$. In the limit $m \to \infty$, the domain $[-m, m]^{N_1} \times [-m, m]^{N_2}$ converges to $\mathbf{R}^{N_1} \times \mathbf{R}^{N_2}$ whereas the rest has vanishing measure.

Approximating $S_{\text{int}}$ at large $m$ in the domain $-m < \sigma_a < m$ and $-m < \phi_{\dot{a}} < m$, we have

$$
S_{\text{int}} \approx \left( \pi F_2 N_1 m + \pi F_2 \sum_{a=1}^{N_1} \sigma_a \right) + \left( \pi F_1 N_2 m - \pi F_1 \sum_{\dot{a}=1}^{N_2} \phi_{\dot{a}} \right)
$$
$$
- \left( 2\pi N_1 N_2 m + 2\pi N_2 \sum_{a=1}^{N_1} \sigma_a - 2\pi N_1 \sum_{\dot{a}=1}^{N_2} \phi_{\dot{a}} \right).
$$

The three contributions, grouped in brackets, account for:

- Integrating out the $F_2$ massive hypermultiplets on the $\vec{\sigma}$-branch;

- Integrating out the $F_1$ massive hypermultiplets on the $\vec{\phi}$-branch;

- Integrating out the massive W-bosons from the Higgsing $U(N_1 + N_2) \to U(N_1) \times U(N_2)$.

We emphasise that integrating out fields produces *imaginary* FI terms. While the presence of FI couplings was already noted in [92], it is important to stress that they are imaginary. This unusual feature is essential to ensure the convergence of the partition function, as it must be if we start with a good quiver. Furthermore, notice that, assuming the balancing condition (C.4), these imaginary FI terms cancel out.

### C.2.3 Large mass: determining the gauge ranks

What remains to do is to determine which are the correct values of $N_1$ and $N_2$, closely following [92, Sec.5]. As already anticipated, the answer will be the balanced result (C.4).

From the perspective of the Coulomb branch, the problem can be phrased as follows. The change of variables from $\vec{\sigma}'$ to $(\vec{\sigma}, \vec{\phi})$ probes different subspaces of the Coulomb branch for different values of $N_2$. We ought to determine which of them survives the RG flow triggered by the mass deformations, and describes the IR fixed point. Equivalently, from the brane perspective of Appendix C.1, we ought to determine which one, among all the brane configurations with the D3 branes split in two different stacks, gives the dominant contribution when the two stacks are pulled far apart.

The CB vacuum that dominates in the large $m$ limit is determined looking at $S_{\text{int}}$. In particular, focusing on the $m$-dependence and using $2(N_1 + N_2) = F_1 + F_2$, we immediately obtain

$$
S_{\text{int}} = \frac{\pi}{2} m \left[ F_1 F_2 + (F_2 - 2N_2)^2 \right] + \cdots,
$$

where we have written down only the terms which determine the leading behaviour as $m \to \infty$. In this expression, $F_\alpha$ are input data of the deformation, while $N_2$ is to be determined. Note that, starting with a balanced quiver, having odd $F_\alpha$ would result in an overbalanced and an underbalanced quiver, thus we do not explore this case further.

Due to the appearance of $e^{-S_{\text{int}}}$ in the partition function, the choice of CB vacuum that dominates in the limit corresponds to $N_2 = \frac{F_2}{2}$. Explicitly, at every finite $m$ we can write

$$
\mathcal{Z}_{\mathbb{S}^3} = \frac{1}{N+1} \sum_{N_2=0}^{N} \mathcal{Z}_{\mathbb{S}^3}[N_2], \tag{C.6}
$$

where the right-hand side is given in (C.5). This rewriting is obvious, since the terms on the right-hand side are all related by a change of variables, thus attain the same value. However,

the convenience of the rewriting (C.6) is that only one of the terms $\mathcal{Z}_{\mathbb{S}^3}[N_2]$ will survive in the IR. This will allow us to directly read off the IR quivers.

We have already obtained that, at large $m$, expression (C.6) reads

$$\mathcal{Z}_{\mathbb{S}^3} = \frac{1}{N+1} \sum_{N_2=0}^{N} e^{-\left\{\frac{\pi}{2}m\left[F_1 F_2 + (F_2 - 2N_2)^2\right] + \cdots\right\}} \prod_{\alpha=1}^{2} \mathcal{Z}_{\mathbb{S}^3}^{\mathrm{SQCD}_\alpha}(\xi_\alpha), \tag{C.7}$$

where $\mathcal{Z}_{\mathbb{S}^3}^{\mathrm{SQCD}_\alpha}$ is the sphere partition function of 3d $\mathcal{N} = 4$ $U(N_\alpha)$ gauge theory with $F_\alpha$ fundamental hypermultiplets, and imaginary FI parameter $\xi_\alpha$ given by

$$\xi_1 = i\left(\frac{F_2}{2} - N_2\right), \qquad \xi_2 = -i\left(\frac{F_1}{2} - N_1\right).$$

Sending $m \to \infty$, the least suppressed term in (C.7) yields the leading contribution. We conclude that, at the end of the flow triggered by the mass deformation, we obtained the balancing condition (C.4). This result agrees with [92, Sec.5] and generalises it to the case $F_2 \geq 2$.

### C.2.4 Take-home lesson

To sum up, the lesson from this example is that, for a balanced theory with two mass scales for the hypermultiplets,

$$\lim_{|m_1 - m_2| \to \infty} [\mathcal{F} - \mathcal{F}_{\mathrm{dec.}}] = \mathcal{F}_1 + \mathcal{F}_2.$$

Here $\mathcal{F}$ is the free energy of the theory we started with, namely $U(N_1 + N_2)$ with $F_1 + F_2$ flavours in this example, and $\mathcal{F}_\alpha$ is the free energy of the quiver $\mathcal{Q}_\alpha$, with gauge and flavour ranks $N_\alpha$ and $F_\alpha$ respectively. Finally, $\mathcal{F}_{\mathrm{dec.}}$ is the contribution from the heavy fields, that will decouple at the end of the RG flow. In this SQCD example:

$$\mathcal{F}_{\mathrm{dec.}} = \underbrace{\pi m(F_1 N_2 + F_2 N_1)}_{\text{hypermultiplets}} \underbrace{- 2\pi m N_1 N_2}_{\text{W-bosons}}.$$

The necessity to discard the scheme-dependent part $\mathcal{F}_{\mathrm{dec.}}$ from the definition of the free energy was already pointed out in [98,99] (see also [125]). In thus simple example we chose the simplest and most transparent regularisation, in which we subtract the gravitational counterterm listed in [125] with coefficient an integer multiple of $\pi$. In the next subsection we will adopt a choice of regularisation scheme closer in spirit to [99].

### C.2.5 A remark on the F-theorem

We have reviewed in Subsection 3.6 that the F-theorem requires to discard the contributions to $\mathcal{F}$ which are polynomial in the mass parameters [99]. Let us exemplify this in SQED with $F$ flavours. Giving opposite masses $\pm m$ to two hypermultiplets, we have

$$\mathcal{F} = -\ln\left[\int_{-\infty}^{\infty} \frac{d\sigma}{(2\mathrm{ch}(\sigma))^{F-2} \cdot 2\mathrm{ch}(\sigma + m) \cdot 2\mathrm{ch}(\sigma - m)}\right].$$

Ranging $0 \leq m < \infty$ describes the RG flow from SQED with $F$ flavours to SQED with $F - 2$ flavours, plus two decoupled hypermultiplets with large mass $\pm m$. As argued above, we consider $\mathcal{F}_{\mathrm{EFT}} = \mathcal{F} - 2\pi|m|$. This quantity is plotted in Figure 12, and clearly interpolates monotonically between the free energies of the two SCFTs.

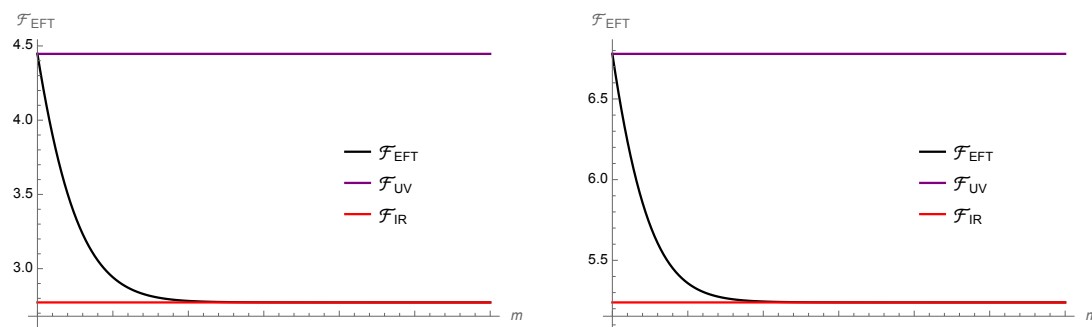

Figure 12: RG flow in $d = 3$ from SQED with $F$ flavours in the UV to SQED with $F-2$ flavours in the IR. $\mathcal{F}_{\text{EFT}}$ interpolates monotonically between the free energy $\mathcal{F}_{\text{UV}}$ of the UV CFT and that of the $\mathcal{F}_{\text{IR}}$ IR CFT. Left: $F = 5$. Right: $F = 8$.

### C.3   Massive fields in long quivers

We now apply the above discussion to long quivers. We work in the setup of Subsection 3.4: we consider a long linear quiver with $\mathscr{F} = 2$ mass scales, $m_1 = 0$ and $m_2 = m$. In Subsection 3.4 we used the long quiver analogue of the change of variables (C.2). It was also shown therein that, in perfect analogy with the SQCD example, the effective action splits into

$$S_{\text{eff}} = S_{\text{eff},1}(\vec{\sigma}) + S_{\text{eff},2}(\vec{\phi}) + S_{\text{int}}(\vec{\sigma}, \vec{\phi}, m).$$

Again, as in the toy example, the mass dependence is entirely encoded in the interaction term, whereas the two terms $S_{\text{eff},\alpha}$ describe superconformal quivers $\mathcal{Q}_\alpha$.

From (3.25), we have that $S_{\text{int}}$ splits into four terms:

$$S_{\text{int}} = \underbrace{S_{\text{fund},1} + S_{\text{fund},2}}_{\text{fund. hypers}} + \underbrace{S_0 + S_{\text{der}}}_{\text{W-bosons + bifund. hypers}}, \tag{C.8}$$

where the contributions of Higgsed W-bosons and of bifundamental hypermultiplets have been combined together and rewritten in the form $S_0 + S_{\text{der}}$. The details are in Subsections 3.2-3.4. Here we evaluate (C.8) in the large $\mu$ limit, in 5d and 3d. We adopt the notation and normalisation

$$\mathcal{F}_\bullet^{\text{dec.}} = (-1)^{\frac{d-3}{2}} S_\bullet|_{\text{on-shell}, m \to \infty}.$$

For instance, the sphere free energy of free massive hypermultiplets is

$$\mathcal{F}_{\text{fund}}^{\text{dec.}} = (-1)^{\frac{d-3}{2}} P^{(d-2)} f_h(\mu) \cdot (\# \text{ fund. hypermultiplets}).$$

In our long quivers, the number of fundamental hypermultiplets that eventually decouple is

$$\# \text{ fund. hypermultiplets} = \sum_{j=1}^{P-1} \left( N_{1,j} F_{2,j} + N_{2,j} F_{1,j} \right).$$

In the long quiver limit, this becomes

$$\# \text{ fund. hypermultiplets} = P N^2 \int_0^1 dz \left[ \nu_1(z) \zeta_2(z) + \nu_2(z) \zeta_1(z) \right].$$

Recalling the normalisation condition $\int dx \varrho(z, x) = \nu(z)$, we get:

$$\mathcal{F}_{\text{fund},1}^{\text{dec.}} + \mathcal{F}_{\text{fund},2}^{\text{dec.}} = (-1)^{\frac{d-3}{2}} N^2 P^{(d-1)} f_h(\mu) \int_0^1 dz \int_{-|\mu|}^{|\mu|} \left[ \varrho_1(z, x) \zeta_2(z) + \varrho_2(z, x) \zeta_1(z) \right].$$

Crucially, we are working in an EFT description, in which we cut off the modes larger than $|\mu|$. This effective description becomes exact in the IR limit, $|\mu| \to \infty$. However, to compute the contribution from the background fields at intermediate mass scales, we assume that only the modes $-|\mu| < x < |\mu|$ remain dynamical (i.e. have not been integrated out), and hence contribute to the eigenvalue density.

While the integration range may appear counter-intuitive at first sight, we stress that we are focusing on the modes that will eventually decouple. Their free energy must vanish at $|\mu| \to 0$ and must become that of free massive fields at $|\mu| \to \infty$, which is precisely accounted for by our choice of integration range.

The computations that follow in this appendix are similar to those in Section 3.4, but rely on a careful analysis of the integration domain, to identify the regions corresponding to heavy modes that decouple in the IR.

### C.3.1 The contribution of massive fields in 5d

We compute the contribution of massive fields in 5d using the EFT approach just outlined. We utilise the definitions (3.8). For the contribution of the massive fundamental hypermultiplets, we find

$$
\frac{1}{N^2 P^2} \mathcal{F}_{\text{fund},1}^{\text{dec.}} = \frac{9\pi}{2} |\mu|^3 \int_0^1 dz P^2 \zeta_2(z) \int_{-|\mu|}^{+|\mu|} dx \varrho_1(z,x)
$$

$$
= \frac{9\pi^4}{2} |\mu|^3 \sum_{k,\ell=1}^{\infty} \mathfrak{a}_{1,k} \mathfrak{a}_{2,\ell} k \ell^2 \int_0^1 dz \sin(k\pi z) \sin(\ell \pi z) \int_{-|\mu|}^{+|\mu|} dx \, e^{-2\pi\ell|x|}
$$

$$
= \frac{9\pi^3}{4} |\mu|^3 \sum_{k=1}^{\infty} \mathfrak{a}_{1,k} \mathfrak{a}_{2,k} k^2 \left(1 - e^{-2\pi k|\mu|}\right).
$$

$\mathcal{F}_{\text{fund},2}^{\text{dec.}}$ is evaluated switching the labels $1 \leftrightarrow 2$, but the final expression is invariant under this operation. Therefore

$$
\mathcal{F}_{\text{fund},1}^{\text{dec.}} + \mathcal{F}_{\text{fund},2}^{\text{dec.}} = \frac{9\pi^3}{2} |\mu|^3 \sum_{k=1}^{\infty} \mathfrak{a}_{1,k} \mathfrak{a}_{2,k} k^2 \left(1 - e^{-2\pi k|\mu|}\right).
$$

The next term we need is $\mathcal{F}_{\text{der}}^{\text{dec.}}$. The integration domain this time involves two variables $x, y$. Let us assume $\mu \gg 0$ for clarity. Restricting $|x| \le \mu$, any $y > 0$ is allowed, so that $|x - y - \mu|^3$ and $\mu^3$ have the same sign. On the other hand, restricting $|y| \le \mu$, the integral ranges over all $x < 0$. The total contribution is the sum of these two expressions, which contribute equally. We thus find:

$$
\frac{1}{N^2 P^2} \mathcal{F}_{\text{der}}^{\text{dec.}} = 2 \cdot \frac{9\pi}{2} |\mu|^3 \int_0^1 dz \int_{-|\mu|}^{|\mu|} dx \int_0^{\infty} dy \left[ \frac{\varrho_1(z,x) \partial_z^2 \varrho_2(z,y) + \varrho_2(z,y) \partial_z^2 \varrho_1(z,x)}{2} \right]
$$

$$
= 9\pi^3 |\mu|^3 \sum_{k,\ell=1}^{\infty} \mathfrak{a}_{1,k} \mathfrak{a}_{2,\ell} k \ell \int_0^1 dz \left[ \frac{-(\pi\ell)^2 - (\pi k)^2}{2} \right] \sin(k\pi z) \sin(\ell \pi z)
$$

$$
\times \int_{-|\mu|}^{|\mu|} dx \, e^{-2\pi k|x|} \int_0^{\infty} dy \, e^{-2\pi\ell|y|}
$$

$$
= -\frac{9\pi^5}{2} |\mu|^3 \sum_{k=1}^{\infty} \mathfrak{a}_{1,k} \mathfrak{a}_{2,k} \cdot k^4 \cdot \int_{-|\mu|}^{|\mu|} dx \, e^{-2\pi k|x|} \int_0^{\infty} dy \, e^{-2\pi\ell|y|}
$$

$$
= -\frac{9\pi^3}{4} |\mu|^3 \sum_{k=1}^{\infty} \mathfrak{a}_{1,k} \mathfrak{a}_{2,k} k^2 \left(1 - e^{-2\pi k|\mu|}\right).
$$

Finally, for $\mathcal{F}_0^{\text{dec.}}$ we use $\mathfrak{f}_0(\mu)|_{d=5} = \frac{27\pi}{8}|\mu|$ from (3.8) and repeat the analysis of the integration range identically as for the companion term $\mathcal{F}_{\text{der}}^{\text{dec.}}$. Indeed, the contribution have the same origin in the matrix model and moreover $\text{sgn}((\mu+y-x)^3) = \text{sgn}(\mu+y-x)$. We get

$$
\begin{aligned}
\frac{1}{N^2 P^2}\mathcal{F}_0^{\text{dec.}} &= -2 \cdot \left(-\frac{27\pi}{8}|\mu|\right) \cdot 2 \int_0^1 dz \int_{-|\mu|}^{|\mu|} dx \int_0^\infty dy\, \varrho_2(z,y) \\
&= \frac{27\pi^3}{2}|\mu| \sum_{k,\ell=1}^\infty \mathfrak{a}_{1,k}\mathfrak{a}_{2,\ell}k\ell \int_0^1 dz \sin(k\pi z)\sin(\ell\pi z) \int_{-|\mu|}^{|\mu|} dx\, e^{-2\pi k|x|} \int_0^\infty dy\, e^{-2\pi k|y|} \\
&= \frac{27\pi^3}{4}|\mu| \sum_{k=1}^\infty \mathfrak{a}_{1,k}\mathfrak{a}_{2,k}\cdot k^2 \cdot \int_{-|\mu|}^{|\mu|} dx\, e^{-2\pi k|x|} \int_0^\infty dy\, e^{-2\pi k|y|} \\
&= \frac{27\pi^3}{8}|\mu| \sum_{k=1}^\infty \mathfrak{a}_{1,k}\mathfrak{a}_{2,k}\left(1-e^{-2\pi k|\mu|}\right).
\end{aligned}
$$

Summing all three contributions, we finally arrive at:

$$
\mathcal{F}_{\text{dec.}}|_{d=5} = \frac{9}{8}N^2 P^2 \sum_{k=1}^\infty \mathfrak{a}_{1,k}\mathfrak{a}_{2,k}\cdot\frac{1}{k}\cdot\left(1-e^{-2\pi k|\mu|}\right)\cdot\left(2(\pi k|\mu|)^3 + 3(\pi k|\mu|)\right). \tag{C.9}
$$

In particular, the leading order term in the large $\mu$ limit is:

$$
\frac{9\pi^3}{4}N^2 P^2 \mu^3 \sum_{k=1}^\infty k^2 \mathfrak{a}_{1,k}\mathfrak{a}_{2,k}.
$$

### C.3.2 The contribution of massive fields in 3d

Let us now deal with the 3d case. The integrals involved are analogous to but simpler than the 5d case. In particular, we adopt *the same* integration domain for the EFT approach.

Starting with the fundamental hypermultiplets, we get

$$
\begin{aligned}
\frac{1}{N^2}\mathcal{F}_{\text{fund},1}^{\text{dec.}} &= \pi|\mu| \int_0^1 dz P^2 \zeta_2(z) \int_{-|\mu|}^{+|\mu|} dx\, \varrho_1(z,x) \\
&= \frac{\pi^2}{4}\sum_{k=1}^\infty \mathfrak{a}_{1,k}\mathfrak{a}_{2,k}k\cdot(2\pi k|\mu|)\cdot\left(1-e^{-2\pi k|\mu|}\right),
\end{aligned}
$$

and $\mathcal{F}_{\text{fund},2}^{\text{dec.}} = \mathcal{F}_{\text{fund},1}^{\text{dec.}}$ due to the invariance under exchange of the label $1 \leftrightarrow 2$.

For the other two contributions, we analyse the range of the eigenvalues for which the W-bosons are massive. Let us assume $\mu \gg 0$ without loss of generality. A reasoning akin to the 5d case yields

$$
\begin{aligned}
\frac{1}{N^2}\mathcal{F}_{\text{der}}^{\text{dec.}} &= 2 \cdot \pi|\mu| \int_0^1 dz \int_{-|\mu|}^{+|\mu|} dx \int_0^\infty dy \left[\frac{\varrho_1(z,x)\partial_z^2\varrho_2(z,y) + \varrho_1(z,y)\partial_z^2\varrho_2(z,x)}{2}\right] \\
&= -\pi^5|\mu| \sum_{k=1}^\infty \mathfrak{a}_{1,k}\mathfrak{a}_{2,k}k^4 \int_{-|\mu|}^{|\mu|} dx\, e^{-2\pi k|x|} \int_0^\infty dy\, e^{-2\pi k|y|} \\
&= -\frac{\pi^2}{4}\sum_{k=1}^\infty \mathfrak{a}_{1,k}\mathfrak{a}_{2,k}k\cdot(2\pi k|\mu|)\cdot\left(2-e^{-2\pi k|\mu|}\right).
\end{aligned}
$$

For $\mathcal{F}_0^{\text{dec.}}$, when $-\mu \leq x \leq \mu$, the value $y = x - \mu < 0$ is not consistent with the region $y > 0$, and likewise for the other contribution. Therefore, $\mathcal{F}_0^{\text{dec.}}\big|_{d=3} = 0$.

Summing all the contributions we find

$$\mathcal{F}_{\text{dec.}}|_{d=3} = \frac{\pi^2}{4}N^2 \sum_{k=1}^{\infty} \mathfrak{a}_{1,k}\mathfrak{a}_{2,k}k \cdot 2\pi k|\mu| \cdot \left(1 - e^{-2\pi k|\mu|}\right). \tag{C.10}$$

In particular, the leading order term in the large $\mu$ limit is:

$$\frac{\pi^3}{2}N^2|\mu| \sum_{k=1}^{\infty} \mathfrak{a}_{1,k}\mathfrak{a}_{2,k}k^2.$$

Along the way, we notice that these background contributions in 3d satisfy the analogue of the equilibrium condition:

$$\mathcal{F}_0^{\text{dec.}} + \mathcal{F}_{\text{der}}^{\text{dec.}} = -\frac{1}{2}\sum_{\alpha=1}^{2} \mathcal{F}_{\text{fund},\alpha}^{\text{dec.}}.$$

# D  F-theorem

In this appendix we provide the details of the proof of the stronger version of the F-theorem, for the RG flows we consider in the main text — the ones sketched in Figures 1 and 2 in the introduction. In Appendix D.1 we give a holographic proof of the F-theorem by studying the monotonicity of $c_{hol}$ as a function of $\sigma_0/P$. In Appendix D.2 we give an independent derivation of the same result from the field theory point of view.

## D.1  Holographic F-theorem with two rank functions

Let us consider the two quantities in (2.30) and (2.31). Generically we write

$$c_{hol}[\mathcal{R}_1, \mathcal{R}_2] = \gamma \sum_{k=1}^{\infty} k^{4-d}\left[R_{1,k}^2 + R_{2,k}^2 + 2R_{1,k}R_{2,k}e^{-\frac{k\pi\sigma_0}{P}}\left(1 + \frac{k\pi\sigma_0}{P}\right)\right]. \tag{D.1}$$

Here $\gamma > 0$ is a dimension-dependent positive number, not important in what follows. We define the two quantities,

$$K_1 = \lim_{\sigma_0 \to 0} c_{hol}[\mathcal{R}_1, \mathcal{R}_2] - \lim_{\sigma_0 \to \infty} c_{hol}[\mathcal{R}_1, \mathcal{R}_2] = 2\gamma \sum_{k=1}^{\infty} k^{4-d}R_{1,k}R_{2,k}, \tag{D.2}$$

$$K_2 = \frac{\partial}{\partial\sigma_0}c_{hol}[\mathcal{R}_1, \mathcal{R}_2] = -2\frac{\gamma\pi^2\sigma_0}{P^2}\sum_{k=1}^{\infty} k^{6-d}R_{1,k}R_{2,k}e^{-\frac{k\pi\sigma_0}{P}}. \tag{D.3}$$

We want to show that $K_1 \geq 0$ and $K_2 \leq 0$. To do this, we use the expressions for the Fourier transform of the rank functions without offsets ($N_0 = N_P = 0$), derived in (2.18),

$$R_{\alpha,k} = \frac{2P}{\pi^2 k^2}\sum_{j=1}^{P-1} F_{\alpha,j}\sin\left(\frac{k\pi j}{P}\right), \qquad \alpha = 1, 2. \tag{D.4}$$

We rewrite $K_1$ and $K_2$ in D.2 and D.3, using the special functions in Appendix A, as

$$K_1 = -\frac{\gamma P^2}{\pi^2}\sum_{j=1}^{P-1}\sum_{\ell=1}^{P-1} F_{1,j}F_{2,\ell}\left[G_d^c\left(0, \frac{\pi}{P}(j+\ell)\right) - G_d^c\left(0, \frac{\pi}{P}(j-\ell)\right)\right], \tag{D.5}$$

$$K_2 = \gamma\sigma_0\sum_{j=1}^{P-1}\sum_{\ell=1}^{P-1} F_{1,j}F_{1,\ell}\left[G_{d-2}^c\left(\frac{\sigma_0}{P}, \frac{\pi}{P}(j+\ell)\right) - G_{d-2}^c\left(\frac{\sigma_0}{P}, \frac{\pi}{P}(j-\ell)\right)\right]. \tag{D.6}$$

To show that $K_1$ is positive and $K_2$ is negative, it is thus sufficient to show that the function

$$G_d^{\text{dif}}(\sigma, v, w) := G_d^{\text{c}}(\sigma, v+w) - G_d^{\text{c}}(\sigma, v-w). \tag{D.7}$$

is negative for $v, w \in (0,1)$ and $\sigma > 0$. If $v + w < 1$ we can show this by first using that $v + w > v - w$ and then using that $G_d^{\text{c}}(\sigma, \eta)$ is a decreasing function in the second argument on $(0,1)$, because $\partial_\eta G_d^{\text{c}}(\sigma, \eta) = -\pi G_{d-1}^{\text{s}}(\sigma, \eta)$ and $G_{d-1}^{\text{s}}(\sigma, \eta)$ is a positive function for $\eta \in (0,1)$. If $v + w > 1$, we first use the the periodicity and even property of $G_d^{\text{c}}$ to write

$$G_d^{\text{dif}}(\sigma, v, w) = G_d^{\text{c}}(\sigma, 2-v-w) - G_d^{\text{c}}(\sigma, v-w). \tag{D.8}$$

Because $v < 1$, we know $2 - v - w = (2-2v) + (v+w) > v + w$, and we can use the same argument that $G_d^{\text{c}}(\sigma, \eta)$ is decreasing on $\eta \in (0,1)$. As a side remark, a corollary of the negativity of $G_d^{\text{dif}}(\sigma, v, w)$ is that the potential $\hat{W}(\sigma, \eta)$ of eq. (A.4), without offsets, is also negative.

## D.2 F-theorem in mass-deformed SCFTs

Our starting point is the definition (3.31), which we recall:

$$\mathcal{F}_{\text{EFT}} = \mathcal{F} - \mathcal{F}_{\text{dec.}} .$$

For $\mathcal{F}$ we use expression (3.15), and we refer to Appendix C for a proper definition of the contribution $\mathcal{F}^{\text{dec.}}$ from the fields that decouple at the end of the RG flow. Without loss of generality, we also assume $\mu > 0$.

As explained in Subsection 3.5, for an RG flows that splits the quiver $\mathcal{Q} \to \mathcal{Q}_1 \sqcup \mathcal{Q}_2$, this effective free energy satisfies

$$\lim_{\mu \to 0} \mathcal{F}_{\text{EFT}} = \mathcal{F}[\mathcal{Q}], \qquad \lim_{\mu \to \infty} \mathcal{F}_{\text{EFT}} = \mathcal{F}[\mathcal{Q}_1] + \mathcal{F}[\mathcal{Q}_2].$$

The part of $\mathcal{F}_{\text{EFT}}$ we are interested in is the one carrying the mass dependence, namely $\mathcal{F}_{\text{EFT}}^{\text{int}} = \mathcal{F}_{\text{EFT}} - \mathcal{F}[\mathcal{Q}_1] - \mathcal{F}[\mathcal{Q}_2]$.

We have

$$\mathcal{F}_{\text{EFT}}^{\text{int}} = \frac{(-1)^{\frac{d-3}{2}}}{2} N^2 P^{d-3} \int_0^1 dz \, \zeta_2(z) \left[ \int_{-\infty}^{+\infty} dx \, \varrho_1(z,x) f_{\text{h}}(x-\mu) - \int_{-\mu}^{+\mu} dx \, \varrho_1(z,x) f_{\text{h}}(\mu) \right]$$
$$+ (1 \leftrightarrow 2).$$

The two contributions with subscripts 1 and 2 swapped are equal, hence we trade the $(1 \leftrightarrow 2)$ term for a factor of 2. Next, we use (3.8) written in the form

$$f_{\text{h}}(x-\mu) = C(-1)^{\frac{d-3}{2}} \pi |x-\mu|^{d-3}, \qquad C = \begin{cases} 1, & d = 3, \\ \frac{9}{2}, & d = 5. \end{cases}$$

Then, to simplify the absolute values, we use the elementary rewriting $\int_{-\infty}^{+\infty} dx = \int_{-\infty}^{\mu} dx + \int_{\mu}^{\infty} dx$ and $\int_{-\mu}^{+\mu} dx = \int_{-\infty}^{\mu} dx - \int_{-\infty}^{-\mu} dx$.
These manipulations give

$$\frac{\mathcal{F}_{\text{EFT}}^{\text{int}}}{CN^2 P^{d-3}} = \pi \int_0^1 dz \, \zeta_2(z) \left[ \int_{-\infty}^{+\infty} dx \, \varrho_1(z,x) |x-\mu|^{d-3} - |\mu|^{d-3} \int_{-\mu}^{+\mu} dx \, \varrho_2(z,x) \right]$$

$$= \pi \int_0^1 dz \, \zeta_2(z) \left( \int_{-\infty}^{\mu} dx \, \varrho_1(z,x) \left[ (\mu-x)^{d-2} - \mu^{d-2} \right] \right.$$

$$\left. + \int_{\mu}^{\infty} dx \, \varrho_1(z,x) \left[ (x-\mu)^{d-2} + \mu^{d-2} \right] \right),$$

where in the second line we have used $\varrho_1(z,-x) = \varrho_1(z,x)$.

Focusing now on $d = 3$, we obtain

$$\left. \frac{\mathcal{F}_{\text{EFT}}^{\text{int}}}{N^2} \right|_{d=3} = \pi \int_0^1 \mathrm{d}z\, \zeta_2(z) \left[ -\int_{-\infty}^{\mu} \mathrm{d}x\, x \varrho_1(z,x) + \int_{\mu}^{\infty} \mathrm{d}x\, x \varrho_1(z,x) \right]. \qquad (D.9)$$

Setting $\mu = 0$ we have

$$\left. \frac{\mathcal{F}_{\text{EFT}}^{\text{int}}}{N^2} \right|_{d=3} = \pi \int_0^1 \mathrm{d}z\, \zeta_2(z) \int_{-\infty}^{+\infty} |x| \varrho_1(z,x) \mathrm{d}x > 0\,,$$

which is the integral of positive definite quantities. This immediately implies $\mathcal{F}[\mathcal{Q}] > \mathcal{F}[\mathcal{Q}_1] + \mathcal{F}[\mathcal{Q}_2]$.

Moreover, differentiating (D.9) we find

$$\left. \partial_\mu \frac{\mathcal{F}_{\text{EFT}}^{\text{int}}}{CN^2} \right|_{d=3} = \pi \int_0^1 \mathrm{d}z\, \zeta_2(z) \left[ -2\mu \varrho_1(z,\mu) \right] \le 0\,.$$

The last function is negative semi-definite: there is a factor $-2$ multiplying the integral of non-negative quantities. Besides,

$$\lim_{\mu \to 0} \mu \varrho_1(z,\mu) = 0\,, \qquad \lim_{\mu \to \infty} \mu \varrho_1(z,\mu) = 0\,,$$

which directly implies $\partial_\mu \mathcal{F}_{\text{EFT}} = 0$ at the CFT points, proving that $\mathcal{F}_{\text{EFT}}$ is stationary there.

Let us now consider $d = 5$. By similar rewriting of the integration domain, we find

$$\left. \frac{\mathcal{F}_{\text{EFT}}^{\text{int}}}{N^2 P^2} \right|_{d=5} = \frac{9\pi}{2} \int_0^1 \mathrm{d}z\, \zeta_2(z) \left[ -2 \int_0^{\mu} \mathrm{d}x\, \varrho_1(z,x) \left( x^3 - 3\mu x^2 + 3\mu^3 x \right) \right]. \qquad (D.10)$$

As above, at $\mu = 0$ we obtain a positive quantity, thus $\mathcal{F}[\mathcal{Q}] > \mathcal{F}[\mathcal{Q}_1] + \mathcal{F}[\mathcal{Q}_2]$. Differentiating with respect to $\mu$, we find

$$\left. \partial_\mu \frac{\mathcal{F}_{\text{EFT}}^{\text{int}}}{N^2 P^2} \right|_{d=3} = -9\pi \int_0^1 \mathrm{d}z\, \zeta_2(z) \left[ \mu^3 \varrho_1(z,\mu) + 3 \int_0^{\mu} x(2\mu - x) \varrho_1(z,\mu) \right] \le 0\,,$$

again with a negative overall coefficient multiplying a non-negative integral.

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
