# Peer review of "Matrix Models and Holography: Mass Deformations of Long Quiver Theories in 5d and 3d"

_SciPost Physics, doi:SciPost Phys. 15, 086 (2023)_

## Round 1 · Referee Report · Anonymous (Referee 1) · 2023-5-22

Report

The authors improved the manuscript by correcting typos and clarifying several points. The manuscript may be publishable if further improvements are made in a transparent way.

1) While the 3d partition function obtained from the localization is originally defined for real FI and mass parameters, one can turn on the imaginary parts of them in the physical strip of the complex plane. By analytic continuation, it may be well-defined beyond that. Then the complex components of mass parameters and scalars in the vector multiplet can appear. This observation can be found in the reference [138] and it would become more crucial when one evaluates the Coulomb branch correlation functions which can be more general observables for the dictionary between matrix models and the (twisted) holographic dual as the sphere partition function is identified with a trivial example of the correlation function of the identity operator.

9) Still it is not clear to see the authors' claim that one can get a finite partition function in the limit $m\rightarrow \infty$ by adding a counterterm. As the original matrix integral is well-defined without any counterterm, I cannot see any reason to introduce it. Also once the additional term is introduced, it may modify the original theory. As long as the factor $e^{-\frac{\pi}{2} m F_1F_2}$ remains in (C.7), it would vanish in the limit $m\rightarrow \infty$ and it does not seem to be a finite partition function for the product of the decoupled theories. If it vanishes, it may indicate that the limit $m\rightarrow \infty$ of the partition function does not make sense. They should explicitly demonstrate what the counterterm is and how the counterterm and the finite partition function are obtained so that the $F_2=2N_2$ term diverges as $m\rightarrow \infty$. Mathematically the integral vanishes, so they should at least explain the physical interpretation. The situation does not seem to be so simple, as the factor $e^{-\frac{\pi}{2} m [...]}$ is not overall factor which may be cancelled by the counterterm but rather it differs for each term labeled by distinct $N_2$. Also it is not clear what occurs when the given $F_1$ is odd for which $F_2=2N_2$ does not hold.

  • validity: -
  • significance: -
  • originality: -
  • clarity: -
  • formatting: -
  • grammar: -

Author:  Leonardo Santilli  on 2023-05-26  [id 3684]

(in reply to Report 1 on 2023-05-22)
Category:
reply to objection

The Referee writes "If it vanishes, it may indicate that the limit $m \to \infty$ of the partition function does not make sense."
We do not understand what this comment means, since the $m \to \infty$ limit of every partition function vanishes. Taking the Referee's objection literally would mean that every theory with massive hypermultiplets "does not make sense".
We attach an extensive list of simple examples to this comment (even beyond the linear quivers considered in the paper), and encourage the Referee to plot the partition function of their favourite supersymmetric theory. They will observe that it is exponentially damped at $\lvert m \rvert \to \infty$.
We provide a more exhaustive answer to the other queries in the resubmission.

Attachment:

plots-of-Z-massive.pdf

---

## Round 1 · Author Response

We thank both Referees for their careful reading of our work.
We have edited the manuscript, amended the typos and introduced additional clarifications, following the input of one of the Referees. We answer to the Referee's points in the section "List of changes".

---

## Round 1 · List of Changes

We thank the Referee for the review and the additional comments and corrections. We have edited the manuscript taking them into account. We answer point by point below.

1) Our choice of mass deformation, and the way it is enforced, is explained in detail in Subsection 3.1.3. Our claim is that we couple the hypers to a background 3d $\mathcal{N}=4$ vector multiplet in the canonical way. The scalar in this background vector multiplet is given a vev. With respect to the $\mathcal{N}=2$ subalgebra fixed by localization, the vev splits into a real mass and a complex mass. Our choice of mass deformation is that the real mass is $m$, and the complex mass is 0, where $m$ is a real number and 0 is a complex number. The quantities we aim to compute are functions of the real mass $m$, and are insensitive to the complex mass. The dependence on the complex mass appears only through to the constraints imposed by a superpotential. This fact can be seen for instance in the exact evaluations given in the papers of Benvenuti-Pasquetti (ref. [135]) or Gaiotto-Okazaki (ref. [138]), or in Willett's review (ref. [108]) cited in Subsection 3.1.3. The goal of our work is to study a particular deformation by real masses, not a superpotential deformation. This can be done without breaking any supersymmetry. The study of superpotential deformations is a separate problem that we do not address in the present paper.

If the Referee disagrees with the explanation of Subsection 3.1.3, it would be helpful for us if they could pinpoint what they believe is wrong there. If the Referee is still unsatisfied with our explanation, it would be helpful for us if they could explain to us in more concrete terms what they mean by studying $\mathcal{F}$ as a function of the complex masses. 2) In Figures 1 and 2, the horizontal arrows are meant to be the same in the CFT and gauge theory. They pictorially indicate the entire flow, which starts at $m=0$ and ends at $m=\infty$ in both cases. We have made it more explicit in the figures, indicating these values as suggested by the Referee. 3) Following the Referee's suggestion, we have indicated the $m \to \infty$ limit at the end of the arrow that represents the RG flow in Figure 8. 4) Misprint corrected, thanks for noticing it. 5) We have fixed the factors of $1/N!$ in Eq.(C.1) and subsequent. Thank you. 6) To answer to this point, we would like to emphasize that the detour in Appendix C is to exhaustively and explicitly explain in a simple example how to figure out what is the CFT at the endpoint of the RG flow. We of course do not claim any factorization at finite $m$. As usual, we start with a CFT and deform it by a relevant operator, triggering an RG flow. At every finite $m$, the way of thinking of this model is as a QFT which is the deformation of the UV CFT. In general, one would like to know what is the endpoint of an RG flow. Here, we use the power of supersymmetry and localization to answer this question. The idea of exploring different candidate IR fixed points using changes of variables in the matrix integral was first used in Ref. [93].

This is the philosophy of Appendix C. To answer the question of what happens at $m\to \infty$, we do not need to carefully keep track of all the terms. We only need to understand which ones are not killed by the limit and keep track of them. To use the approximation in the equation for $S_{\text{int}}$ below (C.5) we split the integral into a sum of contributions. One of them is the integral over $-m < \sigma_a < m$ and $-m < \phi_{\dot{a}} < m$, which becomes an integral over the two Cartan subalgebras of $\mathfrak{u}(N_1)$ and $\mathfrak{u}(N_2)$ as $m \to \ifnty$. All the other terms vanish in the large $m$ limit, due to the domain shrinking to zero measure. These terms vanish faster than the one we retain: terms $e^{- \pi \lvert \sigma -m \rvert - \pi \lvert \sigma +m \rvert }$ are more suppressed than $e^{-2\pi m}$ if $\sigma >m$ or $\sigma <-m$, and, in addition, the integration domain shrinks. These terms are doubly-suppressed and can be safely neglected in the limit $m \to \infty$. We added a comment below (C.5) to clarify this point. 7) Eq. (C.6) is obtained as follows: We are making $(N+1)$ different changes of variables in the same integral, which is the UV partition function. We thus obtain $(N+1)$ different-looking integrals that are in fact the same partition function. We sum all of them and divide by $N+1$, and get again the same result. Eq (C.6) reads $\mathcal{Z}= (\mathcal{Z}+\mathcal{Z}+...+\mathcal{Z})/(N+1)$, where the notation $\mathcal{Z}[N_2]$ on the r.h.s. of (C.6) is to remember which change of variables we use in each summand. $\mathcal{Z}[N_2]$ is given in (C.5), and the $[N_2]$ notation is to insist that we are making one choice of $N_2$. We have improved the definition and notation in Eq. (C.5) in accordance with the Referee's comment. No approximation whatsoever is made in (C.6). 8) We have corrected the missing factor of 1/2 in passing from the equation below (C.5) to the equation above (C.6). Thanks a lot for pointing it out. 9) Yes, (C.7) becomes 0 in the limit $m \to \infty$. This is a generic feature of massive QFTs, and is cured by appropriately adjusting the regularization scheme. It is straightforward to check, for instance, that also the partition function of SQED3 with $F$ flavours vanishes if $m\to \infty$. This follows immediately from expressions (2.13)-(2.16) of Benvenuti-Pasquetti (ref. [135]) or (4.19) of Gaiotto-Okazaki (ref. [138]). The various terms in the sum (C.7) will in general go to 0 in different ways as $m \to \infty$. Thanks to our rewriting, we identify the term that is less suppressed.

What we want to conclude is that we must focus on this term as $m \to \infty$, since we can get a finite partition function out of it by adding a counterterm in the standard way. The other way around: imagine we pick a summand randomly in the sum in (C.7) and add a counterterm to the action so that the randomly chosen term is finite in the limit $m \to \infty$. Then, the $F_2=2N_2$ term will diverge as $m \to \infty$ due to the counterterm. 9bis) The factor of 2 is due to the fact that we are giving mass $+m$ to one hyper and mass $-m$ to another hyper, as per the first equation of Appendix C.2.5. Also, we are not subtracting the free energy $2\ln (2\cosh (\pi m))$, but subtracting the scheme-dependent part using a gravitational counterterm with coefficient $m$ if $m>0$ and $-m$ if $m<0$. Subtracting $2\ln (2\cosh (\pi m))$ would not correspond to such a counterterm (see e.g. ref. [124] of our manuscript), and moreover would not interpolate between the correct values. Our definition of $\mathcal{F}_{\text{EFT}}$ is of the form $\mathcal{F} $ minus local, gauge-invariant counterterms, which is what we are after. 10) We have improved the caption to Figure 12 to amend the lack of clarity pointed out by the Referee. The UV and IR CFTs in the plot are ($g_{YM}^2 \to \infty$ fixed points of) SQED with $F$ and $F-2$ flavours, respectively. We denote by $\mathcal{F}_{\text{UV}}$ and $\mathcal{F}_{\text{IR}}$ the respective free energies.

---

## Round 2 · Referee Report · Anonymous (Referee 1) · 2023-5-27

Report

I thank the authors for clarifying all the issues I raised. I recommend the manuscript for publication.
  • validity: -
  • significance: -
  • originality: -
  • clarity: -
  • formatting: -
  • grammar: -

Author:  Leonardo Santilli  on 2023-07-14  [id 3808]

(in reply to Report 1 on 2023-05-27)

We are grateful to the Referee for their deep analysis of the manuscript and the several suggestions.

---

## Round 2 · Author Response

1) We thank the Referee for the suggestion. We have shown in the last paragraphs of Subsection 3.3.3 that the complexification of the real mass does not modify the free energy we compute. We note, however, that in this report the Referee is talking about the complexification of the real mass, which is a different thing than the complex masses the Referee alluded to in their previous reports. A complex mass combines with the real $m$ to form an $SU(2)$ triplet. The holomorphy in $m$ the Referee discusses in their latest report, on the contrary, was first discovered by Jafferis [1012.3210] in the context of 3d $\mathcal{N}=2$ theories, which do not admit complex masses at all, since the R-symmetry is $U(1)$ and not $SU(2)$. Notwithstanding, we have seriously and carefully addressed the latest Referee's concern in the manuscript.

2) We thank the Referee for the comment. We have added clarifications on the scheme-dependence at the end of Subsection C.2.4, where the counterterm is explicitly discussed, and therein we refer to the works of Jafferis-Klebanov-Pufu-Safdi (Ref. [98,99]) and Gerchkovitz-Gomis-Komargodski (Ref.[125]). We have also added further clarifications around Eq. (C.6), including a comment on the case of odd $F_1$.

The other comments of the Referee are based on certain misguided assumptions. -- The Referee writes "If it vanishes, it may indicate that the limit $m \to \infty$ of the partition function does not make sense." We do not understand this comment, since the $m \to \infty$ limit of every partition function vanishes. It would seem that the Referee suggests that it does not make sense to give a large mass to hypermultiplets, in any theory. In the answer to the report on the previous submission, we attached some simple examples (even beyond the linear quivers considered in the paper), and encourage the Referee to plot the partition function of their favourite supersymmetric theory. They will observe that it is exponentially damped at $m \to \infty$. -- In the second sentence of the report, the Referee writes "As the original matrix integral is well-defined without any counterterm, I cannot see any reason to introduce it." This is not entirely true. The localized partition function is defined assuming some regularization scheme. For instance, it is known that there is a parity anomaly in 5d. When computing the partition function, one needs to make a choice of whether to preserve parity, at the expense of invariance under background gauge transformation, or save the latter at the expense of parity. Both choices are well-defined, and differ by a counterterm. This argument applies to every anomaly of global symmetries, mixed gravitational-global and mixed R-global anomaly. These facts are discussed in the references [121,122] (for the cases of interest to us), which we cited in the main text since v1. Phrased differently, the choice of regularisation of the one-loop determinants that leads to the localized partition function is a choice of regularisation scheme. Therefore, the partition function has a certain scheme-dependence built in. When the Referee writes that "the original matrix integral is well-defined without any counterterm" probably means that it is well-defined with the counterterms assumed by the authors who computed the localized partition function. This is true, but it does not exclude that any other choice of regularisation scheme is equally valid. -- The Referee also writes "once the additional term is introduced, it may modify the original theory." We do not understand this comment. The fact that we can cancel the exponential suppression with a scheme-dependent counterterm was shown in the references [98,99,125], which we adequately cite at various points in the manuscript. We have added a further clarifying comment in Appendix C.2.4. The fact that adding scheme-dependent counterterms does not change the physical theory is textbook material, and it essentially boils down to the definition of counterterm. -- The sentence "The situation does not seem to be so simple, as the factor $e^{- \frac{\pi m}{2} [...]}$ is not overall factor which may be cancelled by the counterterm but rather it differs for each term labeled by distinct $N_2$". We totally agree: that one is not an overall factor. This was the whole point of the discussion: to figure out which quivers survive in the deep IR. As we have already explained below Eq. (C.7) as well as in our previous reply, we can save the least-suppressed term with a counterterm. The terms with other choices of $N_2 \ne F_2/2$ are more suppressed and vanish in the limit $m \to \infty$. We are left with a pair of balanced quivers, as claimed. That sentence of the Referee's report is correct and indeed supports our claim that only one choice of $N_2$ survives at $m \to \infty$.

---

## Round 2 · List of Changes

1) We have shown in Subsection 3.3.3 that the complexification of the real mass does not modify the free energy. 2) We have added clarifications on the scheme-dependence at the end of Subsection C.2.4, where the counterterm is explicitly discussed. We have also added further clarifications around Eq. (C.6), including a comment on the case of odd $F_1$. 3) Corrected two typos.

---

## Editorial Decision

published